PREPARED FOR SUBMISSION TO JHEP

# Minimal $(D, D)$ conformal matter and generalizations of the van Diejen model

**Belal Nazzal,**[a] **Anton Nedelin,**[a] **Shlomo S. Razamat** [a,b]

[a]*Department of Physics, Technion, Haifa, 32000, Israel*
[b]*School of Natural Sciences, Institute for Advanced Study, Princeton NJ, USA*

*E-mail:* sban@campus.technion.ac.il, anton.nedelin@gmail.com, razamat@physics.technion.ac.il

ABSTRACT: We consider supersymmetric surface defects in compactifications of the 6d minimal $(D_{N+3}, D_{N+3})$ conformal matter theories on a punctured Riemann surface. For the case of $N = 1$ such defects are introduced into the supersymmetric index computations by an action of the $BC_1 (\sim A_1 \sim C_1)$ van Diejen model. We (re)derive this fact using three different field theoretic descriptions of the four dimensional models. The three field theoretic descriptions are naturally associated with algebras $A_{N=1}$, $C_{N=1}$, and $(A_1)^{N=1}$. The indices of these 4d theories give rise to three different Kernel functions for the $BC_1$ van Diejen model. We then consider the generalizations with $N > 1$. The operators introducing defects into the index computations are certain $A_N$, $C_N$, and $(A_1)^N$ generalizations of the van Diejen model. The three different generalizations are directly related to three different effective gauge theory descriptions one can obtain by compactifying the minimal $(D_{N+3}, D_{N+3})$ conformal matter theories on a circle to five dimensions. We explicitly compute the operators for the $A_N$ case, and derive various properties these operators have to satisfy as a consequence of 4d dualities following from the geometric setup. In some cases we are able to verify these properties which in turn serve as checks of said dualities. As a by-product of our constructions we also discuss a simple Lagrangian description of a theory corresponding to compactification on a sphere with three maximal punctures of the minimal $(D_5, D_5)$ conformal matter and as consequence give explicit Lagrangian constructions of compactifications of this 6d SCFT on arbitrary Riemann surfaces.

## 1 Introduction

Supersymmetric quantum field theories (SQFTs) provide a plethora of interesting interconnections between various subjects in mathematical physics: or quoting L. Tolstoy, *"Happy families are all alike; every unhappy family is unhappy in its own way."*. The *happy family* subjects of the ilk of supersymmetric QFTs include, among others, two dimensional conformal field theories and integrable quantum mechanical models.

Here we will focus on one such connection: relation between six dimensional $(1,0)$ superconformal field theories (SCFTs) and one dimensional elliptic relativistic quantum integrable models. This relation takes many guises, with one of the more notorious studied by Nekrasov and Shatashvili [1] which goes through an intermediate five dimensional step and involvs

eight supercharges. A way to think about the relation is through surface defects in four dimensional theories with only four supercharges which are obtained by compactifying a six dimensional SCFT on a (punctured) surface [2]. In principle for every 6d $(1,0)$ SCFT, such that upon circle compactification to five dimensions an effective five dimensional gauge theory exists (upon some proper choice of holonomies around the circle for the 6d global symmetry), one can associate a one dimensional integrable quantum mechanical system. This integrable system is related to introducing surface defects into the supersymmetric index [3] of the four dimensional theories obtained by compactifying the chosen 6d SCFT on a generic (punctured) surface preserving four supercharges.[1] This correspondence draws an interesting parallel between the classification program of 6d SCFTs [6, 7] and classification of elliptic relativistic quantum integrable systems.

There are various instances of this correspondence known by now. For example, in the case of the $(2,0)$ theory of type $G \in ADE$ the associated integrable model is given in terms of the Ruijsenaars-Schneider elliptic analytic difference operators (A$\Delta$Os)[2, 8]. [2] In the case of $A_2$ and $D_4$ minimal 6d SCFTs [19–21] one can derive novel integrable models [22, 23] associated to the $A_2$ and $A_3$ root systems respectively. In the case of the 6d SCFT being the rank one E-string [24] one obtains [25] the $BC_1$ van Diejen model [26].[3]

Another interesting question is to compile the dictionary between compactifications of 6d SCFTs and 4d Lagrangian theories. Such a dictionary is completely and explicitly known starting with a handful of 6d SCFTs. For example: $A_1$ $(2,0)$ gives rise to 4d quiver theories built from tri-fundamentals of $SU(2)$s [33]; minimal $A_2$ 6d SCFT gives rise to quivers built from tri-fundamentals of $SU(3)$s [21]: rank one E-string theory gives rise to generalized quiver theories built from $SU(3)$ SQCD with $N_f = 6$ and deformations thereof [34]. In other cases one can construct the relevant theories in 4d starting with weakly coupled Lagrangians but gauging symmetries emergent either in the IR or at some loci of the conformal manifolds [24, 35–37]. An example of the latter construction which will be relevant for us here is that of minimal $(D_{N+3}, D_{N+3})$ conformal matter with $N > 1$ [34, 38].[4] However, starting from a generic 6d SCFT an explicit Lagrangian construction in 4d, and even whether it in principle exists, is not not known at the moment. Those models for which a Lagrangian is not known at the moment are often referred to as "non-Lagrangian". A major motivation of this program is that once the dictionary is compiled many interesting strong coupling effects,

---

[1]In principle the five dimensional intermediate step might not be needed and one could be able to derive the integrable models studying defects in 6d, see [4]. See also for another five dimensional discussion [5].

[2]In general indices in compactification scenarios can be also associeate to $2d$ topological field theories [9]. In turn these are long known to be related to integrable models by themselves [10]. In particular the indices of the compactifications of $A$ type $(2,0)$ theory give rise to $2d$ q-deformed YM theory [11, 12]. See also [13–18] for relevant discussions.

[3]Naturally, for rank $Q$ E-string theory one would expect to obtain the $BC_Q$ van Diejen model. This was not shown explicitly yet, but the results of [27–29] should be helpful to establish this relation. Recently the $Q = 1$ relation was also derived [30] directly in six dimensions by studying the relevant Seiberg-Witten curve [31, 32].

[4]In [38] general compactifications of the 6d SCFT residing on two M5 branes probing a $\mathbb{Z}_k$ singularity using such methods was also discussed. Moreover, many more examples of some special compactifications (such as on tori and/or spheres with special collections of punctures) are known, see $e.g.$ [27, 33, 39–45].

such as emergence of symmetry and duality, can be understood in terms of the consistency of the dictionary with the geometry behind the compactifications.

The purpose of this paper is to study yet another entry in the two dictionaries. On one hand we will start from the 4d theories obtained by compactifications of the minimal $(D_{N+3}, D_{N+3})$ conformal matter theories in 6d (with the E-string being the $N = 1$ case) recently constructed in [34, 38, 46] and will be interested in the consistency of the dictionary relating these models to the geometry defining the compactifications. In particular we will perform several checks of the dualities implied by the geometry.

On the other hand we will derive an infinite set of integrable models which are a generalization of the correspondence between rank one E-string and $BC_1$ van Diejen model ($N = 1$ above) corresponding to the minimal $(D_{N+3}, D_{N+3})$ conformal matter theories in 6d. This can be viewed as $A_N$ generalization of the $BC_1 \sim A_1$ van Diejen model. In fact there are yet two additional 4d descriptions known in terms of $USp(2N)$ and $SU(2)^N$ gauge theories which will give rise to a $C_N$ and an $(A_1)^N$ generalizations of the van Diejen model. Each one of these would lead to an additional set of integrable systems. The fact that we have *three* different quantum mechanical integrable models corresponding to the same 6d SCFT is related to the fact that these SCFTs have more than one effective quantum field theory description in five dimensions [47, 48]. The different descriptions are usually called dual (in the sense that they have same UV completion in 6d).

The paper is organized as follows. In Section 2 we review the technology of deriving integrable models from supersymmetric indices with surface defects and the type of properties these models have to satisfy following from the physics of the 4d models. In Section 3 we will apply this technology and discuss in detail three different derivations of the $BC_1$ van Diejen model starting from three different QFTs corresponding to compactifications of the rank one E-string theory on three punctured spheres. In Section 4 we discuss in detail a generalization to compactifications of the minimal $(D_{N+3}, D_{N+3})$ conformal matter theories and in particular derive the $A_N$ generalization of the $BC_1$ van Diejen model. In Section 5 we discuss our results as well as possible generalizations and extensions. Several appendices contain technical details of the computations presented in the bulk of the paper. In particular as an intermediate step of our constructions we discuss a four-punctured sphere for general $(D_{N+3}, D_{N+3})$ conformal matter. For the case of $N = 2$ this gives us an explicit and simple Lagrangian description for a sphere with three maximal punctures (one USp(4) and two SU(3)) which we discuss in Appendix C. This thus makes the compactifications of $(D_5, D_5)$ minimal conformal matter theories completely Lagrangian.

## 2 Integrable models from supersymmetric index and dualities

We begin by discussing a concrete way integrable models can be associated to a 6d SCFT via supersymmetric index computations in presence of surface defects. We will overview schematically the general logic and the readers can consult the original papers for the details and subtleties. A reader familiar with the procedure can skip this section.

Let us start from some 6d $(1,0)$ SCFT and assume it has global symmetry $G_{6d}$. We place this theory on a Riemann surface $\mathcal{C}$ and turn on background gauge fields, fluxes supported on $\mathcal{C}$, preserving four supercharges [37, 49], and then flow to four dimensions. We denote the 4d theories thus obtained by $\mathcal{T}[\mathcal{C}, \mathcal{F}]$. These models might be interacting SCFTs, free chiral fields, or even contain IR free gauge fields: this will not be essential for our discussion. The Riemann surface might have punctures. In general there can be various types of punctures which can be understood and classified using several methods (see $e.g.$ [33, 50, 51]). For 6d theories that have 5d effective gauge theory description with gauge group $G_{5d}$ once they are compactified on a circle with a proper choice of holonomies, there is a natural choice of a puncture, usually called maximal. This choice amounts to specifying certain supersymmetric boundary conditions for the 5d fields, and in particular setting Dirichlet boundary condition for the 5d gauge fields at the puncture. This equips the 4d effective theory with additional factors of flavor symmetry $G_{5d}$ associated to each puncture. Certain 5d fields which are assigned with Neumann boundary condition give rise to natural 4d chiral operators (which we will denote by $M$) charged under $G_{5d}$: by abuse of notation we will refer to these fields as *moment maps*.[5] The choice of flux and choices of boundary conditions break $G_{6d}$ to a subgroup, with a generic choices leaving only the Cartan generators, $C[G_{6d}]$. Note that given an effective 5d description the maximal puncture might not be unique as it involves choices of boundary conditions. Naively all these choices are equivalent, but for a surface with several punctures the relative differences are important. Such differences are usually called different colors of maximal punctures (see $e.g.$ [24, 37]). Moreover, in certain cases there can be different five dimensional effective gauge theories depending on the choice of holonomies giving rise to maximal punctures with different symmetry groups. This will be important for us and we will see explicit examples in this paper.

One can obtain a plethora of other types of punctures by giving vacuum expectation values (vevs) to the moment map operators, breaking sequentially $G_{5d}$ to sub-groups: this procedure is called partially closing the puncture (see $e.g.$ [52, 53]). Turning on sufficient number of vevs $G_{5d}$ can be completely broken in which case one says that the puncture is closed. Importantly, the moment maps receiving the vevs might also be charged under the 6d symmetry $G_{6d}$. The theory obtained by completely closing a punctures then can be associated to the Riemann surface of the same genus but with one puncture less than the theory we started with, and with the flux $\mathcal{F}$ shifted by amount related to the charges of the moment maps (see [24, 34, 37] for details). Puncture which can be obtained in this partial closure procedure which only can be further completely closed is called *minimal*. Typically, the rank of the symmetry of such a puncture is one ($U(1)$ or $SU(2)$). [6]

---

[5]The motivation for this notation is that in the case of the 6d theory being the $(2,0)$ SCFT such chiral operators are indeed the moment maps inside $\mathcal{N}=2$ conserved current multiplet. However, as our models will be only $\mathcal{N}=1$ supersymmetric, the moment map operators we will discuss do not have such a group theoretic meaning.

[6]See however [22] where the maximal puncture with $SU(3)$ symmetry can be only broken to no-symmetry puncture. The punctures carrying no symmetry typically either support a discrete twist line or are not obtainable by closing a maximum puncture (are irregular). The example of [22] has a twist supported by the puncture.

Starting from two theories corresponding to compactifications on two surfaces, which might be different but each has at least one maximal puncture, one can build the theory corresponding to the surface glued along the maximal punctures. Field theoretically there are two procedures one can perform. First one is called *S-gluing* and it amounts to gauging the diagonal combination of the two puncture symmetries and turning on a superpotential involving the moment maps of the two theories, $W \sim M \cdot M'$. In this case the flux associated to the resulting surface is the difference of the fluxes of the constituents. Note that the overall sign of flux is immaterial. Second procedure is called $\Phi$-gluing and it involves first adding chiral superfields $\Phi$ in a representation under $G_{5d} \times C[G_{6d}]$ conjugate to the one of $M$, turning on superpotential $W \sim M \cdot \Phi - M' \cdot \Phi$, and gauging the diagonal combination of $G_{5d}$. In this case the flux of the resulting surface is the sum of the two fluxes. One can also consider a combination of these two gluing, S-gluing for some components of the moment maps and $\Phi$-gluing for the rest, as long as the gauging is non anomalous.

We assume that we have an *explicit* Lagrangian construction[7] of at least one compactification corresponding to a sphere with two maximal and one minimal punctures and some value of flux. Without loss of generality we will assume that the flux is such that there is a preferred choice of a vev to a moment map (will be denoted $\hat{M}^-$) such that one closes the minimal puncture and obtains (formally) the theory associated to a two punctured sphere with zero flux.[8] We will denote this surface by $\mathcal{T}^{\mathcal{A}}_{z,u,\hat{a}}$ with $z$ and $u$ being parameters of $G_{5d}$ associated to the two maximal punctures and $\hat{a}$ a $U(1)$ parameter associated to the minimal puncture. We denote by $\mathcal{A}$ the flux of this surface.

Given the above, the derivation of the integrable models proceeds by considering the supersymmetric index [3, 54–56] of the theories obtained in the compactification. The supersymmetric index in four dimensions is defined as a supersymmetric trace over the Hilbert space of the $\mathcal{N} = 1$ theory quantized on $\mathbb{S}^3$,

$$\mathcal{I} = \mathrm{Tr}_{\mathbb{S}^3}(-1)^F q^{j_2 - j_1 + \frac{1}{2}R} p^{j_2 + j_1 + \frac{1}{2}R} \prod_{i \in C[G_{4d}]} u^{q_i} . \tag{2.1}$$

Here $F$ is the fermion number, $j_i$ are the Cartans of the $Spin(4)$ isometry of $\mathbb{S}^3$, $R$ is the R-symmetry, and $q_i$ are charges under the Cartan sub-group of the 4d Symmetry group $G_{4d}$. This is the most general Witten index one can define preserving a certain chosen supercharge (and its superconformal Hermitean conjugate). For more details the reader can consult [57]. Importantly, the index always depends on two parameters $q$ and $p$, and can depend on additional fugacities associated to the global symmetry a given theory has. Moreover, being a Witten index [58], it does not depend on continuous parameters of the theory. In particular, it does not depend on the RG scale [56] and if a theory has continuous couplings parametrizing a conformal manifold, the index will not depend on those, and if there is a duality group acting on this conformal manifold the index will be thus duality

---

[7]For the purpose of deriving the integrable models in fact a description which starts from weakly coupled fields but involves gauging of emergent symmetries is sufficient. See *e.g.* [25].

[8]If the flux of a given three punctured sphere does not satisfy this criterion it is easy to build from it a three punctured sphere which will. For example, one glues two trinions together with S-gluing and closes one of the minimal punctures. We will utilize this idea later in the paper.

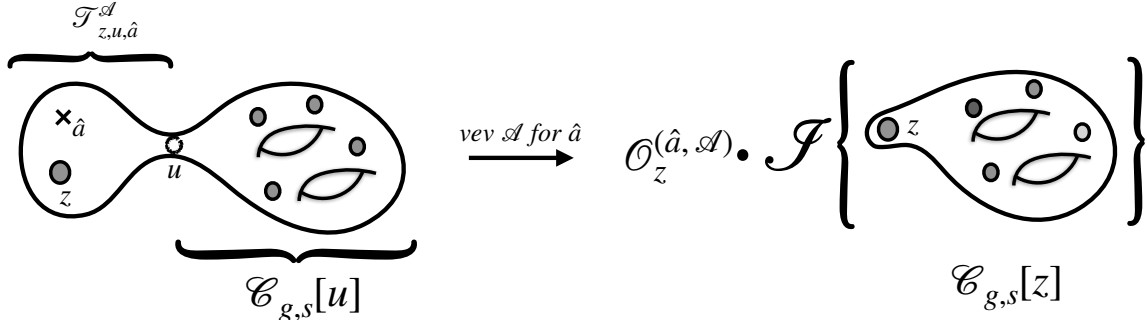

**Figure 1**. Derivation of the AΔO from the index. We denote both the flux of the three punctured sphere we glue in and the preferred choice of the operator we give the vev to by $\mathcal{A}$, as the resulting two punctured sphere has zero flux.

invariant. Thus, given a 4d theory arising in a compactification we can compute the index $\mathcal{I}[\mathcal{C}, \mathcal{F}]$ which will be determined by the geometry and the fluxes and will depend on $q$, $p$ and the fugacities for the global symmetries. The 4d theories *are determined by the geometry* and in particular different ways of constructing the same geometry, such as different pair-of-pants decompositions and different ways to distribute the flux among the pairs of pants, lead to equivalent dual theories. *The index thus will be invariant of the different ways we construct the geometry.* Given the indices of two theories one can construct the index corresponding to glued surface by integrating over the fugacities corresponding to the gauged symmetry,

$$\mathcal{I}\Big[\mathcal{C} \oplus \mathcal{C}', \mathcal{F} \pm \mathcal{F}'\Big] = \oint \prod_{i=1}^{\text{rank}\,G_{5d}} \frac{dz_i}{2\pi i z_i} \Delta_{Haar}(G_{5d}) \Delta^{S/\Phi}(z_{5d}; u_{6d}; q, p) \times \tag{2.2}$$

$$\mathcal{I}\Big[\mathcal{C}, \mathcal{F}\Big](z_{5d}, u_{6d}, \cdots; q, p) \times \mathcal{I}\Big[\mathcal{C}', \mathcal{F}'\Big](z_{5d}, u_{6d}, \cdots; q, p)\,.$$

Here $\Delta^{S/\Phi}(z_{5d}; u_{6d}; q, p)$ is the contribution of the gauge fields and in case of the $\Phi$-gluing also the fields $\Phi$. The fluxes are added/subtracted in case of $\Phi/S$-gluing respectively.

We also need to remind the reader about one more general statement about the superconformal index. If we give a vev to a bosonic operator $\mathcal{O}$, charged under some $U(1)_u$ symmetry with charge $-1$ (without loss of generality), which contributes to the index with weight $U^* = q^{j_2^{\mathcal{O}} - j_1^{\mathcal{O}} + \frac{1}{2}R^{\mathcal{O}}} p^{j_2^{\mathcal{O}} + j_1^{\mathcal{O}} + \frac{1}{2}R^{\mathcal{O}}} u^{-1}$, then the index of the theory in the IR is given by [2],

$$\mathcal{I}_{IR} \sim lim_{u \to U^*} \mathcal{I}_{UV}(u)\,, \tag{2.3}$$

where the $\sim$ denotes that to achieve equality we need to divide by some overall factors related to Goldstone degrees of freedom which will not be important for us.

Finally we perform the following computation. We take a general Riemann surface $\mathcal{C}_{g,s}[u]$ ($g$ is the genus and $s$ is the number of punctures) with at least one maximal puncture parametrized by fugacity $u$. Next we compute the index $\mathcal{I}\Big[\mathcal{C}_{g,s}[u] \oplus \mathcal{T}^{\mathcal{A}}_{z,u,\hat{a}}, \mathcal{F} + \mathcal{A}\Big]$ of the

theory corresponding to this surface with $\mathcal{T}_{z,u,\hat{a}}^{\mathcal{A}}$ glued to it along puncture $u$. Then we study what happens once we close the minimal puncture $\hat{a}$ with the preferred vev $\hat{M}^-$ defined above. The weight in the index of the operator $\hat{M}^-$ we give the vev to is $U^*$ and this vev breaks the minimal puncture $U(1)_{\hat{a}}$ symmetry.[9] By our definitions,

$$lim_{\hat{a} \to U^*} \mathcal{I}\left[\mathcal{C}_{g,s}[u] \oplus \mathcal{T}_{z,u,\hat{a}}^{\mathcal{A}}, \mathcal{F} + \mathcal{A}\right] \sim \mathcal{I}\left[\mathcal{C}_{g,s}[z], \mathcal{F}\right], \tag{2.4}$$

as the geometry after adding the trinion and removing the minimal puncture without changing the flux is the same as the one we started with and we assume that the theories are determined by the geometric data. More mathematically, this implies that the operation of adding $\mathcal{T}_{z,u,\hat{a}}^{\mathcal{A}}$ and then computing residue acts as an identity operator on the index of the theory on $\mathcal{C}_{g,s}[z]$. However, if one gives a vev to certain holomorphic derivatives of $\hat{M}^-$, $\partial_1^r \partial_2^m \hat{M}^-$ , such that the weight in the index is $p^r q^m U^*$, one typically obtains [2],

$$lim_{\hat{a} \to p^r q^m U^*} \mathcal{I}\left[\mathcal{C}_{g,s}[u] \oplus \mathcal{T}_{z,u,\hat{a}}^{\mathcal{A}}, \mathcal{F} + \mathcal{A}\right] \sim \mathcal{O}_z^{(\hat{a},\mathcal{A};r,m)} \cdot \mathcal{I}\left[\mathcal{C}_{g,s}[z], \mathcal{F}\right]. \tag{2.5}$$

Here $\mathcal{O}_z^{(\hat{a},\mathcal{A};r,m)}$ is an analytic difference operator acting on the $G_{5d}$ parameters $z$ in the index. See Figure 1 for an illustration. Physically this flow introduces surface defects into the index computation. The type of defect is defined by the type of the minimal puncture $\hat{a}$ and the flux $\mathcal{A}$ as well as the choices of the number of derivatives $r$ and $m$. The fact that the residue computation gives a difference operator is a non-trivial statement which does not have a direct derivation using this logic. However, the same result was obtained, at least in particular examples, by directly computing the index of a theory in presence of a surface defect [59].

Since the index is invariant under marginal deformations and dualities these operators satisfy various remarkable properties. For example, one can close two minimal punctures in different ways. The different orders to do so correspond to performing the computations in different duality frames. The index thus should be independent of this order. This, under the assumptions that the indices are rather a generic set of functions, leads to the expectation that all the operators obtained using such arguments should commute,

$$\left[\mathcal{O}_z^{(\hat{a},\mathcal{A};r,m)}, \mathcal{O}_z^{(\hat{b},\mathcal{B};r',m')}\right] = 0. \tag{2.6}$$

as shown on Figure 2. For example, studying the procedure detailed here starting with $A_{N-1}$ type $(2,0)$ theories the system of commuting operators of the Ruijsenaars-Schneider model can be obtained [2, 17]. This model has $2(N-1)$ independent commuting operators and these can be related to the choices $(r = 1, \ldots, N-1, m = 0)$ and $(r = 0, m = 1, \ldots, N-1)$ with the more general operators expressible in terms of these. We stress that the commutativity is a consequence of the dualities. Thus once the operators are computed the fact that they commute can be viewed as a non trivial check of the conjectured dualities.

---

[9]Without loss of generality we assume that the charge of the operator under $U(1)_{\hat{a}}$ is $-1$.

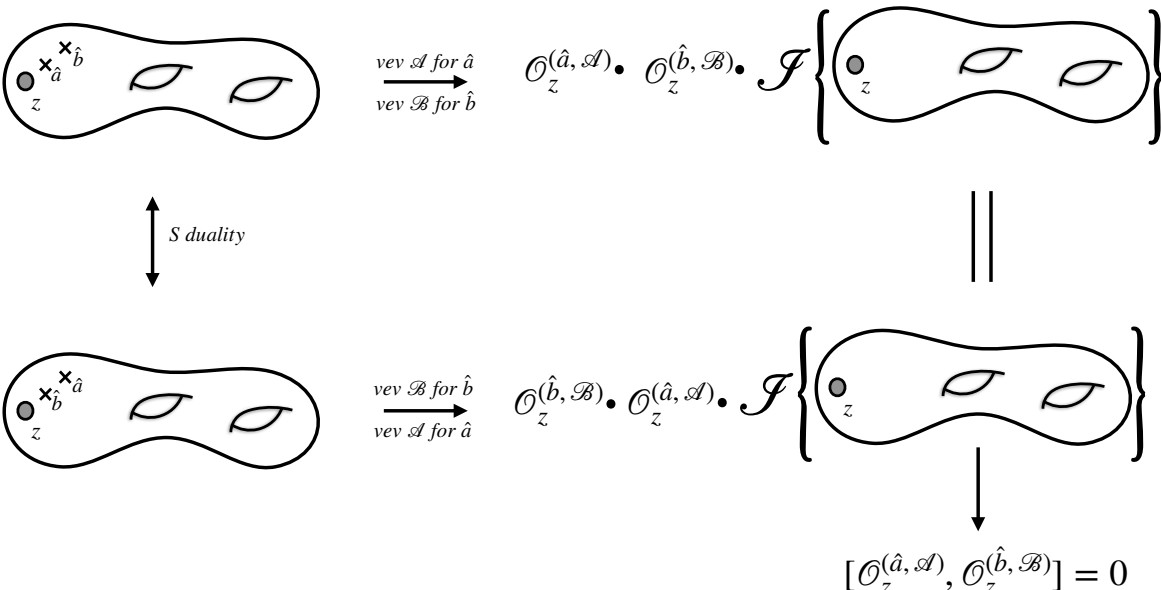

**Figure 2**. We start from a theory with a maximal puncture (labeled by $z$) and two minimal punctures (labeled by $\hat{a}$ and $\hat{b}$). We close both minimal punctures by giving a space-time dependent vacuum expectation value to operators charged under the two puncture symmetries, resulting in a theory with surface defects. The choice of operators is schematically encoded in $\mathcal{A}$ and $\mathcal{B}$. We can perform the computation of the index after giving the vevs in any duality frame and the result should not change. Here we present two frames which will result in the index being computed by acting on the index without the defect with two difference operators in different order. As the result does not depend on the frame the operators have to commute when acting on the index. Assuming that the index is a generic enough function this implies that the operators commute.

Another consequence of the dualities is that the indices themselves are *Kernel* functions for the difference operators, see Figure 3. Since the difference operators correspond to residues of the index in fugacity $\hat{a}$, and the index is invariant under the dualities, it does not matter on which maximal puncture fugacity the difference operator acts,

$$\mathcal{O}_z^{(\hat{a},\mathcal{A};r,m)} \cdot \mathcal{I}\Big[\mathcal{C}_{g,s}[z,\,u],\mathcal{F}\Big] = \mathcal{O}_u^{(\hat{a},\mathcal{A};r,m)} \cdot \mathcal{I}\Big[\mathcal{C}_{g,s}[z,\,u],\mathcal{F}\Big]. \tag{2.7}$$

Note that the operators $\mathcal{O}_z^{(\hat{a},\mathcal{A};r,m)}$ and $\mathcal{O}_u^{(\hat{a},\mathcal{A};r,m)}$ in this equality might not be exactly the same as they depend on the type of maximal puncture, $u$ or $z$, that they act on. If the types are the same the operators are the same, but otherwise they in principle can be different. We will discuss examples of this in what follows.

The discussion here can be generalized to other partition functions. The generalization is specifically straightforward when one can obtain the partition function of a theory after gauging a symmetry from the partition function of the theory before gauging that symme-

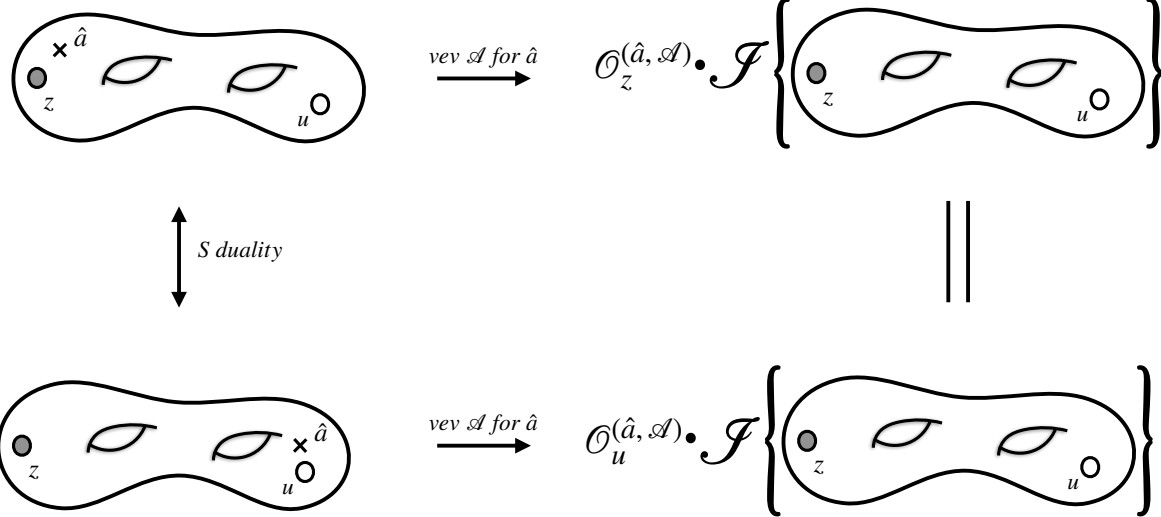

**Figure 3**. Here we give vev to an operator labeled by $\mathcal{A}$ charged under minimal puncture symmetry $\hat{a}$. We consider two duality frames. In each frame the minimal puncture is close to a different maximal puncture, labeled by $z$ and $u$. The fact that the duality frame does not matter for the computation, as the index is independent of the value of the continuous couplings, the two computations give the same result. This implies that the index of the theory without defect generated by the vev, regarded as the function of maximal puncture parameters $z$ and $u$, is a Kernel function for the difference operators introducing the defects.

try directly.[10] An example of that is the lens index [64–66]. In the case of lens index for the compactifications of the $(2,0)$ theory the computation leads to a rich structure [15, 16] involving Cherednik operators which in certain limits generalizes Macdonald polynomials to non symmetric functions.

## 3   Three roads to the van Diejen model

In this section we derive the relation between the rank one E-string theory and the $BC_1$ van Diejen model. This relation was already obtained in [25] using the three punctured sphere $\mathcal{T}^{\mathcal{A}}_{z,u,\hat{a}}$ obtained in [24]. Here we will present three different derivations each of which will then have a different extension to the minimal $(D, D)$ conformal matter theory and the associated integrable models being $A_N$, $C_N$ and $(A_1)^N$ generalizations of the van Diejen model. The rank one E-string theory has $G_{6d} = E_8$ and thus the integrable models in addition to $q$ and $p$ depend on an octet of variables parametrizing the Cartan sub-group of $E_8$. The 5d effective description is an $SU(2)$ gauge theory with an octet of hypermultiplets and thus $G_{5d} = SU(2)$

_______________

[10]In some partition functions, such as $\mathbb{S}^2 \times \mathbb{T}^2$ or elliptic genus in two dimensions, the connection between gauged and not gauged symmetries using matrix model techniques is more obscure and involves more sophisticated methods of computations (JK residues) [60, 61]. See however [62, 63] for a way to define the gauging in a guise possibly better suited for generalizations of our discussion.

with the moment maps forming an octet of fundamentals under $G_{5d}$. The minimal and the maximal punctures are the same for the rank one E-string. For some relevant details about the E-string theory see for example [24].

### 3.1 $BC_1$ van Diejen model

Before we derive operators introducing defects into the index computations of the rank one E-string theory in the following subsections, let us start by defining the basic $BC_1$ van Diejen operator. This operator in various guises will make an appearance throughout the paper. The $BC_N$ van Diejen operators were first defined in [26] and the corresponding integrable structure was discussed in [67]. These models can be viewed as a certain generalization of the elliptic relativistic Calogero-Moser systems (Ruijsenaars-Schneider models, see Appendix D for some details) and Koornwinder operators [68].[11] We will define the $BC_1$ operator here using the notations of [70].

The $BC_1$ van Diejen operator $A_D(h; x)$ depending on an octet of complex parameters $h = \{h_i\}_{i=1}^8$ and acting on a function $f(x)$ (such that $f(x) = f(x^{-1})$) is defined as follows,

$$A_D(h; x)\, f(x) \equiv V(h; x)\, f(qx) + V\big(h; x^{-1}\big)\, f\big(q^{-1}x\big) + V_b(h; x)\, f(x), \tag{3.1}$$

where

$$V(h; x) \equiv \frac{\prod\limits_{n=1}^{8} \theta_p\big((pq)^{\frac{1}{2}} h_n x\big)}{\theta_p(x^2)\theta_p\big(qx^2\big)}, \qquad V_b(h; x) \equiv \frac{\sum\limits_{k=0}^{3} p_k(h)[\mathcal{E}_k(\xi; x) - \mathcal{E}_k(\xi; \omega_k)]}{2\theta_p(\xi)\theta_p\big(q^{-1}\xi\big)}, \tag{3.2}$$

and

$$\omega_0 = 1, \qquad \omega_1 = -1, \qquad \omega_2 = p^{\frac{1}{2}}, \qquad \omega_3 = -p^{-\frac{1}{2}}, \quad \theta_p(x) = \prod_{i=0}^{\infty}(1 - p^i x)(1 - p^{i+1}x^{-1}). \tag{3.3}$$

The functions $p_k(h)$ are

$$p_0(h) \equiv \prod_n \theta_p(p^{\frac{1}{2}} h_n), \qquad p_1(h) \equiv \prod_n \theta_p\big(-p^{\frac{1}{2}} h_n\big),$$

$$p_2(h) \equiv p \prod_n h_n^{-\frac{1}{2}} \theta_p(h_n), \qquad p_3(h) \equiv p \prod_n h_n^{\frac{1}{2}} \theta_p\big(-h_n^{-1}\big), \tag{3.4}$$

and $\mathcal{E}_k$ is

$$\mathcal{E}_k(\xi; z) \equiv \frac{\theta_p\big(q^{-\frac{1}{2}} \xi \omega_k^{-1} x\big)\theta_p\big(q^{-\frac{1}{2}} \xi \omega_k x^{-1}\big)}{\theta_p\big(q^{-\frac{1}{2}} \omega_k^{-1} x\big)\theta_p\big(q^{-\frac{1}{2}} \omega_k x^{-1}\big)}. \tag{3.5}$$

---

[11]See [69] for appearance of the Koornwinder polynomials in the context of class $\mathcal{S}$ compactifications with outer-automorphism twists.

Constant term $V_b(h; x)$ of van Diejen model has following poles in the fundamental domain:

$$x = \pm q^{\pm \frac{1}{2}}, \qquad x = \pm q^{\pm \frac{1}{2}} p^{\frac{1}{2}}. \tag{3.6}$$

Residues at these poles are given by:

$$\text{Res}_{x=sq^{1/2}} V_b(h; x) = -s \frac{\prod_{n=1}^{8} \theta_p \left( sp^{\frac{1}{2}} h_n \right)}{2q^{-\frac{1}{2}} \theta_p \left( q^{-1} \right) (p; p)_\infty^2},$$

$$\text{Res}_{x=sq^{-1/2}} V_b(h; x) = s \frac{\prod_{n=1}^{8} \theta_p \left( sp^{\frac{1}{2}} h_n \right)}{2q^{\frac{1}{2}} \theta_p \left( q^{-1} \right) (p; p)_\infty^2},$$

$$\text{Res}_{x=sq^{1/2}p^{1/2}} V_b(h; x) = -s \frac{\prod_{n=1}^{8} h_n^{-\frac{1}{2}} \theta_p \left( sh_n \right)}{2q^{-\frac{1}{2}} p^{-\frac{3}{2}} \theta_p \left( q^{-1} \right) (p; p)_\infty^2},$$

$$\text{Res}_{x=sq^{-1/2}p^{1/2}} V_b(h; x) = s \frac{\prod_{n=1}^{8} h_n^{-\frac{1}{2}} \theta_p \left( sh_n \right)}{2q^{\frac{1}{2}} p^{-\frac{3}{2}} \theta_p \left( q^{-1} \right) (p; p)_\infty^2}, \tag{3.7}$$

where $s = \pm 1$.

In what follows we will see how the operator (3.1) will appear in the context of studying surface defects in the index computations of the rank one E-string theory.

### 3.2 $A_1$ van Diejen model

We start our discussion of the defects in E-string theory compactifications with the definition of a particular trinion theory, $\mathcal{T}_{x,y,z}^3$, derived in [34]. This is the $SU(3)$ SQCD with $N_f = 6$. Corresponding quiver is specified on the Figure 4. The supersymmetric index of $\mathcal{T}_{x,y,z}^3$ is given by,

$$K_3^{A_1}(x, y, z) = \frac{\kappa^2}{6} \oint \frac{dt_1}{2\pi i t_1} \frac{dt_2}{2\pi i t_2} \frac{1}{\prod_{i \neq j}^3 \Gamma_e(t_i/t_j)} \times \tag{3.8}$$

$$\prod_{i=1}^{3} \Gamma_e((qp)^{\frac{1}{6}} u^6 t_i x^{\pm 1}) \Gamma_e((qp)^{\frac{1}{6}} v^6 t_i y^{\pm 1}) \Gamma_e((qp)^{\frac{1}{6}} w^6 t_i z^{\pm 1}) \prod_{j=1}^{6} \Gamma_e((qp)^{\frac{1}{3}} (uvw)^{-2} a_j t_i^{-1}).$$

The definitions of the various special functions can be found in Appendix A. The parameters $t_i$ parametrize the gauged $SU(3)$ ($\prod_{i=1}^3 t_i = 1$), and the parameters $u, v, w, a_1, \ldots, a_5$

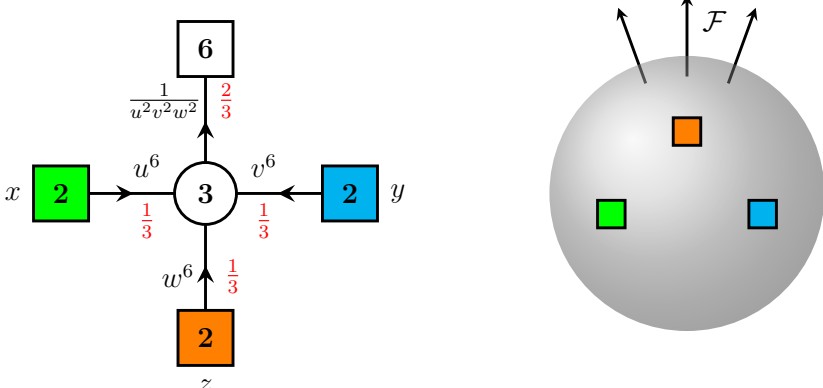

**Figure 4**. The 4d theory obtained by compactifying the rank one E-string on a three punctured sphere with a particular choice of flux (breaking the $E_8$ global symmetry down to $E_7 \times U(1)$) and definition of punctures. We will refer to this theory as the $A_1$ trinion theory, and it is given by $SU(3)$ SQCD with $N_f = 6$.

parametrize a choice of Cartans of $E_8$ such that,

$$\prod_{i=1}^{6} a_i = 1, \qquad E_8 \to E_7 \times SU(2)_{u^6 v^6 w^6} \to SU(6)_{a_i} \times SU(3)_{\frac{u^8}{v^4 w^4}, \frac{v^8}{u^4 w^4}} \times U(1)_{u^6 v^6 w^6} . \quad (3.9)$$

The three puncture $SU(2)$ symmetries are parametrized by $x$, $y$, and $z$. The octets of the moment map operators have the following charges,

$$
\begin{aligned}
M_u &= \mathbf{2}_x \ \otimes \ \left( \mathbf{6}_{u^4/v^2 w^2} \oplus \mathbf{1}_{u^6 v^{12}} \oplus \mathbf{1}_{u^6 w^{12}} \right) , \\
M_v &= \mathbf{2}_y \ \otimes \ \left( \mathbf{6}_{v^4/u^2 w^2} \oplus \mathbf{1}_{v^6 u^{12}} \oplus \mathbf{1}_{v^6 w^{12}} \right) , \\
M_w &= \mathbf{2}_z \ \otimes \ \left( \mathbf{6}_{w^4/u^2 v^2} \oplus \mathbf{1}_{w^6 u^{12}} \oplus \mathbf{1}_{w^6 v^{12}} \right) ,
\end{aligned}
\qquad (3.10)
$$

where these are built by a sextet of mesons and two baryons. Everywhere in this paper the charges of operators (fields) under various symmetries are encoded in the powers of fugacities for corresponding symmetries. For example the operator with weight $\mathbf{6}_{u^4/v^2 w^2}$ is a sextet of $SU(6)$ and has charges $+4$ under $U(1)_u$ and $-2$ under $U(1)_v$ and $U(1)_w$. The mesons are built from a fundamental of $SU(3)$ transforming under the puncture symmetry and the sextet of antifundamentals. The two baryons are built from one copy of the fundamental of $SU(3)$ transforming under the given puncture symmetry and two fundamentals of $SU(3)$ transforming under a different puncture symmetry. Note that the three punctures are of different types (colors) as the pattern of charges of the moment maps is different for each puncture.

The derivation of the AΔO here will lead to a version of the $BC_1$ van Diejen operator we will refer to as the $A_1$ operator. The reason is that the three punctured sphere used here has a direct generalization to the $(D_{N+3}, D_{N+3})$ conformal matter theories with $SU(N+1)$ punctures: here we discuss the case of $N = 1$. Thus the operators will act on the $SU(N+1)$ punctures, and we will refer to them as $A_N$ type of operators. The generalization will be

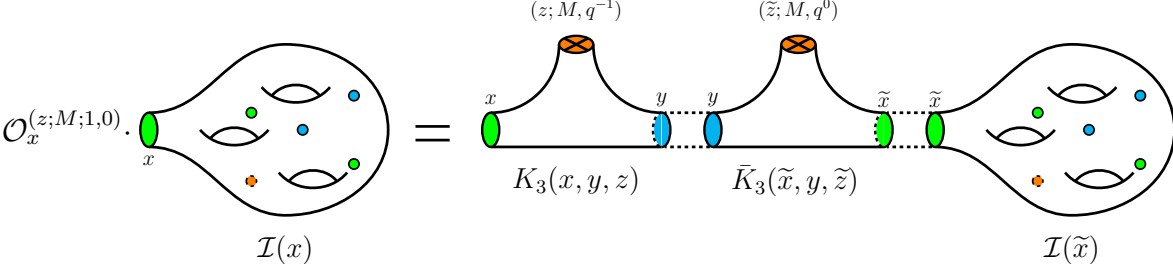

**Figure 5**. Construction of the AΔO. We start here with some theory with the superconformal index $\mathcal{I}(x)$. Then we S glue it to two trinions and close minimal punctures. In the figure above we close $\mathrm{SU}(2)_z$ punctures and denoted closure operation as the cross on top of the puncture. The notation $(z; M, q^{-1})$ means that we close $z$-puncture by giving vev to the moment map $M$. The last label, $q^{-1}$, denotes which kind of vev it is. In particular $q^0$ corresponds to constant vev, while non-zero negative powers, like $q^{-1}$, correspond to the space dependent vev. As the result of gluing operation we obtain theory with the very same superconformal index $\mathcal{I}(x)$ but this time with some AΔO $\mathcal{O}_x^{(z;M;1,0)}$ acting on it. Particular form of operator depends on the trinion theories we use in the construction and the way we close minimal punctures.

discussed in Section 4. Many of the explicit technical details of the discussion here are presented in Appendix B for the general $N$ case.

We will apply the algorithm of Section 2 to derive an AΔO introducing defects in the index using $\mathcal{T}_{x,y,z}^{\mathcal{A}}$. We start with an arbitrary theory $\mathcal{T}^0$ with an SU(2) global symmetry and the corresponding index $\mathcal{I}^0$. Geometrically this corresponds to the compactification on some Riemann surface $\mathcal{C}$ with at least one maximal (which is the same here as minimal) puncture. In order to obtain AΔO we want to glue this surface to one or more trinions and close the punctures of the latter ones in such a way that the total flux through $\mathcal{C}$ is not shifted in the end. The easiest way to do it in our case is to perform S gluing of two trinions $\mathcal{T}_{x,y,z}^{\mathcal{A}}$ and $\mathcal{T}_{x,y,z}^{\bar{\mathcal{A}}}$ with conjugated fluxes to the original surface $\mathcal{C}$ along the SU(2) puncture as shown on the Figure 5. At the level of the index according to (2.2) this operation is expressed as follows:

$$\mathcal{I}\left[\mathcal{C}_{\tilde{x}} \oplus \mathcal{T}_{\tilde{x},y,\tilde{z}}^{\bar{\mathcal{A}}} \oplus \mathcal{T}_{x,y,z}^{\mathcal{A}}\right] = \kappa^2 \oint \frac{dy}{4\pi i y}\frac{d\tilde{x}}{4\pi i \tilde{x}}\frac{1}{\Gamma_e\left(y^{\pm 2}\right)\Gamma_e\left(\tilde{x}^{\pm 2}\right)} \times$$
$$K_3^{A_1}(x,y,z)\bar{K}_3^{A_1}(\tilde{x},y,\tilde{z})\mathcal{I}\left[\mathcal{C}_{\tilde{x}}\right], \qquad (3.11)$$

Here as a particular example we started with the theory having $\mathrm{SU}(2)_x$ minimal puncture and S glued trinions along $\mathrm{SU}(2)_y$ punctures. In a completely identical way one can choose gluings along other combinations of punctures since in rank one E string case all the punctures are minimal. In case of $A_N$ generalization of this model described in Section 4 situation will be different and only one of three punctures will be of minimal type. The index of the conjugated theory $\mathcal{T}_{\tilde{x},y,\tilde{z}}^{\bar{\mathcal{A}}}$ can be obtained from the index (3.8) by simply inverting all the flavor fugacities: $(a_i, u, v, w) \to (a_i^{-1}, u^{-1}, v^{-1}, w^{-1})$.

Now we should close $\mathrm{SU}(2)_z$ and $\mathrm{SU}(2)_{\tilde{z}}$ punctures. There are two ways to do it that lead to identical results. In the first approach we start with gluing two trinions forming four-punctured sphere and then close two conjugated punctures giving vev to two operators

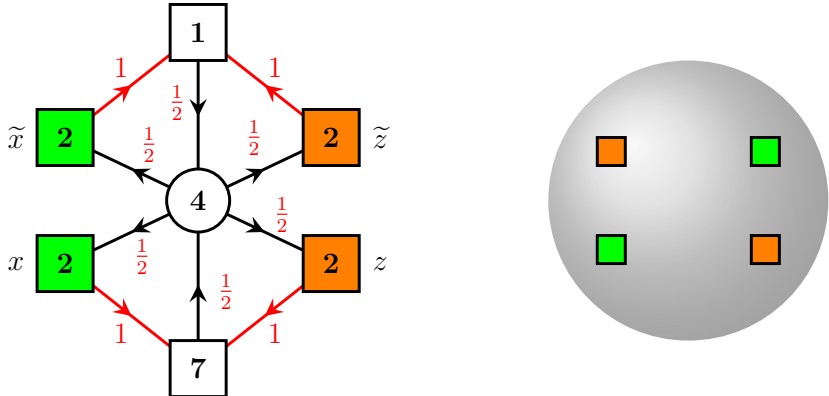

**Figure 6**. The $A_1$ four-punctured sphere theory with zero flux: $SU(4)$ SQCD with $N_f = 8$. Fugacities of all of the fields can be identified from the index specified in (B.4) upon putting $N = 1$. Red lines correspond to the flips of mesonic operators of this theory. This theory has superpotentials turned on corresponding to the triangles in the quiver and baryonic superpotentials preserving the puncture symmetries (and including quarks charged under orange and green $SU(2)$ symmetries). Here, and everywhere in the paper, we do not discuss the subtle issues of the relevance of the superpotentials as this does not play a role in the derivation of the A$\Delta$Os.

corresponding to them. This will result in a certain tube theory which we can in turn S glue to an arbitrary theory. Another way to do this calculation is first to close corresponding punctures of the two conjugated trinions obtaining expressions for two tube theories. After that we can glue them together and to an arbitrary theory. These two approaches just correspond to two different orders of performing operations of closing puncture and gluing specified in (3.11). The final result of course does not depend on this order, which we have checked. For presentation purposes here in $A_1$ case we choose the first approach consisting of deriving the four-punctured sphere theory first. Further in other cases we will also demonstrate details of the other approach.

Gluing two conjugated trinion theories $\mathcal{T}^{\mathcal{A}}_{x,y,z}$ and $\bar{\mathcal{T}}^{\mathcal{A}}_{\tilde{x},y,\tilde{z}}$ along the $SU(2)_y$ puncture and performing a chain of Seiberg dualities we obtain relatively simple $SU(4)$ SQCD with 8 flavors theory. The quiver of this theory is shown on the Figure 6. Derivation of this theory is summarized in the Appednix B.1 for the $A_N$ case and its index $K_4^{A_1}(x, \tilde{x}, z, \tilde{z})$ is specified in (B.4). Expressions for the $A_1$ case can be directly obtained from this Appendix by putting $N = 1$. Gluings along $SU(2)_x$ and $SU(2)_z$ punctures can be similarly discussed using appropriate permutation of the $(u, v, w)$ fugacities.

As the second step we close $SU(2)_z$ and $SU(2)_{\tilde{z}}$ punctures giving vev to one of the moment maps. At this point for each puncture we have a choice of 8 moment map operators specified in the last line of (3.10). On top of this we can "flip" components of the moment map operators (adding a chiral field linearly coupled to the operator with a superpotential) before closing the puncture.[12] Notice that we aim to have a zero-flux tube in the end. This requires

---

[12]Such flippings amount to changing the definition of the puncture. In particular in $5d$ this will amount to the question which matter fields acquire Dirichlet and which Neumann boundary conditions. See *i.e.* [24, 71].

punctures $\mathrm{SU}(2)_z$ and $\mathrm{SU}(2)_{\tilde z}$ to be closed consistently, *i.e.* both should be either flipped or not and both should be closed using the same moment map. This leads to 16 possible AΔOs in total.

Let's start with an example before summarizing general result. First we consider closing $\mathrm{SU}(2)_z$ and $\mathrm{SU}(2)_{\tilde z}$ punctures using the baryon $\mathbf{1}_{w^6 u^{12}}$. At the level of the index this amounts to computing the residue of the pole located at

$$z = (pq)^{-\frac{1}{2}} u^{-12} w^{-6} q^{-K} p^{-M}, \quad \tilde z = (pq)^{-\frac{1}{2}} u^{12} w^6 q^{-\tilde K} p^{-\tilde M} \tag{3.12}$$

where $K$, $M$, $\tilde K$, $\tilde M$ are positive integers. Here for the sake of simplicity we will concentrate on the case $\tilde K = 0$ and $K = 1$.[13] Physically this corresponds to giving vev to the derivative of $\mathbf{1}_{w^6 u^{12}}$ baryonic moment map of the $\mathrm{SU}(2)_z$ puncture introducing surface defect into the theory. The puncture $\mathrm{SU}(2)_{\tilde z}$ in turn is closed using space-independent vev of the conjugated baryonic moment map.

Corresponding calculation of the residue is summarized in the Appendix B.2 for the more general case of $A_N$ trinions. Derivation for the $A_1$ case can be simply reproduced from it by putting $N = 1$ in all of the equations. This calculation results in the following AΔO,

$$\mathcal{O}_x^{(z;u^{12}w^6;1,0)} \cdot \mathcal{I}(x) = \frac{\prod_{i=1}^{8} \theta_p((qp)^{\frac{1}{2}} h_i^{-1} x)}{\theta_p(qx^2)\theta_p(x^2)} \mathcal{I}(qx) + \frac{\prod_{i=1}^{8} \theta_p((qp)^{\frac{1}{2}} h_i^{-1} x^{-1})}{\theta_p(qx^{-2})\theta_p(x^{-2})} \mathcal{I}(qx^{-1})$$
$$+ W(x; h_i)^{(z;u^{12}w^6;1,0)} \mathcal{I}(x). \tag{3.13}$$

Here we have introduced the following notations. First of all we encoded all required information in the indices of the operator. Subscript $x$ means that we act on the $\mathrm{SU}(2)_x$ puncture of $\mathcal{N} = 1$ theory. First argument $z$ in the superscript stands for the $\mathrm{SU}(2)_z$-type punctures that we close. Second argument of the superscript is the charge of the moment map we give space-dependent vev to in order to close the puncture. In our case it is the charge $u^{12} w^6$ of the corresponding baryon. Finally the last pair of integers $(1, 0)$ stands for the choice of $K$ and $M$ integers in the pole (3.12). On the r.h.s. $h_i$ is the octet of charges of the moment maps of the puncture we act on. In this case it is $M_u$ with charges $h_i = \left( \mathbf{6}_{u^4/v^2 w^2} \oplus \mathbf{1}_{u^6 v^{12}} \oplus \mathbf{1}_{u^6 w^{12}} \right)$.

---

[13]For the E-string, as the puncture symmetry is $A_1$, we expect the higher numbers of derivatives to give rise to operators which are expressible in terms of the one we will derive below. It is analagous to $A_1$ class $\mathcal{S}$ with the basic pole giving the RS operator and the higher poles giving polynomials of it. For general $A_{N-1}$ class $\mathcal{S}$ the first $N - 1$ poles give rise to a commuting set of independent operators while the higher poles are expressible in terms of the basic ones. For details see [2, 17].

The function $W(x; h_i)^{(z; u^{12} w^6; 1, 0)}$ is given by

$$W(x; h_i)^{(z; u^{12} w^6; 1, 0)} \equiv \left[ \frac{\theta_p\left((pq)^{\frac{1}{2}} u^6 w^{12} x\right) \theta_p\left((pq)^{\frac{1}{2}} v^{12} u^6 x\right)}{\theta_p\left((pq)^{\frac{1}{2}} u^{18} qx\right) \theta_p\left(q^{-1} x^{-2}\right) \theta_p\left(x^2\right)} \theta_p\left((pq)^{\frac{1}{2}} u^{18} x^{-1}\right) \times \right.$$

$$\left. \prod_{j=1}^6 \theta_p\left((pq)^{\frac{1}{2}} u^4 v^{-2} w^{-2} a_j x\right) + (x \to x^{-1}) \right] + \prod_{j=1}^6 \theta_p\left(u^{-14} v^{-2} w^{-2} q^{-1} a_j\right) \times$$

$$\frac{\theta_p\left(q^{-1} v^{12} u^{-12}\right) \theta_p\left((pq)^{\frac{1}{2}} u^6 w^{12} x^{\pm 1}\right)}{\theta_p\left(pq^2 w^{12} u^{24}\right) \theta_p\left((pq)^{-\frac{1}{2}} u^{-18} q^{-1} x^{\pm 1}\right)} . \quad (3.14)$$

Notice that while the shift part of the operator (3.13) depends only on the charges of the moment maps of the puncture we act on, the constant part specified above depends also on the charges of the moment map we use to close the puncture $SU(2)_z$.

The constant part presented in (3.14) is elliptic function in $x$ with periods 1 and $p$ just as is the constant part $V_b(x)$ of the $BC_1$ van Diejen operator specified in (3.1) and (3.2). Also the poles of both functions in the fundamental domain are located at

$$x = s q^{\pm \frac{1}{2}}, \quad x = s q^{\pm \frac{1}{2}} p^{\frac{1}{2}}, \quad s = \pm 1. \quad (3.15)$$

However matching residues of $W(x; h_i)^{(z; u^{12} w^6; 1, 0)}$ and $V_b(x)$ functions requires flip of one of the charges which is implemented by the following conjugation of the operator

$$\tilde{\mathcal{O}}_x^{(z; u^{12} w^6; 1, 0)} \equiv \Gamma_e\left((pq)^{\frac{1}{2}} u^6 w^{12} x^{\pm 1}\right)^{-1} \mathcal{O}_x^{(z; u^{12} w^6; 1, 0)} \Gamma_e\left((pq)^{\frac{1}{2}} u^6 w^{12} x^{\pm 1}\right). \quad (3.16)$$

This conjugation affects only the shift part of the operator (3.13) leading to

$$\tilde{\mathcal{O}}_x^{(z; u^{12} w^6; 1, 0)} \cdot \mathcal{I}(x) = \frac{\prod_{i=1}^8 \theta_p((qp)^{\frac{1}{2}} \tilde{h}_i^{-1} x)}{\theta_p(qx^2) \theta_p(x^2)} \mathcal{I}(qx) + \frac{\prod_{i=1}^8 \theta_p((qp)^{\frac{1}{2}} \tilde{h}_i^{-1} x^{-1})}{\theta_p(qx^{-2}) \theta_p(x^{-2})} \mathcal{I}(qx^{-1})$$

$$+ W(x; \tilde{h}_i)^{(z; u^{12} w^6; 1, 0)} \mathcal{I}(x), \quad (3.17)$$

where $\tilde{h}_i$ is the octet of the moment map charges with one of the baryons flipped $\tilde{h}_i = \left(\mathbf{6}_{u^4/v^2 w^2} \oplus \mathbf{1}_{u^6 v^{12}} \oplus \mathbf{1}_{u^{-6} w^{-12}}\right)$. [14]

Now in order to compare this operator to the van Diejen operator (3.1) we start with the shift part. From it we clearly see that octet of the van Diejen model parameters $h_n$ is

---

[14]Let us make a comment here. Note that the flip on one hand just redefines the type of the puncture by changing the charges of one of the components of the moment map. On the other hand this is essential as doing so for odd number of components changes the global Witten anomaly of the $SU(2)$ puncture symmetry. Note that the three punctured sphere of Figure 4 has $SU(2)$ punctures with three fundamentals and thus have a Witten anomaly for the global symmetry. The flipping takes us to a definition without such an anomaly. In particular when $\Phi$ gluing punctures one should always be careful that the Witten anomaly is zero. For example the trinion defined in [24] (used in [25] to derive the van Diejen model) has no Witten anomaly for punctures. The same is true for the trnion derived in [38] and used in the next subsection. Thus gluing these trinions to the trinion of Figure 4 one cannot use S or $\Phi$ gluing but a combination of these with odd number of gluings of each type.

equal to the octet of the inverse parameters $\tilde{h}_i^{-1}$ specified above, i.e. $h_n^{vD} = \tilde{h}_n^{-1}$. Using this simple identification we can check that all the residues of the constant part $W(x; h_i)^{(z; u^{12} w^6; 1, 0)}$ specified in (3.14) are given by (3.7). Hence since the constant part is elliptic in $x$ and poles with residues coincide with those of the van Diejen model we conclude that up to a constant independent of $x$ the operator $\tilde{\mathcal{O}}_x^{(z; u^{12} w^6; 1, 0)}$ is just the $BC_1$ van Diejen model. Above we have made the choice of which single component of the moment map to flip. This choice is immaterial and in fact we can flip any odd number of components and find a map between parameters to match with the $BC_1$ van Diejen model.

Above we gave an example of the result of a particular computation with a particular puncture we act on and a particular moment map we give vev to. Similar computations can be performed for all other possible combinations of punctures and moment maps.

Next, after we have discussed several subtleties using particular example, let us describe the general operator as a function of the puncture we act on and the moment map we give vev to. Calculations for other combinations can be done in a similar way. Since most of them are almost identical to the calculation of the operator above, details of which we present in the Appendix B.2, we leave these derivations to the interested reader. Without loss of generality from now on we will only consider the situation when we close $\mathrm{SU}(2)_z$ puncture. In general for each of the two remaining puncture types we have an octet $h_i^{(a)}$ of $\mathrm{U}(1)$ charges where index $a = (u, v)$ stands for the puncture type. We have to flip one of the charges in each octet so that the charges of the moment maps become $\tilde{h}_i^{(a)}$ defined as follows

$$
\begin{aligned}
\widetilde{M_u} &= \mathbf{2}_x \;\otimes\; \left(\mathbf{6}_{u^4/v^2 w^2} \oplus \mathbf{1}_{u^6 v^{12}} \oplus \mathbf{1}_{u^{-6} w^{-12}}\right), \\
\widetilde{M_v} &= \mathbf{2}_y \;\otimes\; \left(\mathbf{6}_{v^4/u^2 w^2} \oplus \mathbf{1}_{v^6 u^{12}} \oplus \mathbf{1}_{v^{-6} w^{-12}}\right). \\
M_w &= \mathbf{2}_z \;\otimes\; \left(\mathbf{6}_{w^4/u^2 v^2} \oplus \mathbf{1}_{w^6 u^{12}} \oplus \mathbf{1}_{w^6 v^{12}}\right),
\end{aligned}
\tag{3.18}
$$

Notice that the charges of $\mathrm{SU}(2)_z$ puncture moment maps are the same as in (3.10). Now let's write down the operator acting on the puncture of type $a$ obtained by giving vev to one of the $M_w$ moment maps. As one can notice from (3.18) $M_w$ moment maps are related to the moment maps $\widetilde{M}_{u,v}$ as follows

$$
\tilde{h}_i^{(w)} = \tilde{h}_i^{(a)} \tilde{h}_a^{-\frac{1}{4}}, \quad \tilde{h}_a \equiv \prod_{i=1}^{8} \tilde{h}_i^{(a)}, \quad a = (u, v),
\tag{3.19}
$$

where we have to flip some of the $M_w$ depending on which type $a$ is the puncture we act on. Since moment maps $M_w$ are related to moment maps $\widetilde{M}_{u,v}$ as specified above further we will parametrize everything, including the moment map we give v.e.v. to, using charges $h_i^{(a)}$ of the moment maps of the puncture we act on.

Then we can write the general shift operator arising from this construction:

$$\tilde{\mathcal{O}}_a^{(z;\tilde{h}_i^{(a)}\tilde{h}_a^{-\frac{1}{4}};1,0)} \cdot \mathcal{I}(x_a) = \frac{\prod_{i=1}^8 \theta_p((qp)^{\frac{1}{2}}\left(\tilde{h}_i^{(a)}\right)^{-1} x_a)}{\theta_p(qx_a^2)\theta_p(x_a^2)}\mathcal{I}(qx_a) +$$

$$\frac{\prod_{i=1}^8 \theta_p((qp)^{\frac{1}{2}}\left(\tilde{h}_i^{(a)}\right)^{-1} x_a^{-1})}{\theta_p(qx_a^{-2})\theta_p(x_a^{-2})}\mathcal{I}(qx_a^{-1}) + W(x_a;\tilde{h}_i^{(a)})^{(z;\tilde{h}_i^{(a)}\tilde{h}_a^{-\frac{1}{4}};1,0)}\mathcal{I}(x_a)\,,$$

$$W(x_a;\tilde{h}_i^{(a)})^{(z;\tilde{h}_i^{(a)}\tilde{h}_a^{-\frac{1}{4}};1,0)} = \frac{\prod_{j\neq i}^8 \theta_p\left(q^{-1}\left(\tilde{h}_i^{(a)}\tilde{h}_j^{(a)}\right)^{-1}\tilde{h}_a^{\frac{1}{2}}\right)}{\theta_p\left(q^{-2}\left(\tilde{h}_i^{(a)}\right)^{-2}\tilde{h}_a^{1/2}\right)} \times$$

$$\frac{\theta_p\left((pq)^{\frac{1}{2}}\tilde{h}_i^{(a)}x_a^{\pm 1}\right)}{\theta_p\left((pq)^{-\frac{1}{2}}\left(\tilde{h}_i^{(a)}\right)^{-1}\tilde{h}_a^{1/2}q^{-1}x_a^{\pm 1}\right)} + \left[\frac{\prod_{j\neq i}^8 \theta_p\left((pq)^{\frac{1}{2}}\left(\tilde{h}_j^{(a)}\right)^{-1}x_a^{-1}\right)}{\theta_p\left((pq)^{\frac{1}{2}}\tilde{h}_i^{(a)}\tilde{h}_a^{-1/2}qx_a^{-1}\right)} \times$$

$$\frac{\theta_p\left((pq)^{\frac{1}{2}}\tilde{h}_i^{(a)}\tilde{h}_a^{-1/2}x_a\right)\theta_p\left((pq)^{\frac{1}{2}}\tilde{h}_i^{(a)}x_a^{-1}\right)}{\theta_p\left(q^{-1}x_a^{-2}\right)\theta_p\left(x_a^2\right)} + \left(x_a \to x_a^{-1}\right)\right]\,, \qquad (3.20)$$

where $x_u = x$, $x_v = y$ are SU(2) charges of the puncture.

The expression above gives an octet of possible operators originating from 8 moment maps we can give vev to. Another octet comes when we give space-dependent vev to the operators with charges $\left(\tilde{h}_i^{(a)}\right)^{-1}\tilde{h}_a^{\frac{1}{4}}$. The latter one can be obtained in two ways. First we can flip corresponding moment map before giving it a vev. This is achieved by introducing the contribution $\Gamma_e\left((pq)^{\frac{1}{2}}\left(\tilde{h}_i^{(a)}\right)^{-1}\tilde{h}_a^{\frac{1}{4}}z^{\pm 1}\right)\Gamma_e\left((pq)^{\frac{1}{2}}\tilde{h}_i^{(a)}\tilde{h}_a^{-\frac{1}{4}}z^{\pm 1}\right)$ of the flip multiplet into the index $K_{4,A1}\left(x,\tilde{x},z,\tilde{z}\right)$ of the four-punctured sphere theory. Second approach is to work with the original no-flip theory and give $z$, $\tilde{z}$ fugacities the following weights:

$$z = (pq)^{-\frac{1}{2}}\left(\tilde{h}_i^{(a)}\right)^{-1}\tilde{h}_a^{\frac{1}{4}}\,, \quad \tilde{z} = (pq)^{-\frac{1}{2}}\tilde{h}_i^{(a)}\tilde{h}_a^{-\frac{1}{4}}q^{-1}\,, \qquad (3.21)$$

which corresponds this time to giving space-dependent vev to the moment map of SU(2)$_{\tilde{z}}$ puncture introducing corresponding surface defect into the theory. Notice that this is opposite to our previous choice in (3.12). In Appendix B.3 we give a detailed derivation of one example of this kind. Performing this derivation for various moment maps we arrive to the general expression of the following form:

$$\tilde{\mathcal{O}}_a^{(z;(\tilde{h}_i^{(a)})^{-1}\tilde{h}_a^{\frac{1}{4}};1,0)} \cdot \mathcal{I}(x_a) = \frac{\prod_{i=1}^8 \theta_p((qp)^{\frac{1}{2}}\left(\tilde{h}_i^{(a)}\right)^{-1} x_a)}{\theta_p(qx_a^2)\theta_p(x_a^2)}\mathcal{I}(qx_a) +$$

$$\frac{\prod_{i=1}^8 \theta_p((qp)^{\frac{1}{2}}\left(\tilde{h}_i^{(a)}\right)^{-1} x_a^{-1})}{\theta_p(qx_a^{-2})\theta_p(x_a^{-2})}\mathcal{I}(qx_a^{-1}) + W(x_a;\tilde{h}_i^{(a)})^{(z;(\tilde{h}_i^{(a)})^{-1}\tilde{h}_a^{\frac{1}{4}};1,0)}\,\mathcal{I}(x_a)\,,$$

$$W(x_a;\tilde{h}_i^{(a)})^{(z;(\tilde{h}_i^{(a)})^{-1}\tilde{h}_a^{\frac{1}{4}};1,0)} = \frac{\prod\limits_{j\neq i}^8 \theta_p\left(q^{-1}\tilde{h}_i^{(a)}\tilde{h}_j^{(a)}\tilde{h}_a^{-\frac{1}{2}}\right)}{\theta_p\left(q^{-2}\left(\tilde{h}_i^{(a)}\right)^2 \tilde{h}_a^{-1/2}\right)} \times$$

$$\frac{\theta_p\left((pq)^{\frac{1}{2}}\left(\tilde{h}_i^{(a)}\right)^{-1} x_a^{\pm 1}\right)}{\theta_p\left((pq)^{-\frac{1}{2}}\tilde{h}_i^{(a)}\tilde{h}_a^{-1/2}q^{-1}x_a^{\pm 1}\right)} + \left[\frac{\prod\limits_{j\neq i}^8 \theta_p\left((pq)^{\frac{1}{2}}\tilde{h}_j^{(a)}x_a\right)}{\theta_p\left((pq)^{\frac{1}{2}}\left(\tilde{h}_i^{(a)}\right)^{-1}\tilde{h}_a^{1/2}qx_a\right)} \times \right.$$

$$\left. \frac{\theta_p\left((pq)^{\frac{1}{2}}\left(\tilde{h}_i^{(a)}\right)^{-1}\tilde{h}_a^{1/2}x_a^{-1}\right)\theta_p\left((pq)^{\frac{1}{2}}\left(\tilde{h}_i^{(a)}\right)^{-1} x_a\right)}{\theta_p\left(q^{-1}x_a^{-2}\right)\theta_p\left(x_a^2\right)} + \left(x_a \to x_a^{-1}\right)\right]\,, \quad (3.22)$$

Notice that the operator (3.13) is exactly of this kind and can be reproduced from the operator above if we put $a = u$ and $\tilde{h}_i^{(u)} = u^{-6}w^{-12}$.

In all of the operators (3.20) and (3.22) constant parts $W(x_a;\tilde{h}_i^{(a)})$ are elliptic functions of $x_a$ with periods 1 and $p$. Poles of all these functions are located at (3.15). Residues are given by the residues of van Diejen model specified in (3.7) upon the identification of parameters $h_i^{vD} = \left(\tilde{h}_i^{(a)}\right)^{-1}$. Hence all of the 16 operators specified above actually collapse to the one single operator equal to van Diejen operator up to a constant. We will later see that in the generalization of this discussion to $(D_{N+3}, D_{N+3})$ minimal conformal matter, we will obtain an $A_N$ generalization of the system of operators and these 16 different operators will generalize to $4N + 12$ operators. Also of course we have operators $\tilde{\mathcal{O}}_a^{(z;...;0,1)}$ which are simply obtained from the operators above by the exchange $(p \leftrightarrow q)$. In subsection 3.5 we will comment more on some of the properties of the operators derived here.

### 3.3  $A_1^1$ van Diejen model

We proceed with yet another trinion of the E-string compactification. The theory $\mathcal{T}_{v,z,\epsilon}^3$ we will consider in the present section was derived in [38] and corresponds to a compactification on a three punctured sphere with vanishing flux. The quiver of this theory is presented on the Figure 7. Just as in the previous section there are three types of punctures possessing SU(2)

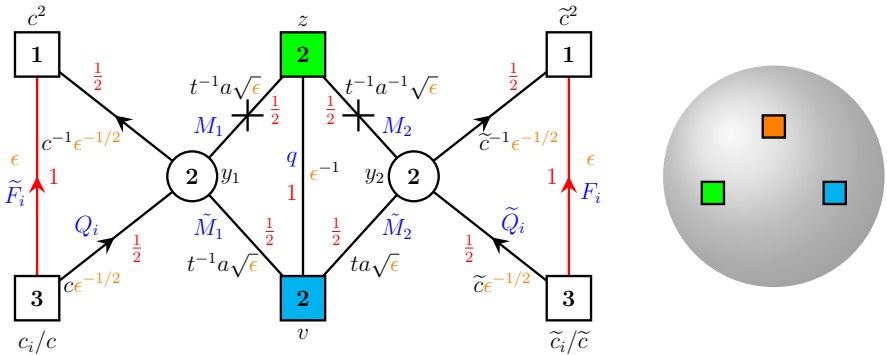

**Figure 7**. Quiver of $A_1^1$ trinion theory. This trinion is obtained by compactifying the E-string on a sphere with certain choice of three punctures and vanishing flux. Crosses on lines stand for gauge singlet chiral fields flipping a quadratic gauge invariant built from the field they cross out. All fields charged under $SU(2)$ gauge symmetries have $R$-charge $\frac{1}{2}$ while the gauge singlets have $R$-charge 1. Red lines as usually stand for the contributions of flip singlets $F_i$ and $\tilde{F}_i$. In our notations blue corresponds to the names of the fields, red to the fugacities of gauge and global symmetries and black to the U(1) charges of the fields. We also write $\epsilon$ in orange denoting the $SU(2)_\epsilon$ global symmetry of the corresponding type puncture that is not present explicitly and only arises in the IR. Types of two other punctures are denoted in blue and green. For the fugacities we also use notation $c = (c_1 c_2 c_3)^{1/3}$ and $\tilde{c} = (\tilde{c}_1 \tilde{c}_2 \tilde{c}_3)^{1/3}$. For convenience in the text we will instead use fugacities $c_4 = c^{-3}$ and $\tilde{c}_4 = \tilde{c}^{-3}$.

global symmetry. Two of the $SU(2)$ puncture symmetries are explicit in the UV theory. The third puncture symmetry is obtained from $U(1)_\epsilon$ of the quiver. It has been argued in [38] using dualities that the global symmetry enhances to $SU(2)_\epsilon$ in the IR. Parameters $c_{1,2,3}$, $\tilde{c}_{1,2,3}$, $t$, $a$ parametrize Cartans of $E_8$ symmetry. There is also superpotential of the following form

$$W = \left( M_1 \tilde{M}_1 + M_2 \tilde{M}_2 \right) q + F_1^M M_1^2 + F_2^M M_2^2 + \varepsilon^{ijk} F_i Q_j Q_k + \varepsilon^{ijk} \tilde{F}_i \tilde{Q}_j \tilde{Q}_k \qquad (3.23)$$

where $F$ denotes various flip fields and in the last two terms indices run through $1, 2, 3$. These are crucial for the IR enhancement of the global symmetry to $SU(2)_\epsilon$.

The trinion used here has a generalization to $(D_{N+3}, D_{N+3})$ minimal conformal matter theories such that the maximal punctures have $SU(2)^N$ symmetry. We will thus refer to the version of the $BC_1$ van Diejen model derived here as the $(A_1)^{N=1}$ model. We will not consider this higher $N$ generalization explicitly in this paper.

The superconformal index of the theory described above is given by the following expression,

$$K_3^{A_1}(v,z,\epsilon) = \kappa^2 \oint \frac{dy_1}{4\pi i y_1} \oint \frac{dy_2}{4\pi i y_2} \frac{\prod_{i=1}^4 \Gamma_e\left((pq)^{\frac{1}{4}}\epsilon^{-\frac{1}{2}}c_i y_1^{\pm 1}\right) \Gamma_e\left((pq)^{\frac{1}{4}}\epsilon^{-\frac{1}{2}}y_2^{\pm 1}\widetilde{c}_i\right)}{\Gamma_e\left(y_1^{\pm 2}\right) \Gamma_e\left(y_2^{\pm 2}\right)}$$

$$\prod_{i=1}^3 \Gamma_e\left(\sqrt{pq}\epsilon c_i c_4\right) \Gamma_e\left(\sqrt{pq}\epsilon \widetilde{c}_i \widetilde{c}_4\right) \Gamma_e\left(\sqrt{pq}t^2 a^{\pm 2}\epsilon^{-1}\right)$$

$$\Gamma_e\left((pq)^{1/4}t^{-1}a\epsilon^{1/2}y_1^{\pm 1}z^{\pm 1}\right) \Gamma_e\left((pq)^{1/4}t^{-1}a^{-1}\epsilon^{1/2}z^{\pm 1}y_2^{\pm 1}\right) \Gamma_e\left(\sqrt{pq}\epsilon^{-1}z^{\pm 1}v^{\pm 1}\right)$$

$$\Gamma_e\left((pq)^{1/4}ta^{-1}\epsilon^{1/2}y_1^{\pm 1}v^{\pm 1}\right) \Gamma_e\left((pq)^{1/4}ta\epsilon^{1/2}v^{\pm 1}y_2^{\pm 1}\right) , \qquad (3.24)$$

Here $y_1$ and $y_2$ parametrize the two gauged SU(2) nodes and we also introduced fugacities $c_4 \equiv (c_1 c_2 c_3)^{-1}$ and $\tilde{c}_4 \equiv (\tilde{c}_1 \tilde{c}_2 \tilde{c}_3)^{-1}$. For each of three punctures there is an octet of operators in the fundamental representation of the puncture symmetry (and having R-charge $+1$) with the following charges:

$$\widehat{M}_v : \quad \{ta^{-1}c_1,\, ta^{-1}c_2,\, ta^{-1}c_3,\, ta^{-1}c_4,\, ta\widetilde{c}_1,\, ta\widetilde{c}_2,\, ta\widetilde{c}_3,\, ta\widetilde{c}_4\}$$
$$\widehat{M}_z : \quad \{t^{-1}ac_1,\, t^{-1}ac_2,\, t^{-1}ac_3,\, t^{-1}ac_4,\, t^{-1}a^{-1}\widetilde{c}_1,\, t^{-1}a^{-1}\widetilde{c}_2,\, t^{-1}a^{-1}\widetilde{c}_3,\, t^{-1}a^{-1}\widetilde{c}_4\} \quad (3.25)$$
$$\widehat{M}_\epsilon : \quad \{c_1^{-1}c_2^{-1},\, c_1^{-1}c_3^{-1},\, c_2^{-1}c_3^{-1},\, \widetilde{c}_1^{-1}\widetilde{c}_2^{-1},\, \widetilde{c}_1^{-1}\widetilde{c}_3^{-1},\, \widetilde{c}_2^{-1}\widetilde{c}_3^{-1},\, t^2a^{-2},\, t^2a^2\}$$

Now we can derive the A$\Delta$Os along the lines summarized in the Section 2. As we specified previously there are two possible sequences of operations in the gluing (3.11). In the $A_1$ case considered in the previous section we first glued two trinions, then closed punctures introducing defects and finally glued the tube theory obtained in this way to an arbitrary theory with an SU(2) global symmetry. Here, although the same approach can be used, we will take another route. Namely we first close punctures obtaining tube theories and then glue them together and to a theory associated to arbitrary surface with at least one maximal puncture.

We start with closing a puncture and obtaining a theory corresponding to a tube. This amounts to giving a vev to one of the moment map operators (3.25) and its derivatives. Unlike in $A_1$ case the punctures here are not exactly on the same footing since SU(2)$_\epsilon$ puncture is not present explicitly in the gauge theory. Let's consider particular setting closing $\epsilon$-puncture and acting on $v$ puncture.

To close the $\epsilon$ puncture we give a vev to one of $\widehat{M}_\epsilon$ moment maps. We have an octet of choices and to be concrete we choose the moment map component with the charges equal to $t^2a^2$. Giving vev to this operator is equivalent to setting $\epsilon = (pq)^{\frac{1}{2}}t^2a^2q^Kp^M$ in the index, with $K$ and $M$ being integers corresponding to the number of derivatives of the moment map in the operator we give vev to. As usual at these loci the index (3.24) has a pole and we aim at computing the residue at this pole. Just as in the $A_1$ case and everywhere else in the paper we choose $M = 0$ and $K = 0, 1$ for simplicity. In case $K = 0$ we calculate the residue and obtain the following expression for the index of $\mathcal{T}_{v,z}^2$,

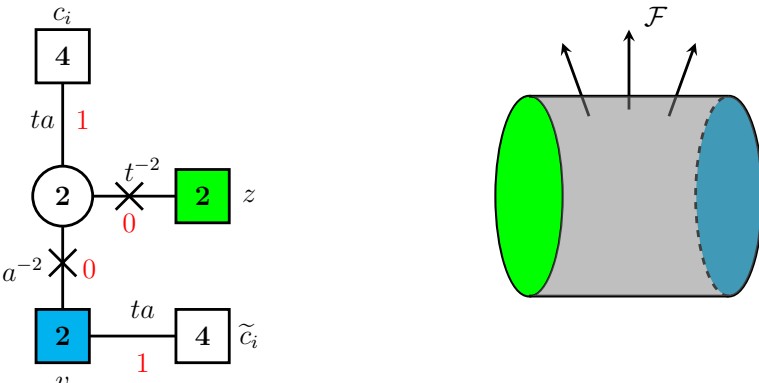

**Figure 8**. Quiver of the $A_1^1$ tube theory with $SU(2)_z$ and $SU(2)_v$ minimal punctures. Note that we assign here various charges to fields. These charges (with the exception of the flip field associated to the cross) are not fixed by a superpotential of this theory but rather by superpotentials one will need to turn on gluing this model to another theory.

$$K_{(2;0)}^{A_1^1}(v,z) = \kappa\Gamma(pqa^4)\Gamma_e(pqt^4)\oint\frac{dy_1}{4\pi i y_1}\frac{\prod_{i=1}^{4}\Gamma_e\left((pq)^{\frac{1}{2}}ta\widetilde{c}_i v^{\pm 1}\right)\Gamma_e\left((pq)^{\frac{1}{2}}tac_i y_1^{\pm 1}\right)}{\Gamma_e\left(y_1^{\pm 2}\right)}\times$$
$$\Gamma_e\left(a^{-2}y_1^{\pm 1}v^{\pm 1}\right)\Gamma_e\left(t^{-2}y_1^{\pm 1}z^{\pm 1}\right),\qquad(3.26)$$

which is gauge theory with the quiver shown on the Figure 8. Giving vev with $K = 1$, *i.e.* introducing defect into the tube theory we obtain the following index:

$$K_{(2;1)}^{A_1^1}(v,z) = \kappa\frac{\Gamma_e(v^2)}{\Gamma_e(qv^2)}\oint\frac{dy_1}{4\pi i y_1}\frac{\prod_{i=1}^{4}\Gamma_e\left(q^{-\frac{1}{2}}t^{-1}a^{-1}c_i y_1^{\pm 1}\right)}{\Gamma_e\left(y_1^{\pm 2}\right)}\Gamma_e\left((pq)^{1/2}q^{\frac{1}{2}}a^2 y_1^{\pm 1}z^{\pm 1}\right)\times$$
$$\Gamma_e\left((pq)^{1/2}q^{\frac{1}{2}}t^2 y_1^{\pm 1}v^{\pm 1}\right)\Gamma_e\left(t^{-2}a^{-2}vz^{\pm 1}\right)\Gamma_e\left(q^{-1}t^{-2}a^{-2}v^{-1}z^{\pm 1}\right)\times$$
$$\Gamma_e\left((pq)^{\frac{1}{2}}qta\widetilde{c}_i v\right)\Gamma_e\left((pq)^{\frac{1}{2}}ta\widetilde{c}_i v^{-1}\right)+\{v\leftrightarrow v^{-1}\}.\qquad(3.27)$$

Finally, we S glue tube (3.27) with a defect to a tube (3.26) with no defect and to a general theory with for example $v$-puncture and the index $\mathcal{I}(v)$. The computation is detailed in Appendix B.4. The end result is,

$$
\mathcal{O}_v^{(\epsilon;t^2a^2;1,0)} \cdot \mathcal{I}(v) = \frac{\prod_{i=1}^4 \theta_p\left((pq)^{\frac{1}{2}}t^{-1}ac_i^{-1}v\right)\theta_p\left((pq)^{\frac{1}{2}}t^{-1}a^{-1}\widetilde{c}_i^{-1}v\right)}{\theta_p(v^2)\theta_p(qv^2)}\mathcal{I}(qv)
$$

$$
+ \frac{\theta_p(q^{-1}t^{-4})\prod_{i=1}^4 \theta_p\left((pq)^{\frac{1}{2}}ta^3c_iv^{-1}\right)\theta_p\left((pq)^{\frac{1}{2}}ta\widetilde{c}_iv\right)}{\theta_p(v^2)\theta_p(a^4v^{-2})\theta_p(q^{-2}t^{-4}a^{-4})}\mathcal{I}(v)
$$

$$
+ \frac{\theta_p(q^{-1}a^{-4})\theta_p(q^{-1}t^{-4}a^{-4}v^2)\prod_{i=1}^4 \theta_p\left((pq)^{\frac{1}{2}}ta^{-1}c_iv\right)\theta_p\left((pq)^{\frac{1}{2}}ta\widetilde{c}_iv\right)}{\theta_p(q^{-2}t^{-4}a^{-4})\theta_p(a^{-4}v^2)\theta_p(v^2)\theta_p(q^{-1}v^{-2})}\mathcal{I}(v)
$$

$$
+ \{v \leftrightarrow v^{-1}\} \tag{3.28}
$$

This operator once again has the form of the van Diejen operator (3.1) with shift and constant parts. Looking on shift part only we can fix the dictionary between parameters of our gauge theory and octet of parameters $h_i$ of the van Diejen model as follows:

$$
h_i = t^{-1}ac_i^{-1}, \quad h_{i+4} = t^{-1}a^{-1}\widetilde{c}_i^{-1}, \quad i = 1, \ldots, 4. \tag{3.29}
$$

As for the constant part of the operator above we can notice that it is an elliptic function in $v$ with periods 1 and $p$. Mapping the parameters one can show that position of its poles and corresponding residues coincide with the poles (3.6) and residues (3.7) of the constant part of van Diejen model. Hence we can conclude that up to an irrelevant constant term operator (3.28) is just van Diejen operator (3.1).

In a similar manner we can derive A$\Delta$Os acting on any of the three punctures and obtained after closing one of the two remaining punctures giving vev to one of its moment maps. Results for some of these cases are summarized in the Appendix B.4. Studying various combinations of punctures and moment maps we conclude, just as in $A_1$ case, that A$\Delta$Os we obtain are always equal to the van Diejen model (3.1) up to an irrelevant constant (independent of $v, z, \epsilon$) shift. Parameters $h_i$ of these van Diejen Hamiltonians are given by the inverse U(1) charges of the moment maps of the puncture A$\Delta$O operator acts on. This can be seen in the example considered above where the map is given in (3.29). The parameters are indeed given by the octet of charges of $\widetilde{M}_v$ moment maps (3.25).

### 3.4  $C_1$ van Diejen model

Finally we consider one additional E-string trinion theory. This model can be in fact derived from the $A_1$ trinion and we give some details of this in Appendix B.5. As in the previous two cases the trinion we discuss here has a natural higher rank generalization such that two of the punctures become maximal ones with USp$(2N)$ global symmetry while the third one is

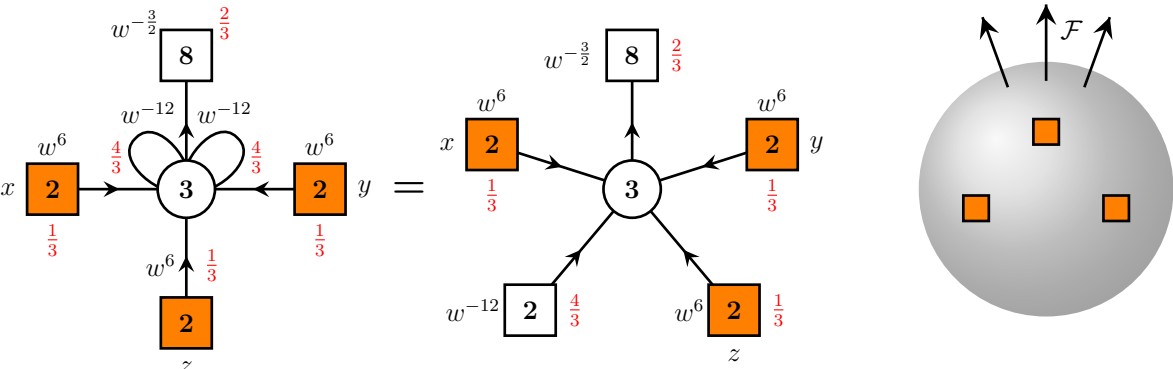

**Figure 9**. Quiver of the $C_1$ trinion theory. It is obtained taking $N = 1$ in the theory presented on the Figure 21 and corresponds to a three punctured sphere compactification of the E-string theory such that all the punctures are of the same type and the flux breaks $E_8$ to $SU(8) \times U(1)$. The line starting and ending on the SU(3) node denotes a field in antisymmetric two index representation which is the same here as fundamental representation. The SU(2) symmetry rotating the two fields in the symmetric representation is broken by a superpotential.

minimal SU(2) puncture. Hence we will refer to this trinion theory as $C_{N=1}$, see Appendix B.5. The quiver of the theory is shown on the Figure 9. It has $SU(2)^3$ global symmetry, where each SU(2) stands for the puncture. Unlike $A_1$ theory here all SU(2) punctures are of the same type.

The index of the $C_1$ trinion theory is given by,

$$
K_3^{C_1}(x, y, z) = \kappa^2 \oint \frac{dt_1}{2\pi i t_1} \frac{dt_2}{2\pi i t_2} \frac{1}{\prod_{i \neq j}^3 \Gamma_e(t_i/t_j)} \prod_{j=1}^3 \left[ \Gamma_e\left( (pq)^{\frac{1}{6}} w^6 t_j x^{\pm 1} \right) \times \right.
$$
$$
\left. \Gamma_e\left( (pq)^{\frac{1}{6}} w^6 t_j y^{\pm 1} \right) \Gamma_e\left( (pq)^{\frac{1}{6}} w^6 t_j z^{\pm 1} \right) \prod_{i=1}^8 \Gamma_e\left( (pq)^{\frac{1}{3}} w^{-\frac{3}{2}} a_i t_j^{-1} \right) \right] \times
$$
$$
\prod_{1 \leq j < k \leq 3} \Gamma_e\left( (pq)^{\frac{2}{3}} w^{-12} t_j^{-1} t_k^{-1} \right)^2 , \quad (3.30)
$$

where $x, y, z$ label the three punctures symmetries and $w, a_i$ parametrize $U(1) \times SU(8)$ subgroup of the $6d$ global $E_8$ symmetry. Each puncture has an octet of mesonic moment map operators with the following charges:

$$
M_a = \mathbf{2}_a \otimes \mathbf{8}_{w^{\frac{9}{2}}} , \qquad a = x, y, z . \quad (3.31)
$$

Since all punctures are of the same type the moment maps also have the same charges for each of them. As usual, we can close one of the punctures giving vev to one of the mesonic moment maps (3.31). Without loss of generality let us close $z$-puncture by setting $z =$

$(pq)^{-\frac{1}{2}}w^{-\frac{9}{2}}a_1^{-1}$ in the index (3.30). When $z$ takes this value the poles of $\Gamma_e\left((pq)^{-\frac{1}{3}}w^{\frac{3}{2}}t_j a_1^{-1}\right)$ and $\Gamma_e\left((pq)^{\frac{1}{3}}w^{-\frac{3}{2}}a_1 t_j^{-1}\right)$ collide leading to pinching of the integration contours. Computing the residue of this pole results in SU(3) gauge theory being higgsed down to SU(2) and the resulting index of the tube with no defect is given by:

$$
K_{(2;0)}^{C_1}(x,y) = \kappa \Gamma_e\left(pqw^{-\frac{27}{2}}a_1\right)^2 \oint \frac{dt_2}{2\pi i t_2} \frac{1}{\Gamma_e\left(t_2^{\pm 2}\right)} \Gamma_e\left((pq)^{\frac{1}{2}}w^{\frac{9}{2}}a_1 x^{\pm 1}\right) \times
$$
$$
\Gamma_e\left((pq)^{\frac{1}{2}}w^{\frac{9}{2}}a_1 y^{\pm 1}\right) \Gamma_e\left(w^{\frac{27}{4}}a_1^{-\frac{1}{2}}t_2^{\pm 1}x^{\pm 1}\right) \Gamma_e\left(w^{\frac{27}{4}}a_1^{-\frac{1}{2}}t_2^{\pm 1}y^{\pm 1}\right) \times
$$
$$
\Gamma_e\left((pq)^{\frac{1}{2}}w^{-\frac{45}{4}}a_1^{-\frac{1}{2}}t_2^{\pm 1}\right) \prod_{i=2}^{8} \Gamma_e\left((pq)^{\frac{1}{2}}w^{-\frac{9}{4}}a_i a_1^{\frac{1}{2}}t_2^{\pm 1}\right) \tag{3.32}
$$

At this point we can in princinple take the same route as we did in the case of $A_1^1$ trinion. Namely we can also consider closing $z$ puncture introducing the defect by computing the residue of the pole of the trinion index (3.30) at $z = (pq)^{-\frac{1}{2}}w^{-\frac{9}{2}}a_i^{-1}q^{-1}$. Then we should consider the gluing similar to (3.11). We take however an alternative route. We start with gluing the tube (3.32) to the trinion (3.30) and perform chain of Seiberg dualities. As a result of applying these dualities mesonic moment map we give vev to turns into baryonic one. So giving vev to it higgses gauge theory completely leading to the Wess-Zumino tube theory. S gluing this tube to the arbitrary theory with for example SU(2)$_x$ puncture we get the following A$\Delta$O,

$$
\mathcal{O}_x^{(z;w^{\frac{9}{2}}a_1;1,0)} \cdot \mathcal{I}(x) = \frac{\prod_{i=1}^{8} \theta_p\left((pq)^{\frac{1}{2}}w^{-\frac{9}{2}}a_i^{-1}x\right)}{\theta_p\left(x^2\right)\theta_p\left(qx^2\right)} \mathcal{I}(qx) + \{x \to x^{-1}\} +
$$
$$
\frac{\theta_p\left((pq)^{\frac{1}{2}}w^{18}x\right)\theta_p\left((pq)^{\frac{1}{2}}w^{\frac{9}{2}}a_i x^{\pm 1}\right)\prod_{j\neq i}^{8}\theta_p\left((pq)^{\frac{1}{2}}w^{-\frac{9}{2}}a_j^{-1}x\right)}{\theta_p\left((pq)^{\frac{1}{2}}w^{-18}x\right)\theta_p\left((pq)^{\frac{1}{2}}qw^{\frac{9}{2}}a_i x\right)\theta_p\left(x^2\right)\theta_p\left(q^{-1}x^{-2}\right)} \mathcal{I}(x) + \{x \to x^{-1}\} +
$$
$$
\frac{\theta_p\left(q^{-1}w^{\frac{27}{2}}a_i^{-1}\right)\prod_{j\neq i}^{8}\theta_p\left(q^{-1}w^{-9}a_i^{-1}a_j^{-1}\right)\theta_p\left((pq)^{\frac{1}{2}}w^{\frac{9}{2}}a_i x^{\pm 1}\right)}{\theta_p\left(q^{-2}w^{-9}a_i^{-2}\right)\theta_p\left(q^{-1}w^{-\frac{45}{2}}a_i^{-1}\right)\theta_p\left((pq)^{-\frac{1}{2}}q^{-1}w^{-\frac{9}{2}}a_i^{-1}x^{\pm 1}\right)} \mathcal{I}(x) +
$$
$$
\frac{\prod_{j\neq i}^{8}\theta_p\left(w^{\frac{27}{2}}a_j^{-1}\right)\theta_p\left((pq)^{\frac{1}{2}}w^{\frac{9}{2}}a_i x^{\pm 1}\right)}{\theta_p\left(qw^{\frac{45}{2}}a_i\right)\theta_p\left((pq)^{-\frac{1}{2}}w^{18}x^{\pm 1}\right)} \mathcal{I}(x) \tag{3.33}
$$

Details of this calculation can be found in the Appendix B.6. Just as before we can look at the shift part and see that it coincides with the shift part of the van Diejen model (3.1)

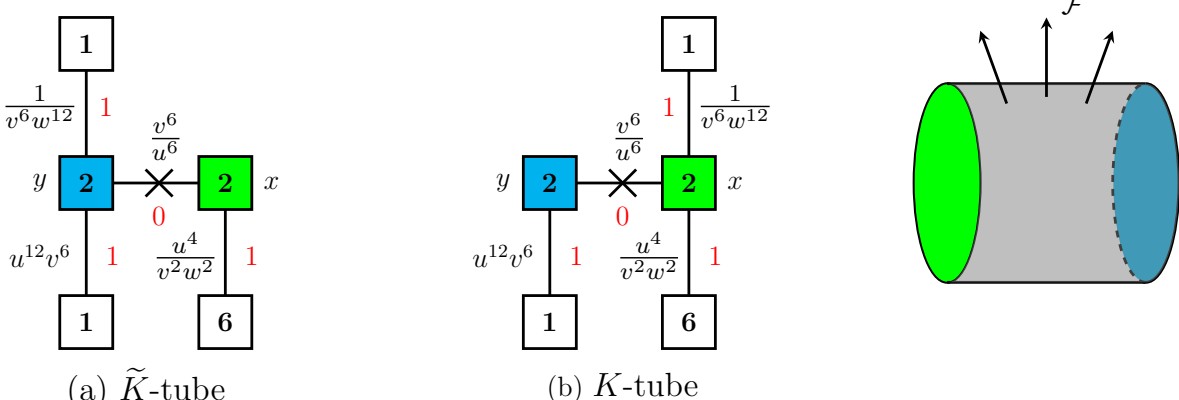

(a) $\widetilde{K}$-tube          (b) $K$-tube

**Figure 10**. Quiver of the tube theories. In Figure (a) we show tube with the conjugation with the index given (3.40). In Figure (b) we show tube theory with no conjugation and index given by (3.39). The theories differ only by singlets which are important once we start gluing these tubes to other surfaces.

and see that they are the same after the following identification:

$$h_i = (pq)^{\frac{1}{2}} w^{-\frac{9}{2}} a_i^{-1} \tag{3.34}$$

As in previous cases this identifications maps van Diejen parameters $h_i$ to the inverse U(1) charges of the moment maps (3.31) of the puncture we act on. The constant part of the operator (3.33) is an elliptic function with periods 1 and $p$, and poles with residues coinciding with those of the $BC_1$ van Diejen model upon identification (3.34). Hence once again we obtain van Diejen operator acting on the puncture. Since all punctures and all moment maps are on the same footing we can of course act with operators of exactly the same form on $y$ and $z$ punctures.

## 3.5 Duality properties

In this section we have discussed the derivation of a collection of A$\Delta$O operators specified in (3.20), (3.22), (3.28) and (3.33). It was also shown that all of these operators are actually $BC_1$ van Diejen operators (3.1) with octet of parameters given by the inverse charges of the moment maps of the puncture the operators act on.

As we have discussed in Section 2 operators we derived should posses a number of interesting and important properties that follow directly from their construction. In this section we will discuss checks and proofs of some of these properties using explicit expressions of the operators we have obtained.

Let us first discuss the commutation of the operators specified in (2.6). In terms of our derivations this means that all of the operators acting on the same type of the puncture should be commuting with each other. Namely we should have:

$$\left[\tilde{\mathcal{O}}_a^{(b;\xi_i^{(b)};1,0)}, \tilde{\mathcal{O}}_a^{(c;\xi_j^{(c)};1,0)}\right] = \left[\tilde{\mathcal{O}}_a^{(b;\xi_i^{(b)};0,1)}, \tilde{\mathcal{O}}_a^{(c;\xi_j^{(c)};0,1)}\right] = \left[\tilde{\mathcal{O}}_a^{(b;\xi_i^{(b)};1,0)}, \tilde{\mathcal{O}}_a^{(c;\xi_j^{(c)};0,1)}\right] = 0 , \tag{3.35}$$

where $a$ is the label denoting the type of the puncture we act on, $b$ and $c$ are types of the punctures we close, $\xi_i^{(b)}$ and $\xi_j^{(c)}$ are U(1) charges of the moment maps we give vev to. Finally labels $(1,0)$ and $(0,1)$ stand on the kind of the moment map derivative we give map to (corresponding to factors of either $q$ or $p$). Operators with these two choices are identical up to permutation of $p$ and $q$ parameters. We would like to check this identity for all of the operators we have derived. However since all of these operators are actually $BC_1$ van Diejen operators (3.1) (up to a constant shift) with parameters depending only on the type of puncture operators act on, these operators do trivially commute.

The more complicated property to check is the kernel property (2.7). As explained in Section 2 superconformal index of any 4d model obtained in compactifications of the E-string theory with several maximal punctures is the kernel function of our difference operators. One interesting conclusion of our paper is that the indices of all the trinions as well as indices of the tubes we derive from them can be used as the kernel functions of van Diejen model. Despite main kernel function property (2.7) directly follows from our geometrical construction shown on the Figure 3 we would like to present here some checks of it in particular cases.

Good examples of kernel functions of van Diejen model are superconformal indices of all trinions (3.8), (3.24) and (3.30) and tubes (3.32), (3.26). For example if we take index of $A_1$ trinion theory it should satisfy the following property:

$$\tilde{\mathcal{O}}_x^{(z;u^{12}w^6;1,0)} \cdot K_3^{A_1}(x,y,z) = \tilde{\mathcal{O}}_y^{(z;u^{12}w^6;1,0)} \cdot K_3^{A_1}(x,y,z) \tag{3.36}$$

Instead of this particular choice of the operator we can choose any other $A_1$ operator from (3.20) and (3.22). This relation is very hard to proof due to technical reasons. However we can study explicitly the tubes of $A_1$ theory which are on one hand much simpler and on the other hand are of course also expected to be Kernel functions. Index for this tube theory can be obtained by computing the residue of the trinion index (3.8) at for example $z = (pq)^{-\frac{1}{2}} u^{-12} w^{-6}$. This corresponds to closing $z$-puncture by giving vev to a baryon $\mathbf{1}_{u^{12}w^6}$. At this value of $z$ pinching of the integration contours of (3.8) happens. In particular integrand of this trinion index has poles at

$$(pq)^{\frac{1}{6}} w^6 t_1 z = q^{-k_1} p^{-m_1}, \quad (pq)^{\frac{1}{6}} u^6 t_2 x = q^{-k_2} p^{-m_2}, \quad (pq)^{\frac{1}{6}} u^6 t_3 x^{-1} = q^{-k_3} p^{-m_3}. \tag{3.37}$$

Using SU(3) constraint $\prod_{i=1}^3 t_i$ we can rewrite these sets of poles as two sets of poles in for example $t_1$

$$t_1 = t_2^{-1} t_3^{-1} = (pq)^{\frac{1}{3}} u^{12} q^{k_2+k_3} p^{m_2+m_3}, \qquad t_1 = (pq)^{-\frac{1}{6}} w^{-6} z^{-1} q^{-k_1} p^{-m_1}, \tag{3.38}$$

where the first set of poles is inside the integration contour while the second one is outside. Whenever $z = (pq)^{-\frac{1}{2}} u^{-12} w^{-6} q^{-k_1-k_2-k_3} p^{-m_1-m_2-m_3}$ two lines of poles collide and pinch the contour. Choosing $k_i = m_i = 0, \forall i$ we obtain the following expression for the index of the tube Wess-Zumino theory without defect:

$$K_{(2;0)}^{A_1}(x,y) = \Gamma_e\left(pqw^{12}u^{24}\right)\Gamma_e\left((pq)^{\frac{1}{2}}u^{12}v^6y^{\pm1}\right)\Gamma_e\left(u^{-6}v^6x^{\pm1}y^{\pm1}\right)\times$$

$$\Gamma_e\left((pq)^{\frac{1}{2}}u^6w^{12}x^{\pm1}\right)\left[\prod_{j=1}^{6}\Gamma_e\left(u^{-14}v^{-2}w^{-2}a_j\right)\Gamma_e\left((pq)^{\frac{1}{2}}u^4v^{-2}w^{-2}x^{\pm1}a_j\right)\right]. \qquad (3.39)$$

However this index is expected to be the kernel function of the operator of (3.13) rather than the $BC_1$ van Diejen operator. In order to write down the kernel of the latter one we should take into account the flips of the moment maps $M_u$ and $M_v$ as in (3.16). It leads to the following proposal for the kernel function:

$$\tilde{K}_{(2;0)}^{A_1}(x,y) \equiv \Gamma_e\left((pq)^{\frac{1}{2}}u^{-6}w^{-12}x^{\pm1}\right)\Gamma_e\left((pq)^{\frac{1}{2}}v^{-6}w^{-12}y^{\pm1}\right)K_{(2;0)}^{A_1}(x,y) =$$

$$\Gamma_e\left(pqw^{12}u^{24}\right)\Gamma_e\left((pq)^{\frac{1}{2}}u^{12}v^6y^{\pm1}\right)\Gamma_e\left(u^{-6}v^6x^{\pm1}y^{\pm1}\right)\Gamma_e\left((pq)^{\frac{1}{2}}v^{-6}w^{-12}y^{\pm1}\right)\times$$

$$\left[\prod_{j=1}^{6}\Gamma_e\left(u^{-14}v^{-2}w^{-2}a_j\right)\Gamma_e\left((pq)^{\frac{1}{2}}u^4v^{-2}w^{-2}x^{\pm1}a_j\right)\right]. \qquad (3.40)$$

The quivers of the corresponding theories are shown on the Figure 10. Now since we have no integrals in this expression it is straightforward to check kernel property

$$\tilde{\mathcal{O}}_x^{(z;u^{12}w^6;1,0)}\cdot\tilde{K}_{(2;0)}^{A_1}(x,y) = \tilde{\mathcal{O}}_y^{(z;u^{12}w^6;1,0)}\cdot\tilde{K}_{(2,0)}^{A_1}(x,y), \qquad (3.41)$$

where particular expressions for the operators can be read from (3.20) and (3.22). This identity was discussed in [70] (for completeness we prove it also in Appendix E.1). It would be interesting to directly check whether the indices of all of the trinions are indeed kernel functions of the $BC_1$ van Diejen model.

# 4 $A_N$ generalizations

In this section we will generalize considerations of the previous section to the compactifications of the minimal $(D_{N+3}, D_{N+3})$ minimal conformal matter with $N \geq 1$. We discussed three versions of E-string compactifications that upon generalization to the higher rank should lead to three different trinion theories and hence three different finite difference integrable Hamiltonians. We will concentrate only on one of these three possible generalizations. Namely we will consider $A_N$ case which is a direct generalization of the $A_1$ van Diejen model considered in the Section 3.2. We will obtain a system of $(4N+12)$ A$\Delta$Os depending on $p$, $q$ and $(2N+6)$ extra parameters each.

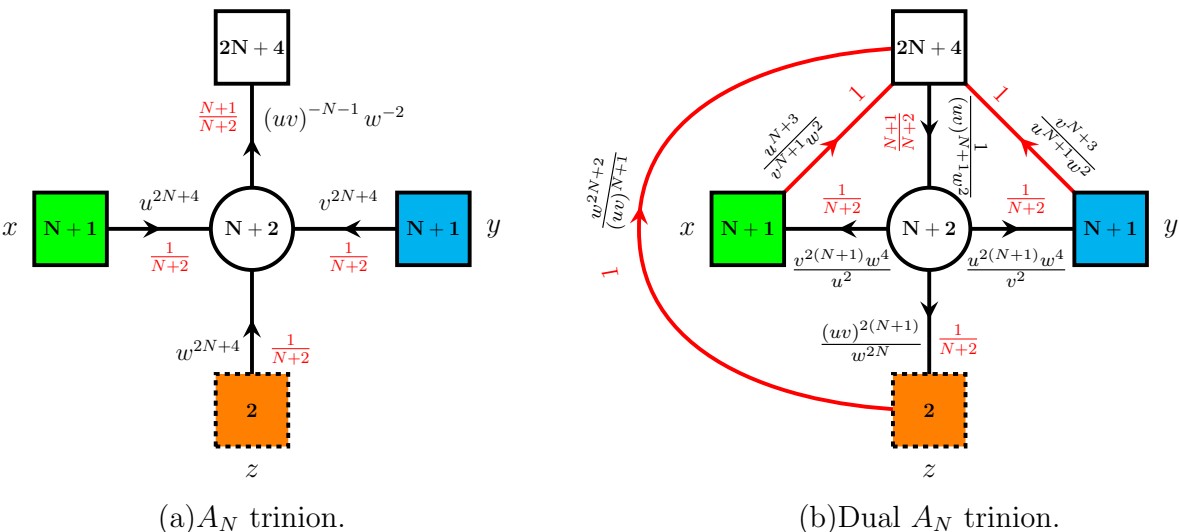

(a)$A_N$ trinion.  (b)Dual $A_N$ trinion.

**Figure 11**. Quiver of the $A_N$ trinion theory and its Seiberg dual. This trinion is obtained by compactifying the $(D_{N+3}, D_{N+3})$ minimal conformal matter theory on a sphere with two maximal $\mathrm{SU}(N+1)$ punctures and a minimal $\mathrm{SU}(2)$ puncture with flux breaking the $SO(12+4N)$ symmetry to $SO(8+4N) \times \mathrm{SU}(2) \times \mathrm{U}(1)$.

The $(D_{N+3}, D_{N+3})$ minimal conformal matter theory is a 6d $(1,0)$ SCFT. The six dimensional symmetry group is $G_{6d} = SO(12+4N)$. For the case of $N = 1$ the $SO(16)$ symmetry enhances to $E_8$. Upon compactification to 5d there are three different possible effective gauge theory descriptions (depending on holonomies turned on the compactification circle): with either $SU(2)^N$, $\mathrm{USp}(2N)$, or $SU(N+1)$ gauge groups. The descriptions relevant to us are the latter one. We will consider maximal punctures with $G_{5d} = SU(N+1)$ and a minimal $\mathrm{SU}(2)$ puncture which can be obtained by partially closing the $\mathrm{USp}(2N)$ puncture. Note that for $N = 1$ the three descriptions in 5d coincide.

The trinion theory $\mathcal{T}^{\mathcal{A}}_{x,y,z}$ we will use was obtained in [34]. It is $SU(N+2)$ $\mathcal{N} = 1$ SQCD with $(2N+4)$ flavors. The quiver of the theory is shown on the Figure 11. While in $A_1$ case all of our punctures were minimal now in higher rank case we have two maximal punctures with $SU(N+1)$ global symmetry and one minimal puncture with $\mathrm{SU}(2)$ symmetry.

The index of the theory is given by

$$
K_3^{A_N}(x,y,z) = \kappa_{N+1} \oint \prod_{i=1}^{N+1} \frac{dt_i}{2\pi i t_i} \prod_{i \neq j}^{N+2} \frac{1}{\Gamma_e\left(\frac{t_i}{t_j}\right)} \prod_{i=1}^{N+2} \prod_{j=1}^{N+1} \Gamma_e\left((pq)^{\frac{1}{2(N+2)}} u^{2N+4} t_i x_j\right) \times
$$

$$
\Gamma_e\left((pq)^{\frac{1}{2(N+2)}} v^{2N+4} t_i y_j\right) \Gamma_e\left((pq)^{\frac{1}{2(N+2)}} w^{2N+4} t_i z^{\pm 1}\right) \times
$$

$$
\prod_{l=1}^{2N+4} \Gamma_e\left((pq)^{\frac{N+1}{2(N+2)}} (uv)^{-N-1} w^{-2} t_i^{-1} a_l\right) , \quad (4.1)
$$

where

$$\kappa_N = \frac{(p;p)_\infty^{N-1} (q;q)_\infty^{N-1}}{N!} \,. \tag{4.2}$$

$\mathrm{SU}(N+2)$ gauge symmetry is parametrized by $t_i$'s with the relation

$$\prod_{i=1}^{N+2} t_i = 1 \,. \tag{4.3}$$

Global $\mathrm{SU}(N+1)$ symmetries of the maximal punctures are parametrized by $x_i$ and $y_i$ satisfying

$$\prod_{j=1}^{N+1} x_j = \prod_{j=1}^{N+1} y_j = 1 \,. \tag{4.4}$$

Each puncture has $(2N+6)$ moment map operators with the following charges:

$$
\begin{aligned}
M_u &= \mathbf{N+1}^x \otimes \left( \mathbf{2N+4}_{u^{N+3} v^{-N-1} w^{-2}} \oplus \mathbf{1}_{(uv^{N+1})^{2N+4}} \right) \oplus \overline{\mathbf{N+1}}^x \otimes \mathbf{1}_{(u^N w^2)^{2N+4}} \,, \\
M_v &= \mathbf{N+1}^y \otimes \left( \mathbf{2N+4}_{v^{N+3} u^{-N-1} w^{-2}} \oplus \mathbf{1}_{(vu^{N+1})^{2N+4}} \right) \oplus \overline{\mathbf{N+1}}^y \otimes \mathbf{1}_{(v^N w^2)^{2N+4}} \,, \\
M_w &= \mathbf{2}^z \otimes \left( \mathbf{2N+4}_{(uvw^{-2})^{-N-1}} \oplus \mathbf{1}_{(wv^{N+1})^{2N+4}} \oplus \mathbf{1}_{(wu^{N+1})^{2N+4}} \right) \,,
\end{aligned}
\tag{4.5}
$$

so that each moment map has 2 baryonic and $2N+4$ mesonic components. Notice also that one of the baryons in $M_u$ and $M_v$ transforms in the antifundamental representation of the puncture global symmetry.

We will proceed along similar lines as in the $A_1$ case considered in Section 3. We will start by considering particular example of the finite difference operator A$\Delta$O. Then we will proceed with giving general expressions covering all possible combinations of puncture we act on and moment map we give vev to in order to introduce defects. Finally we will discuss the duality properties (2.6) and (2.7).

## 4.1 Closing with $\mathbf{1}_{(wu^{N+1})^{2N+4}}$ vev.

The $A_N$ trinion we consider here has one minimal puncture with symmetry $\mathrm{SU}(2)_z$ and we close it by giving vev to one of the $M_w$ moment maps.

We follow the usual algorithm summarized in the Section 2. We start with some arbitrary theory $\mathcal{T}_a^{\mathcal{F}}$ with $\mathrm{SU}(N+1)_a$ maximal puncture and index $\mathcal{I}(a)$, where $a = x$ or $a = y$. For concreteness we consider the theory with an $\mathrm{SU}(N+1)_x$ puncture. Then we also glue two trinions $\mathcal{T}_{x,y,z}^{\mathcal{A}}$ and conjugated one $\bar{\mathcal{T}}_{\tilde{x},y,\tilde{z}}^{\mathcal{A}}$ along $\mathrm{SU}(N+1)_y$ puncture resulting in the four-punctured sphere theory $\mathcal{T}_{x,\tilde{x},z,\tilde{z}}$ shown on the Figure 12. This theory is just $\mathrm{SU}(N+3)$ SQCD with $(2N+6)$ multiplets and some extra flip singlets. The index $K_4^{A_N}(x,\tilde{x},z,\tilde{z})$ of this theory is specified in (B.4). Finally we can S glue this four-punctured sphere to our theory with $\mathrm{SU}(N+1)_{\tilde{x}}$ maximal puncture by gauging this symmetry.

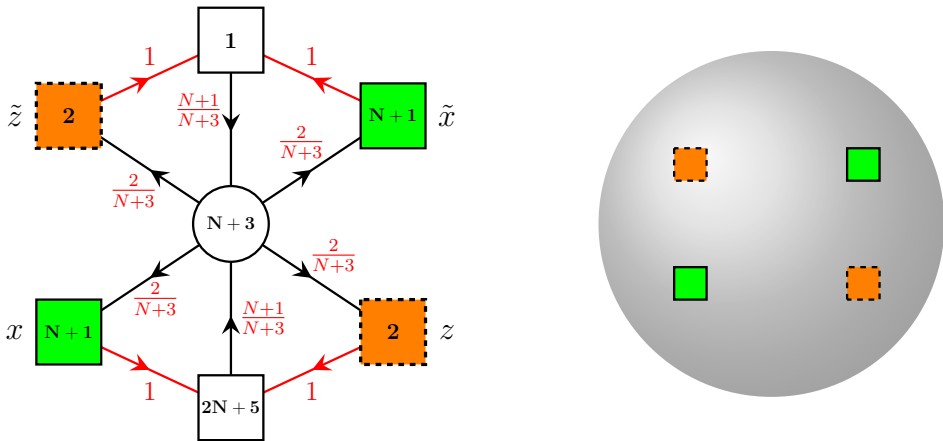

**Figure 12**. Quiver of the $A_N$ four-punctured sphere theory obtained by S gluing two three punctured spheres. Note that there are superpotentials corresponding to the triangles as well as certain baryonic superpotentials preserving the puncture symmetries.

Now we can close $z$ and $\tilde{z}$ minimal punctures which will result in obtaining expression for the index of the tube theory with the defect introduced. For now let us choose to close these punctures by giving vev to the baryon $\mathbf{1}_{(wu^{N+1})^{2N+4}}$. This amounts to giving the following weights to $z$ and $\tilde{z}$ variables:

$$z = (pq)^{-\frac{1}{2}} \left(wu^{(N+1)}\right)^{-(2N+4)} q^{-K} p^{-M} , \qquad \tilde{z} = (pq)^{-\frac{1}{2}} \left(wu^{(N+1)}\right)^{(2N+4)} q^{-\tilde{K}} p^{-\tilde{M}} , (4.6)$$

where $K$, $M$, $\tilde{K}$, $\tilde{M}$ are positive integers. In order to introduce simplest single defect we should choose $M = \tilde{M} = 0$ and either $K = 1 , \tilde{K} = 0$ or $K = 0 , \tilde{K} = 1$. Let's start with the first case corresponding to the $\mathrm{SU}(2)_z$ puncture closed with the defect introduced while $\mathrm{SU}(2)_{\tilde{z}}$ puncture is closed without a defect. This choice of $z$ and $\tilde{z}$ as usually corresponds to the pole of the index originating from the contour pinchings. Computing the residue of this pole and higgsing all of the gauge symmetries we finally obtain the following finite difference operator A$\Delta$O:

$$\mathcal{O}_x^{(\xi;1,0)} \cdot \mathcal{I}(x) \equiv \left(\sum_{l \neq m}^{N+1} A_{lm}^{(\xi;1,0)}(x) \Delta_{lm}^{(1,0)} + W^{(\xi;1,0)}(x)\right) \mathcal{I}(x), \quad \xi \equiv \left(wu^{N+1}\right)^{2N+4} ,$$

$$(4.7)$$

where we have introduced the operator $\Delta_{lm}$ shifting two of the $x$ variables in opposite directions as follows:

$$\Delta_{lm}^{(1,0)}(x) f(x) \equiv f\left(x_l \to q^{-1} x_l , x_m \to q x_m\right) , \tag{4.8}$$

The shift part of the operator is given by

$$A_{lm}^{(\xi;1,0)}(x) \equiv \frac{\theta_p\left((pq)^{\frac{1}{2}}u^{-2N-4}v^{-(N+1)(2N+4)}x_l^{-1}\right)\theta_p\left((pq)^{\frac{1}{2}}w^{-4N-8}u^{-N(2N+4)}x_m\right)}{\theta_p\left(q\frac{x_m}{x_l}\right)\theta_p\left(\frac{x_m}{x_l}\right)} \times$$

$$\prod_{j=1}^{2N+4}\theta_p\left((pq)^{\frac{1}{2}}u^{-N-3}v^{N+1}w^2x_l^{-1}a_j^{-1}\right) \times$$

$$\prod_{i\neq m\neq l}^{N+1}\frac{\theta_p\left((pq)^{\frac{1}{2}}w^{4N+8}u^{N(2N+4)}x_i^{-1}\right)\theta_p\left((pq)^{\frac{1}{2}}u^{(N+2)(2N+4)}x_i\right)}{\theta_p\left(\frac{x_i}{x_l}\right)\theta_p\left(\frac{x_m}{x_i}\right)} , \quad (4.9)$$

Notice that there is sharp difference with the $A_1$ case in (3.13). The latter one depended only on the charges of the moment maps of the puncture we act on. On the other hand it is obvious that while the terms in the first two lines of the expression above also depend on the charges of the moment maps $M_u$ the terms in the last line depend on the charge of the moment map we gave vev to. When $N = 1$ this last product is absent and we return to the expression in (3.13)

Finally the constant part of the operator is given by

$$W^{(\xi;1,0)}(x) \equiv \prod_{j=1}^{2N+4}\theta_p\left(u^{-(N+1)(2N+5)}v^{-N-1}w^{-2}q^{-1}a_j\right)\frac{\theta_p\left(q^{-1}\left(vu^{-1}\right)^{(N+1)(2N+4)}\right)}{\theta_p\left(pq^2w^{4N+8}u^{2(N+1)(2N+4)}\right)} \times$$

$$\prod_{i=1}^{N+1}\frac{\theta_p\left((pq)^{\frac{1}{2}}w^{4N+8}u^{N(2N+4)}x_i^{-1}\right)}{\theta_p\left((pq)^{-\frac{1}{2}}u^{-(N+2)(2N+4)}q^{-1}x_i^{-1}\right)} + \sum_{m=1}^{N+1}\frac{\theta_p\left((pq)^{\frac{1}{2}}u^{2N+4}v^{(N+1)(2N+4)}x_m\right)}{\theta_p\left((pq)^{\frac{1}{2}}u^{(N+2)(2N+4)}qx_m\right)} \times$$

$$\prod_{i\neq m}\frac{\theta_p\left((pq)^{\frac{1}{2}}w^{4N+8}u^{N(2N+4)}x_i^{-1}\right)\theta_p\left((pq)^{\frac{1}{2}}u^{(N+2)(2N+4)}x_i\right)}{\theta_p\left(q^{-1}\frac{x_i}{x_m}\right)\theta_p\left(\frac{x_m}{x_i}\right)} \times$$

$$\prod_{j=1}^{2N+4}\theta_p\left((pq)^{\frac{1}{2}}u^{N+3}v^{-N-1}w^{-2}x_ma_j\right) . \quad (4.10)$$

This constant term is a direct generalization of the one appearing the $A_1$ discussion and depends on all of the moment map charges $M_u$ as well as on the choice of the operator we give vev to. Details of the derivation of this operator can be found in the Appendix B.2.

In $N = 1$ case the operator derived above reproduces operator $\mathcal{O}_x^{(z;u^{12}w^6;1,0)}$ given in (3.13). In Section 3.2 we have also introduced operator $\tilde{\mathcal{O}}_x^{(z;u^{12}w^6;1,0)}$ obtained by conjugation (3.16) of the operator $\mathcal{O}_x^{(z;u^{12}w^6;1,0)}$. The resulting A$\Delta$O was identical to the $BC_1$ van Diejen A$\Delta$O. Now in $A_N$ case we also can perform conjugation which takes the following form,

$$\tilde{\mathcal{O}}^{(\xi;1,0)} = \prod_{i=1}^{N+1}\Gamma_e\left((pq)^{\frac{1}{2}}\left(u^Nw^2\right)^{-2N-4}x_i\right)\mathcal{O}^{(\xi;1,0)}\prod_{i=1}^{N+1}\Gamma_e\left((pq)^{\frac{1}{2}}\left(u^Nw^2\right)^{-2N-4}x_i\right)^{-1} ,$$

$$(4.11)$$

Only the shift part (4.9) of the operator is affected by this conjugation. It changes to the following expression:

$$\tilde{A}_{lm}^{(\xi;1,0)}(x) \equiv$$

$$\prod_{i=1}^{N+1} \Gamma_e \left( (pq)^{\frac{1}{2}} \left( u^N w^2 \right)^{-2N-4} x_i \right) \tilde{A}_{lm}^{(\xi;1,0)}(x) \prod_{i=1}^{N+1} \Gamma_e \left( (pq)^{\frac{1}{2}} \left( u^N w^2 \right)^{-2N-4} x_i \right)^{-1} =$$

$$\frac{\prod_{j=1}^{2N+6} \theta_p \left( (pq)^{\frac{1}{2}} h_j^{-1} x_l^{-1} \right)}{\theta_p \left( q \frac{x_m}{x_l} \right) \theta_p \left( \frac{x_m}{x_l} \right)} \prod_{i \neq m \neq l}^{N+1} \frac{\theta_p \left( (pq)^{\frac{1}{2}} w^{4N+8} u^{N(2N+4)} x_i^{-1} \right) \theta_p \left( (pq)^{\frac{1}{2}} u^{(N+2)(2N+4)} x_i \right)}{\theta_p \left( \frac{x_i}{x_l} \right) \theta_p \left( \frac{x_m}{x_i} \right)},$$

(4.12)

where $h_j$ are the U(1) charges of the moment map operators of $SU(N+1)_x$ with one of the baryonic operators flipped:

$$\widetilde{M}_u = \mathbf{N} + \mathbf{1}^x \otimes \left( \mathbf{2N+4}_{u^{N+3}v^{-N-1}w^{-2}} \oplus \mathbf{1}_{(uv^{N+1})^{2N+4}} \oplus \mathbf{1}_{(u^N w^2)^{-2N-4}} \right),$$

$$\widetilde{M}_v = \mathbf{N} + \mathbf{1}^y \otimes \left( \mathbf{2N+4}_{v^{N+3}u^{-N-1}w^{-2}} \oplus \mathbf{1}_{(vu^{N+1})^{2N+4}} \oplus \mathbf{1}_{(v^N w^2)^{-2N-4}} \right),$$

(4.13)

while the moment maps $M_w$ remain the same. Notice that when $N = 1$ these moment maps are exactly the same as the ones in (3.18) we have obtained flipping one of the baryons in $A_1$ model. However in $A_1$ case the flip did not have any good motivation except that the operators (3.20) and (3.22) are exactly equal to the van Diejen operator (3.1). In the higher rank case it is obvious that the choice of the moment map to be flipped is determined by the form of the moment maps (4.5) itself. In particular we can see that one of the baryons in $M_u$ and $M_v$ moment maps transforms in the antifundamental representation of the $SU(N+1)$ global symmetry of the corresponding maximal puncture. Flipping this baryon we arrive at the moment maps (4.13) all of which transform in the fundamental representation. Hence the choice of the particular operator and the flip itself is clear and natural. Since this flip affects only the shift part and keeps constant part (4.10) the same we arrive at the following operator,

$$\tilde{\mathcal{O}}_x^{(\xi;1,0)} \cdot \mathcal{I}(x) \equiv \left( \sum_{l \neq m}^{N+1} \tilde{A}_{lm}^{(\xi;1,0)}(x) \Delta_{lm}^{(1,0)} + W^{(\xi;1,0)}(x) \right) \mathcal{I}(x), \quad \xi \equiv \left( wu^{N+1} \right)^{2N+4},$$

(4.14)

where $\tilde{A}_{lm}^{(\xi;1,0)}(x)$ is given in (4.12) and $\Delta_{lm}^{(1,0)}$ is the same shift operator given in (4.8).

One more thing to be discussed here is an alternative way to close the punctures. In particular we can make an alternative choice of integers $K, \tilde{K}, M, \tilde{M}$ in (4.6). Previously we introduced defect closing $SU(2)_z$ puncture by choosing $K = 1$ and $\tilde{K} = 0$. But we

can make an opposite choice $K = 0$, $\tilde{K} = 1$ corresponding to the defect introduced closing conjugated $SU(2)_{\tilde{z}}$ puncture. Alternatively the same goal can be achieved by introducing defect in $SU(2)_z$ puncture but also flipping baryons of both $z$ and $\tilde{z}$ that we give vev to. At the level of index this reduces to including the flip singlets contribution of the form $\Gamma_e\left((pq)^{\frac{1}{2}}\left(u^{N+1}w\right)^{-2N-4}z^{\pm 1}\right)\Gamma_e\left((pq)^{\frac{1}{2}}\left(u^{N+1}w\right)^{2N+4}\tilde{z}^{\pm 1}\right)$. One way or another we give space dependent vev to the baryon with the U(1) charge $\xi^{-1} = \left(wu^{N+1}\right)^{-2N-4}$. Both calculations lead to the same finite difference operator. After performing conjugation (4.11) this operator takes the following form:

$$
\tilde{\mathcal{O}}_x^{(\xi^{-1};1,0)} \cdot \mathcal{I}(x) \equiv \left(\sum_{l \neq m}^{N+1} \tilde{A}_{lm}^{(\xi^{-1};1,0)}(x)\Delta_{lm}^{(1,0)} + W^{(\xi^{-1};1,0)}(x)\right)\mathcal{I}(x), \quad \xi \equiv \left(wu^{N+1}\right)^{2N+4},
$$

$$(4.15)$$

where the shift part is given by

$$
\tilde{A}_{lm}^{(\xi^{-1};1,0)}(x) \equiv \frac{\prod\limits_{j=1}^{2N+6}\theta_p\left((pq)^{\frac{1}{2}}h_j^{-1}x_l^{-1}\right)}{\theta_p\left(q\frac{x_m}{x_l}\right)\theta_p\left(\frac{x_m}{x_l}\right)} \times
$$
$$
\prod_{i \neq m \neq l}^{N+1}\frac{\theta_p\left((pq)^{\frac{1}{2}}w^{-4N-8}u^{-N(2N+4)}x_i\right)\theta_p\left((pq)^{\frac{1}{2}}u^{-(N+2)(2N+4)}x_i^{-1}\right)}{\theta_p\left(\frac{x_i}{x_l}\right)\theta_p\left(\frac{x_m}{x_i}\right)}, \quad (4.16)
$$

and the constant part is

$$
W_x^{(\xi^{-1};1,0)}(x) \equiv \prod_{j=1}^{2N+4}\theta_p\left(u^{(N+1)(2N+5)}v^{N+1}w^2q^{-1}a_j^{-1}\right)\frac{\theta_p\left(q^{-1}\left(uv^{-1}\right)^{(N+1)(2N+4)}\right)}{\theta_p\left(q^{-2}w^{4N+8}u^{2(N+1)(2N+4)}\right)} \times
$$
$$
\prod_{i=1}^{N+1}\frac{\theta_p\left((pq)^{\frac{1}{2}}w^{-4N-8}u^{-N(2N+4)}x_i\right)}{\theta_p\left((pq)^{-\frac{1}{2}}u^{(N+2)(2N+4)}q^{-1}x_i\right)} + \sum_{m=1}^{N+1}\frac{\theta_p\left((pq)^{\frac{1}{2}}u^{-2N-4}v^{-(N+1)(2N+4)}x_m^{-1}\right)}{\theta_p\left((pq)^{\frac{1}{2}}u^{-(N+2)(2N+4)}qx_m^{-1}\right)} \times
$$
$$
\prod_{j=1}^{2N+4}\theta_p\left((pq)^{\frac{1}{2}}u^{-N-3}v^{N+1}w^2x_m^{-1}a_j^{-1}\right) \times
$$
$$
\prod_{i \neq m}\frac{\theta_p\left((pq)^{\frac{1}{2}}w^{-4N-8}u^{-N(2N+4)}x_i\right)\theta_p\left((pq)^{\frac{1}{2}}u^{-(N+2)(2N+4)}x_i^{-1}\right)}{\theta_p\left(q^{-1}\frac{x_m}{x_i}\right)\theta_p\left(\frac{x_i}{x_m}\right)}. \quad (4.17)
$$

Details of the calculations of this operator can be found in the Appendix B.3. One interesting feature of the operators derived above is dependence of the shift parts (4.12) and

(4.16) on the fugacities of the moment maps we give vev to. In $N = 1$ case considered in the previous section these shift parts in turn depend only on the fugacities of the puncture operator acts on.

## 4.2 General expression

We can proceed with general expression covering all possible combinations of punctures we act on and operators we give a vev to. We will always be closing $\mathrm{SU}(2)_z$ minimal puncture giving vev to various components of the $M_w$ moment maps operators in (4.5). As an input in each case we have $(2N + 6)$ parameters $h_i$ of the moment map operators we act on. We will always consider A$\Delta$Os with conjugations similar to (4.11) already included. Hence instead of $h_i$ charges it is convenient to consider charges of the moment maps (4.13) with the required flip already included. We will denote corresponding $(2N + 6)$-plet of charges as $\tilde{h}_i$.

Let us assume that we act on $\mathrm{SU}(N + 1)_{x^{(a)}}$ maximal puncture with the conjugations (4.11) already in place. The index $a$ can be $u$ or $v$ with $x^{(u)} \equiv x$ and $x^{(v)} \equiv y$. Charges of the corresponding moment maps $\widetilde{M}_a$ with the flip included are $\tilde{h}_i^{(a)}$. Now assume we give a space dependent vev to one of the minimal puncture moment map operators the $\mathrm{U}(1)$ charge $\hat{h} = \tilde{h}_i^{(a)}(\tilde{h}^{(a)})^{-\frac{1}{4}}$ where $\tilde{h}^{(a)} \equiv \prod_{i=1}^{2N+6} \tilde{h}_i^{(a)}$. Notice that a component of the moment map operator with weight $\hat{h}$ is not necessarily present among the $M_w$ operators. However, if the operator with this weight is not there then necessarily an operator with the conjugated weight, $\hat{h}^{-1}$, is and thus we can introduce a flip field to change the weight of the moment map component to $\hat{h}$. Without loss of generality for notational convenience we will assume then that he operator with weight $\hat{h}$ is present. The flip of the moment maps depends on the puncture we act on. For example if we act on the $\mathrm{SU}(N + 1)_x$ $\tilde{h}^{(u)} = (uw^{-1})^{8(N+2)}$ the weights after the flip can be read from:

$$\widetilde{M}_w^{(u)} \equiv \mathbf{2}^z \otimes \left( \mathbf{2N + 4}_{(uvw^{-2})^{-N-1}} \oplus \mathbf{1}_{(wv^{N+1})^{2N+4}} \oplus \mathbf{1}_{(wu^{N+1})^{-2N-4}} \right) , \qquad (4.18)$$

In case we act on the $\mathrm{SU}(N + 1)_y$ puncture we have $\tilde{h}^{(v)} = (vw^{-1})^{8(N+2)}$ the weights after the flip are given by

$$\widetilde{M}_w^{(v)} \equiv \mathbf{2}^z \otimes \left( \mathbf{2N + 4}_{(uvw^{-2})^{-N-1}} \oplus \mathbf{1}_{(wv^{N+1})^{-2N-4}} \oplus \mathbf{1}_{(wu^{N+1})^{2N+4}} \right) , \qquad (4.19)$$

Thus we have a choice of $(2N+6)$ operators to give vev to. Additionally there are $(2N+6)$ more operators coming from closing punctures with the flipped moment maps vevs $\tilde{h}_i^{-1}\tilde{h}^{\frac{1}{4}}$. In our example summarized in the previous subsection closing with no flip corresponded to the space-dependent vev of $\mathbf{1}_{(wu^{N+1})^{-2N-4}}$ according to $\widetilde{M}_w$ moment maps in (4.18). This results in the operator (4.15). If we consider closing with the flip it would correspond to giving space-dependent vev to the $\mathbf{1}_{(wu^{N+1})^{2N+4}}$ leading to the operator specified in (4.14).

Each of the $(4N + 12)$ total choices leads to a distinct though sometimes related A$\Delta$O. Performing calculations in various cases we can empirically find the expressions covering all possible choices of operators obtained closing punctures with *no flips*:

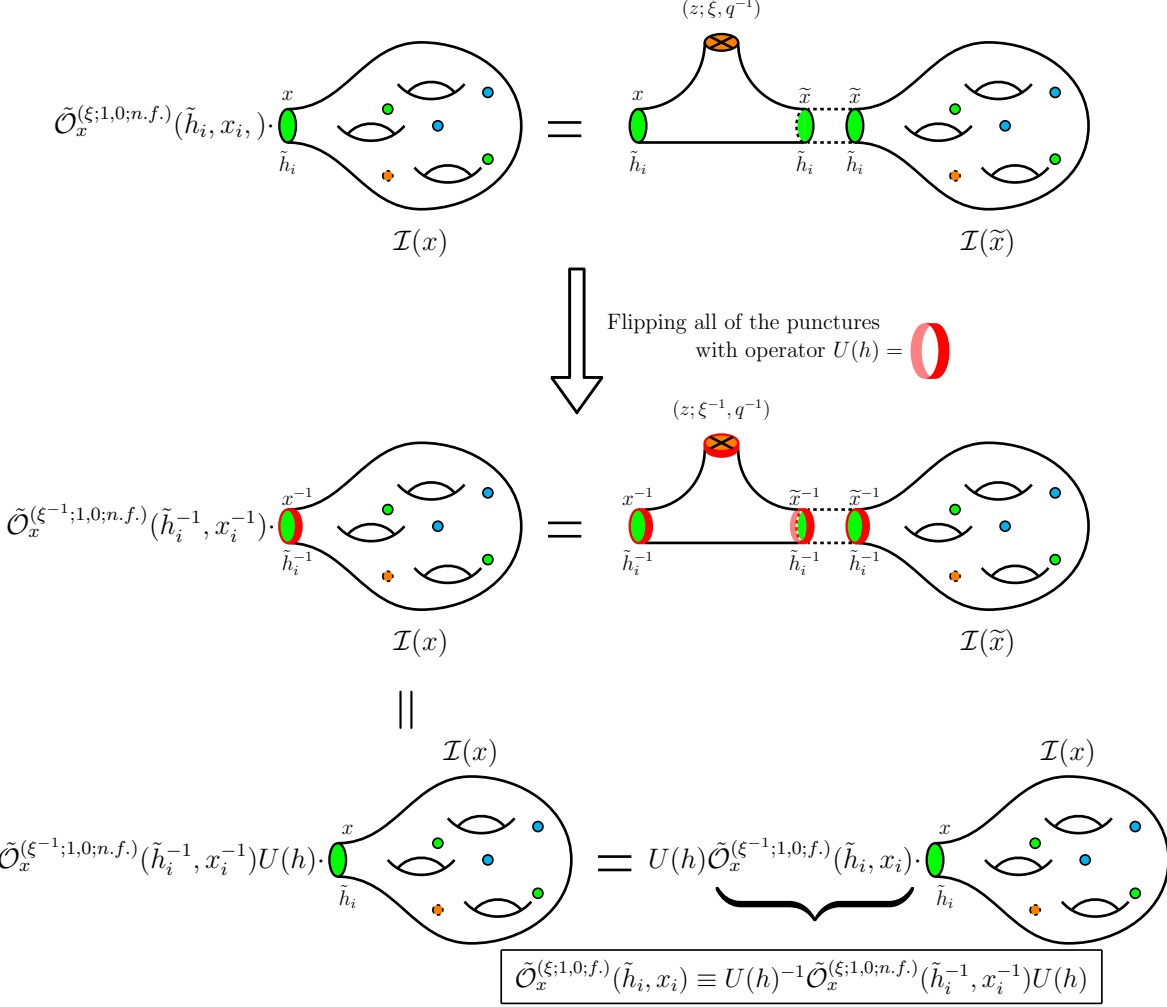

**Figure 13**. Geometric proof of the relation between flip $\widetilde{\mathcal{O}}^{(\xi;1,0;f.)}$ and no flip $\widetilde{\mathcal{O}}^{(\xi;1,0;n.f.)}$ operators. In the first line we present the construction of no-flip operator. Then we proceed with flipping all moment maps of all punctures. Finally in the last line the relation between flip and no flip operators is explained.

$$
\begin{aligned}
\tilde{\mathcal{O}}_{x^{(a)}}^{(\xi_i^{(a)};1,0;n.f.)} \cdot \mathcal{I}(x^{(a)}) \equiv \\
\left( \sum_{l \neq m}^{N+1} \tilde{A}_{lm}^{(\xi_i^{(a)};1,0)}(x^{(a)}) \Delta_{lm}^{(1,0)} + W^{(\xi_i^{(a)};1,0)}\left(x^{(a)}, \tilde{h}^{(a)}\right) \right) \mathcal{I}(x^{(a)}),
\end{aligned} \quad (4.20)
$$

where $\xi_i^{(a)}$ is the charge of the moment map we give space dependent vev to in order to close the minimal puncture. This moment map should be chosen from (4.18) and (4.19) and,

according to what is written above, equals $\tilde{h}_i^{(a)}(\tilde{h}^{(a)})^{-\frac{1}{4}}$. The shift part of the operator above is given by

$$\tilde{A}_{lm}^{(\xi_i^{(a)};1,0)}(x^{(a)}) = \frac{\prod\limits_{j=1}^{2N+6}\theta_p\left((pq)^{\frac{1}{2}}\left(\tilde{h}_j^{(a)}x_l^{(a)}\right)^{-1}\right)}{\theta_p\left(\frac{x_m^{(a)}}{x_l^{(a)}}\right)\theta_p\left(q\frac{x_m^{(a)}}{x_l^{(a)}}\right)} \times$$

$$\prod\limits_{k\neq m\neq l}^{N+1}\frac{\theta_p\left((pq)^{\frac{1}{2}}\xi_i^{(a)}\left(\tilde{h}^{(a)}\right)^{1/4}x_k^{(a)}\right)\theta_p\left((pq)^{\frac{1}{2}}\xi_i^{(a)}\left(\tilde{h}^{(a)}\right)^{-1/4}\left(x_k^{(a)}\right)^{-1}\right)}{\theta_p\left(\frac{x_k^{(a)}}{x_l^{(a)}}\right)\theta_p\left(\frac{x_m^{(a)}}{x_k^{(a)}}\right)} , \quad (4.21)$$

and the constant part is given by:

$$W^{(\xi_i^{(a)};1,0)}(x^{(a)},\tilde{h}^{(a)}) = \frac{\prod\limits_{j\neq i}^{2N+6}\theta_p\left(q^{-1}\left(\xi_i^{(a)}\tilde{h}_j^{(a)}\right)^{-1}\left(\tilde{h}^{(a)}\right)^{\frac{1}{4}}\right)}{\theta_p\left(q^{-2}\left(\xi_i^{(a)}\right)^{-2}\right)} \times$$

$$\prod\limits_{k=1}^{N+1}\frac{\theta_p\left((pq)^{\frac{1}{2}}\xi_i^{(a)}\left(\tilde{h}^{(a)}\right)^{1/4}x_k^{(a)}\right)}{\theta_p\left((pq)^{-\frac{1}{2}}\left(\xi_i^{(a)}\right)^{-1}\left(\tilde{h}^{(a)}\right)^{1/4}q^{-1}x_k^{(a)}\right)} +$$

$$\sum\limits_{m=1}^{N+1}\frac{\prod\limits_{j\neq i}^{2N+6}\theta_p\left((pq)^{\frac{1}{2}}\left(\tilde{h}_j^{(a)}x_m^{(a)}\right)^{-1}\right)}{\theta_p\left((pq)^{\frac{1}{2}}\xi_i^{(a)}\left(\tilde{h}^{(a)}\right)^{-1/4}q\left(x_m^{(a)}\right)^{-1}\right)} \times$$

$$\prod\limits_{k\neq m}^{N+1}\frac{\theta_p\left((pq)^{\frac{1}{2}}\xi_i^{(a)}\left(\tilde{h}^{(a)}\right)^{-1/4}\left(x_k^{(a)}\right)^{-1}\right)\theta_p\left((pq)^{\frac{1}{2}}\xi_i^{(a)}\left(\tilde{h}^{(a)}\right)^{1/4}x_k^{(a)}\right)}{\theta_p\left(q^{-1}\frac{x_m^{(a)}}{x_k^{(a)}}\right)\theta_p\left(\frac{x_k^{(a)}}{x_m^{(a)}}\right)} . \quad (4.22)$$

If we put $a = u$, $x^{(a)} = x$ and $\tilde{h}^{(u)} = \left(u^N w^2\right)^{-2N-4}$ and $\xi_i \equiv \left(\tilde{h}_i^{(u)}\right)^{-1}\left(\tilde{h}^{(u)}\right)^{\frac{1}{4}} = \left(u^{N+1}w\right)^{2N+4}$ using expressions above we reproduce previously obtained operator (4.14), (4.12) and (4.10).

If we close minimal punctures with the flip we obtain similar expressions for $(2N+6)$ operators which appear to be simply related to the no-flip operators summarized above. In particular using simple argument based on gluing constructions (see Figure 13) we can expect these operators to be related by the following conjugation

$$\tilde{\mathcal{O}}_{x^{(a)}}^{(\xi_i^{(a)};1,0;f.)}\left(x^{(a)},\tilde{h}^{(a)}\right) = U_a^{-1}\tilde{\mathcal{O}}_{x^{(a)}}^{(\xi_i^{(a)};1,0;n.f.)}\left((x^{(a)})^{-1},(\tilde{h}^{(a)})^{-1}\right)U_a , \quad (4.23)$$

where $U$ is the conjugation operator given by

$$U_a \equiv \prod\limits_{j=1}^{2N+6}\prod\limits_{i=1}^{N+1}\Gamma_e\left((pq)^{\frac{1}{2}}h_jx_i\right)^{-1} \quad (4.24)$$

Let us discuss the argument of Figure 13. We start with gluing trinion with two maximal punctures of the same type and one minimal puncture. This trinion can be obtained from the four-punctured sphere after closing one of the minimal punctures without introducing defect. Then we flip all of the moment maps in all of the punctures and close minimal puncture with defect introduced. This also corresponds to the substitution $\xi_i^{(a)} \to (\xi_i^{(a)})^{-1}$ and $\tilde{h}_i^{(a)} \to (\tilde{h}_i^{(a)})^{-1}$ inside operator and all indices. Finally we can interpret resulting construction as flipped operator $\widehat{\mathcal{O}}$, i.e. one closed using vev of the flipped moment map with the charge $\xi^{-1}$, on the usual puncture with $\tilde{h}_i^{(a)}$ moment maps charges. As the result we obtain conjugation (4.23).

Applying this conjugation to shift (4.21) and (4.22)

$$
\tilde{\mathcal{O}}_{x^{(a)}}^{(\xi_i^{(a)};1,0;f.)} \cdot \mathcal{I}(x^{(a)}) \equiv
$$
$$
\left( \sum_{l \neq m}^{N+1} \tilde{A}_{lm}^{(\xi_i^{(a)};1,0)}(x^{(a)}) \Delta_{lm}^{(1,0)} + W^{(\xi_i^{(a)};1,0)}\left( (x^{(a)})^{-1}, (\tilde{h}^{(a)})^{-1} \right) \right) \mathcal{I}(x^{(a)}), \quad (4.25)
$$

When we put $N = 1$ in the expressions above reduce to the previously obtained results (3.20) and (3.22). Notice that in the $A_1$ case we were getting 16 operators all of which were equal to the $BC_1$ van Diejen model up to a constant shift. However here the $4N+12$ operators differ more significantly, $e.g.$ also the shift part is different.

## 4.3 Duality relations

We can check some the duality identities introduced in Section 2. We start with the commutation of the operators (2.6). In our case we require all of the $(4N + 12)$ operators (4.20) we derived to commute:

$$
\left[ \tilde{\mathcal{O}}_{x^{(a)}}^{(\xi_i^{(a)};1,0;\alpha)}, \tilde{\mathcal{O}}_{x^{(a)}}^{(\xi_j^{(a)};1,0;\beta)} \right] = \left[ \tilde{\mathcal{O}}_{x^{(a)}}^{(\xi_i^{(a)};0,1;\alpha)}, \tilde{\mathcal{O}}_{x^{(a)}}^{(\xi_j^{(a)};0,1;\beta)} \right] =
$$
$$
\left[ \tilde{\mathcal{O}}_{x^{(a)}}^{(\xi_i^{(a)};1,0;\alpha)}, \tilde{\mathcal{O}}_{x^{(a)}}^{(\xi_j^{(a)};0,1;\beta)} \right] = 0, \quad \forall\, i,\, j,\, a \text{ and } \alpha, \beta = f. \text{ or } n.f., \quad (4.26)
$$

In $A_1$ case all of the operators acting on certain puncture were actually the same van Diejen operator up to a constant shift and their commutativity trivially followed from this fact. This is not the case here since now we have $(4N + 12)$ distinct operators and the proof of their commutation relations is involved. Though we don't have strict proof of it in the Appendix E.2 we give strong evidence in its favor. In particular we break the full commutators (4.26) into parts according to how these parts shift the trial functions. Then we show that each part is zero. However to in our arguments we rely on $q$ and $p$ expansions instead of a strict proof.

Finally we can obtain kernel functions of the operators (4.20). By our construction we know that superconformal index of any $\mathcal{N} = 1$ theory obtained in the compactification is a

kernel function of this finite-difference operators. Simplest examples are trinion index (4.1) and index of the tube theory we will specify below.

For the $A_N$ trinion index (4.1) kernel function property (2.7) reads

$$\tilde{\mathcal{O}}_x^{(\xi_i^{(u)};1,0;\alpha)} \cdot \tilde{K}_3^{A_N}(x,y,z) = \tilde{\mathcal{O}}_y^{(\xi_i^{(v)};1,0;\beta)} \cdot \tilde{K}_3^{A_N}(x,y,z), \qquad \forall\, i, \tag{4.27}$$

where

$$\tilde{K}_3^{A_N}(x,y,z) \equiv \prod_{i=1}^{N+1} \Gamma_e\left((pq)^{\frac{1}{2}}\left(u^N w^2\right)^{-2N-4} x_i\right) \Gamma_e\left((pq)^{\frac{1}{2}}\left(v^N w^2\right)^{-2N-4} y_i\right) K_3^{A_N}(x,y,z) \tag{4.28}$$

is the conjugation of the kernel functions required to accompany corresponding conjugation (4.11) of $\mathcal{O}$ operators. Notice that the moment map we give vev to should be equal on both sides of the kernel equality. But on different sides of equality it can correspond to different types of flips and hence we should use different indices $\alpha$ and $\beta$ which can be equal or not. Just like in $A_1$ case the proof of this kernel identity is technically complicated.

However instead we can follow the lines of Section 3 and study the kernel function properties of tube theories. If we close the minimal puncture of the trinion theory giving vev to one of the baryonic maps in (4.5) we obtain Wess-Zumino tube theory $\mathcal{T}_{(2;0)}^{A_N}$. For a particular example we will consider here the case of giving constant vev to the $\mathbf{1}_{(wu^{N+1})^{2N+4}}$ baryon. In this case we do not introduce any defect into the theory. As usually this corresponds to capturing the residue of the index (4.1) at $z = (pq)^{-\frac{1}{2}}\left(wu^{(N+1)}\right)^{-(2N+4)}$ that emerges due to the contour pinching. It can be easily seen from the positions of the poles in the integrand of (4.1). One of the possible combinations of the poles giving identical contributions is:

$$t_i = (pq)^{-\frac{1}{2(N+2)}} u^{-2N-4} x_i^{-1} q^{-k_i} p^{-m_i}, \quad i = 1,\dots,N+1,$$
$$t_{N+2} = (pq)^{-\frac{1}{2(N+2)}} w^{-2N-4} z^{-1} q^{-k_{N+2}} p^{-m_{N+2}}. \tag{4.29}$$

Using $\mathrm{SU}(N+2)$ constraint we can rewrite poles of the first line in the expression above as poles in $t_{N+2}$:

$$t_{N+2} = \prod_{i=1}^{N+1} t_i^{-1} = (pq)^{\frac{N+1}{2(N+2)}} u^{2(N+1)(N+2)} q^{\sum_{i=1}^{N+1} k_i} p^{\sum_{i=1}^{N+1} m_i}, \tag{4.30}$$

The line of poles in (4.29) are outside the integration contour while (4.30) are inside. When $z = (pq)^{-\frac{1}{2}}\left(wu^{(N+1)}\right)^{-(2N+4)} q^{-\sum_{i=1}^{N+2} k_i} p^{-\sum_{i=1}^{N+2} m_i}$ these two lines of poles collide and contour gets pinched. Choosing $k_i = m_i = 0 \; \forall\, i$ we get the required pinching. Substituting these values into the integrand (4.1) we obtain the index of the tube theory,

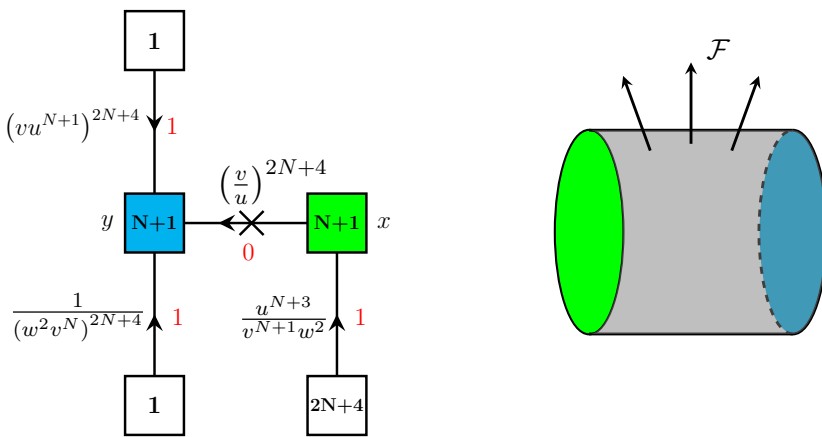

**Figure 14**. Quiver description of the $A_N$ tube theory.

$$
\tilde{K}^{A_N}_{(2,0)}(x,y) = \Gamma_e\left(pqw^{4N+8}u^{2(N+1)(2N+4)}\right)\prod_{l=1}^{2N+4}\Gamma_e\left(u^{-(N+1)(2N+5)}v^{-N-1}w^{-2}a_l\right)\times
$$
$$
\prod_{i,j=1}^{N+1}\Gamma_e\left(\left(u^{-1}v\right)^{2N+4}y_jx_i^{-1}\right)\prod_{i=1}^{N+1}\Gamma_e\left((pq)^{\frac{1}{2}}w^{-4N-8}v^{-N(2N+4)}y_i\right)\times
$$
$$
\Gamma_e\left((pq)^{\frac{1}{2}}v^{2N+4}u^{(N+1)(2N+4)}y_i\right)\prod_{l=1}^{2N+4}\Gamma_e\left((pq)^{\frac{1}{2}}u^{N+3}v^{-N-1}w^{-2}x_ia_l\right)\,, \quad(4.31)
$$

where we have already performed the conjugation of (4.28). Theory described by this index is shown on the Figure 14. The kernel property (2.7) in this case reads

$$
\tilde{\mathcal{O}}^{(\xi_i;1,0;\alpha)}_x \cdot \tilde{K}^{A_N}_{(2,0)}(x,y) = \tilde{\mathcal{O}}^{(\xi_i;1,0;\beta)}_y \cdot \tilde{K}^{A_N}_{(2,0)}(x,y)\,, \qquad \forall\, i\,, \quad \alpha,\beta = f.\ \text{or}\ n.f. \quad (4.32)
$$

Here on both sides of the equation shift operators are obtained giving vev to the same moment map in $M_w$. Notice that it can happen that the operator should be considered as flipped on one side and not flipped on the other side of the equation. For example let's assume we give vev to $\mathbf{1}_{(wu^{N+1})^{2N+4}}$ baryon. When we act on $\mathrm{SU}(N+1)_x$ puncture this baryon should be considered as flipped according to the moment maps summarized in (4.18). So on the lhs of (4.32) we should use expressions (4.23) for the operator. At the same time when we act on $\mathrm{SU}(N+1)_y$ puncture this baryon is not considered as flipped according to (4.19) and hence we should use expressions (4.20), (4.21) and (4.22) for the corresponding operator.

In $A_1$ case we provided the proof of the kernel property (4.32). However for $N > 1$ we do not attempt to do so analytically. Instead we can rely on $q$ and $p$ expansions for the lower rank cases. We summarize this analysis of kernel identity for one particular choice of operators and tubes in Appendix E.3.

## 5 Discussion

Let us briefly summarize and discuss our results. In this paper we have first utilized a variety of explicit 4d descriptions of compactifications of the rank one E-string theory on surfaces to derive an integrable model corresponding to the E-string theory. As was previously derived using yet another description in [25], this model turns out to be the $BC_1$ van Diejen system.[15] In all the different derivation we obtain the same model (up to constant shifts). This is a non trivial check of the dictionary between 4d theories and 6d compactifications as one can think of the integrable models as being associated *locally* on the surface to punctures, while the difference in derivation has to do with how we define the compactification *globally* on the complete surface. The indices of various compactifications are expected to be Kernel functions of the $BC_1$ van Diejen model providing a number of mathematically precise conjectures. It will be very interesting to study these conjectures.

Our second main result is the explicit derivation of a generalization of one of the routes to the $BC_1$ van Diejen model. We have considered the rank one E-string as a first item in a sequence of 6d SCFTs, the minimal $(D_{N+3}, D_{N+3})$ conformal matter theories ($N \geq 1$). Upon compactification to 5d these models have (at least) three different effective gauge theory descriptions. This leads to three different types of maximal punctures one can define on a Riemann surface when compactifying to 4d. We have utilized one of these descriptions, the one with $SU(N+1)$ gauge group, and the associated compactifications to 4d derived in [34], to obtain an $A_N$ generalization of the $BC_1 \sim A_1$ van Diejen model. The derivation leads to a set of commuting analytic difference operators and the indices of the compactifications of the minimal $(D_{N+3}, D_{N+3})$ conformal matter theories are expected to be Kernel functions for these operators.

There are several ways in which our results can be extended. First, one can utilize the other two 5d descriptions of the minimal $(D_{N+3}, D_{N+3})$ conformal matter theories to derive analytic difference operators associated to $C_N$ and $(A_1)^N$ root systems. The relevant three punctured sphere for the former is defined in Appendix B.5 Figure 21, while for the latter it was obtained in [38] (see Figure 9 there). Using the general procedure presented in [2] and discussed here in Section 2 given these 4d theories the relevant operators are thus implicitly defined. However, it would be very interesting to derive them explicitly and study their properties. For example, the index of the WZ model of Figure 17 should be a Kernel function of the $A_N$ operators derived here and the putative $C_N$ operatos. Also gluing together $N$ three punctured spheres (with two $(A_1)^N$ maximal punctures and one $SU(2)$ minimal puncture) of [38] one obtains a three punctured sphere with two $(A_1)^N$ maximal punctures and one $C_N$ maximal puncture: the index of this model thus is expected to be a Kernel function for the putative $(A_1)^N$ and $C_N$ operators. Thus these generalizations and the mathematical properties they are expected to satisfy can give us interesting checks of the various relations between physics in 6d, 5d, and 4d.

---

[15]See also [30] for a higher dimensional derivation using SW curves.

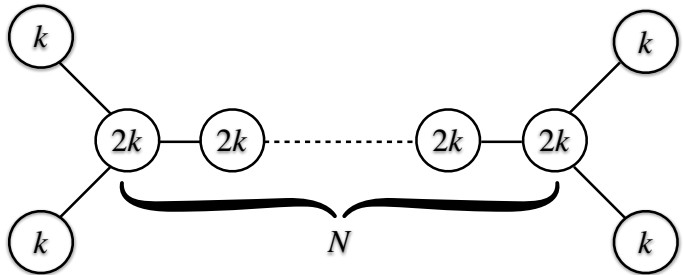

**Figure 15**. Upon a circle compactification of the non minimal $(D, D)$ conformal matter, $k$ M5 branes probing $D_{N+3}$ singularity, with proper holonomies turned on one obtains a 5d effective quiver description shaped as the affine Dynkin diagram of $D_{N+3}$ with $SU(k)$ and $SU(2k)$ gauge groups as depicted here. Upon compactification to four dimensions this five dimensional description gives a puncture with symmetry $SU(k)^2 \times SU(2k)^N \times SU(k)^2$ with a pattern of moment map operators corresponding to the Dynkin diagram.

One can also consider the *non-minimal* $(D_{N+3}, D_{N+3})$ conformal matter theories: these are 6d models obtained on a collection of $k$ M5 branes probing $D_{N+3}$ singularity [72]. A generalization of the effective 5d description with $(A_1)^N$ gauge group is known and it takes form of $(A_{k-1})^2 \times (A_{2k-1})^N \times (A_{k-1})^2$ quiver gauge theory. This quiver gauge theory has a shape of the affine Dynkin diagram of $D_{N+3}$, see Figure 15. The relevant 4d three punctured spheres with two maximal punctures of this type and one minimal $U(1)$ puncture are also known [45]. Thus one can apply the procedure of [2] directly in this case also. The models will depend again on $2N + 6$ parameters, however for $k > 1$ these should be thought to be associated with $D_{N+3} \times D_{N+3}$ instead of $D_{2N+6}$ of $k = 1$.

It is also interesting to understand whether the various models associated to a given sequence of 6d theories have any interesting relations to each other. For example in the case of $k$ M5 branes probing an $A_{N-1}$ singularity, the $(A_{N-1}, A_{N-1})$ conformal matter theories, the various integrable models [39, 73, 74] associated to different values of $N$ were related to same set of transfer matrices [73, 75]. It would be interesting to understand whether any relations of this sort exist also for the $D$ series. Moreover, although three punctured spheres are not know at the moment for the $E$ series of conformal matter models, it would be interesting to try and find a uniform definition of the integrable models which would be applicable to the full ADE series of the conformal matter theories.[16]

As it was already mentioned in our discussion in certain cases there are more than one effective 5d gauge theory descriptions which eventually lead through the procedure applied in this paper, to a number of tightly related integrable models which however might act on different types of parameters. These models will have for example joint Kernel functions. The $A_N$, $C_N$, and $(A_1)^N$ models are examples of this. Such effects go beyond the D-type conformal matter theories. For example in the case of A-type conformal theories in addition to a puncture with $(A_{k-1})^N$ group[17] there are 5d descriptions discussed in [76]. For $k = 2$

---

[16]The two punctured spheres are known for all the ADE series and these do take a rather unified form [71].

[17]This is to be associated to a circular quiver of $N$ $A_{k-1}$ groups, the affine quiver of $A_{N-1}$.

the associated maximal punctures and three punctured spheres were discussed in [38]. It will be very interesting to derive integrable models associated to these theories and study their relations to the ones of [39, 73, 74].

The rank one E-string theory has yet another natural generalization to a rank $Q$ E-string SCFT: $Q$ M5 branes probing the end of the world brane. A corresponding effective 5d description is given in terms of a $USp(2Q)$ gauge theory. A natural generalization of the relation between the $BC_1$ van Diejen model and the rank one case is the $BC_Q$ model associated with the rank $Q$ case. For $Q > 1$ such models have nine parameters which fits the number of the Cartan generators of the $E_8 \times SU(2)$ symmetry group of the corresponding 6d SCFTs. Although here the three punctured spheres are not known, and thus the procedure of [2] cannot be directly applied, the two punctured spheres are known [27]. The indices of these thus are expected to be Kernel functions for the $BC_Q$ van Diejen model. The relevant indices are directly related to the interpolation kernels derived by Rains in [28, 29]. The issue of having more than one 5d description is also applicable here. There is at least one additional effective 5d description for the rank $Q$ E-string theory [77]: $SU(Q + 1)$ with level $\pm \frac{Q-1}{2}$ CS term, eight fundamental and one antisymmetric hypermultiplet. It would be interesting to understand whether thus there is an $A_Q$ relative of the $BC_Q$ van Diejen model in the sense discussed above (e.g. joint Kernel functions). Finally, the $BC_Q$ van Diejen models are related to the $D$ type RS models by specialization of parameters (see Appendix D for a brief review of one facet of this relation.). Thus there is a natural question whether the indices (and not only) of the corersponding 4d theories, compactifications of rank $Q$ E-string and class $\mathcal{S}$ (D-type class $\mathcal{S}$ or A-type class $\mathcal{S}$ with twisted punctures, see for example [8, 78, 79]) have any interesting relations.

Another venue for a search for a systematic understanding of the relation between the integrable models and 6d SCFTs is to consider the non-higgsable cluster theories [72]. These are in a sense minimal theories in 6d. For some of them we know what the integrable models are: $A_1$ RS for the $A_1$ $(2, 0)$ SCFT, $BC_1$ van Diejen for the E-string, the models derived in [22] (and further disucssed in [23]) for the minimal $SU(3)$ and $SO(8)$ SCFTs. However, we lack any understanding for other models in the sequence (the minimal $F_4$, $E_6$, $E_7$, $E_{7\frac{1}{2}}$, and $E_8$ SCFTs [19, 20]). It would be very interesting to understand the full sequence in detail. In more generality, there is a vigorous effort in recent years to classify and systematize our understanding of 6d and 5d SCFTs in recent years (see for a snapshot of examples [6, 7, 48, 72, 80, 81]), and it will be very interesting to understand whether association of integrable systems to these models can on one hand help with this classification and on the other hand whether novel interesting integrable systems can be found by utilizing this classification.

Finally it would also be interesting to better understand the relation between our results and various manifestations of BPS/CFT correspondence [82]. One of such manifestations are $q$- and elliptic Virasoro constraints [83–87] for various partition functions of supersymmetric gauge theories. Particular example interesting for us are elliptic Virasoro constraints for $\mathcal{N} = 1$ superconformal indices [88] with Wilson loop insertions. Usual Virasoro constrains are known to be related to the Calogero-Sutherland integrable model. It would be interesting

to reveal connections between our A$\Delta$Os and these elliptic Virasoro constraints.

## Acknowledgments

We would like to thank E. Rains for useful correspondence. This research is supported in part by Israel Science Foundation under grant no. 2289/18, by I-CORE Program of the Planning and Budgeting Committee, by a Grant No. I-1515-303./2019 from the GIF, the German-Israeli Foundation for Scientific Research and Development, and by BSF grant no. 2018204. The research of SSR is also supported by the IBM Einstein fellowship of the Institute of Advanced Study and by the Ambrose Monell Foundation.

## A  Special functions

We summarize here some definitions and properties of special functions used in the paper.

Elliptic Gamma function is defined through the following infinite product:

$$\Gamma_e (z) \equiv \prod_{k,m=0}^{\infty} \frac{1 - p^{k+1} q^{m+1}/z}{1 - p^k q^m z} \, . \tag{A.1}$$

It can be easily seen that the poles of this function are located at the following values of the argument:

$$z = p^{-k} q^{-m} \, , \qquad k, m \in \mathbb{Z}_{\geq 0} \, . \tag{A.2}$$

The following relation will be useful in our calculations:

$$\Gamma_e \left( \frac{pq}{z} \right) \Gamma_e (z) = 1 \, . \tag{A.3}$$

Also we will often deal with the elliptic beta integral formula

$$\kappa \oint \frac{dz}{4\pi i z} \frac{1}{\Gamma_e (z^{\pm 2})} \prod_{j=1}^{6} \Gamma_e \left( t_i z^{\pm 1} \right) = \prod_{i<j} \Gamma_e (t_i t_j) \, . \tag{A.4}$$

Here $\kappa$ is defined to be

$$\kappa = (q;q)(p;p) = \prod_{\ell=0}^{\infty} (1 - q^{1+\ell})(1 - p^{1+\ell}). \tag{A.5}$$

$A_N$ generalization of this formula is

$$\frac{\kappa^N}{N!} \oint \prod_{i=1}^{N} \frac{dz_i}{2\pi i z_i} \prod_{i \neq j}^{N+1} \Gamma_e \left( \frac{z_i}{z_j} \right)^{-1} \prod_{i=1}^{N+2} \prod_{j=1}^{N+1} \Gamma_e (s_i z_j) \, \Gamma_e \left( t_i z_j^{-1} \right) =$$

$$\prod_{i=1}^{N+2} \Gamma_e \left( S s_i^{-1} \right) \Gamma_e \left( T t_i^{-1} \right) \prod_{i,j=1}^{N+2} \Gamma_e (s_i t_j) \, , \qquad \left( T = \prod_{i=1}^{N+2} t_i \, , \; S = \prod_{i=1}^{N+2} s_i \right) . \tag{A.6}$$

The Theta function is defined as follows:

$$\theta_p\left(x\right) \equiv \left(x;p\right)_\infty \left(x^{-1}p;p\right)_\infty,\tag{A.7}$$

where $\left(z;p\right)_\infty$ is the usual q-Pochhammer symbol defined as follows:

$$\left(x;p\right)_\infty = \prod_{k=0}^{\infty}\left(1-xp^k\right).\tag{A.8}$$

Following properties of theta function will be useful to us

$$\theta_p\left(x\right) = \frac{\Gamma_e\left(qx\right)}{\Gamma_e\left(x\right)},\quad \theta_p\left(x^{-1}\right) = -x^{-1}\theta_p\left(x\right),\quad \theta_p\left(xp^m\right) = \left(-1\right)^m x^{-m}p^{-\frac{1}{2}m\left(m-1\right)}\theta_p\left(x\right).\tag{A.9}$$

We will also use the following duality identity from [89]:

$$V(\underline{t}) = \prod_{1\leq j<k\leq 4}\Gamma_e\left(t_j t_k\right)\Gamma_e\left(t_{j+4}t_{k+4}\right)V(\underline{s}),\tag{A.10}$$

where

$$V(\underline{t}) \equiv \kappa \oint \frac{dz}{2\pi i z}\frac{\prod_{j=1}^{8}\Gamma_e\left(t_j z^{\pm 1}\right)}{\Gamma_e\left(z^{\pm 2}\right)},\qquad \prod_{j=1}^{8}t_i = pq,\quad |t_j|,|s_j|<1.$$

$$s_j = \rho^{-1}t_j,\quad j=1,2,3,4;\quad s_j = \rho t_j,\quad j=5,6,7,8;\quad \rho \equiv \sqrt{\frac{t_1 t_2 t_3 t_4}{pq}}\tag{A.11}$$

## B  Derivations of operators

In this Appendix we will present derivations of AΔOs mentioned in this paper. In Appendix B.1 we derive the theory corresponding to the compactification of minimal (D,D) conformal matter theory on the four punctured sphere. In Appendices B.2 and B.3 we derive two examples of $A_N$ generalizations of van Diejen operator. Finally in Appendices B.4 and B.6 we discuss derivation of the $A_1^1$ and $C_1$ AΔOs both of which turn out to be van Diejen operators.

### B.1  Derivation of $A_N$ four-punctured sphere.

In this appendix we derive the index of the $4d$ theory corresponding to the compactification of the D-type conformal matter theory on the four-punctured sphere. For this purpose we

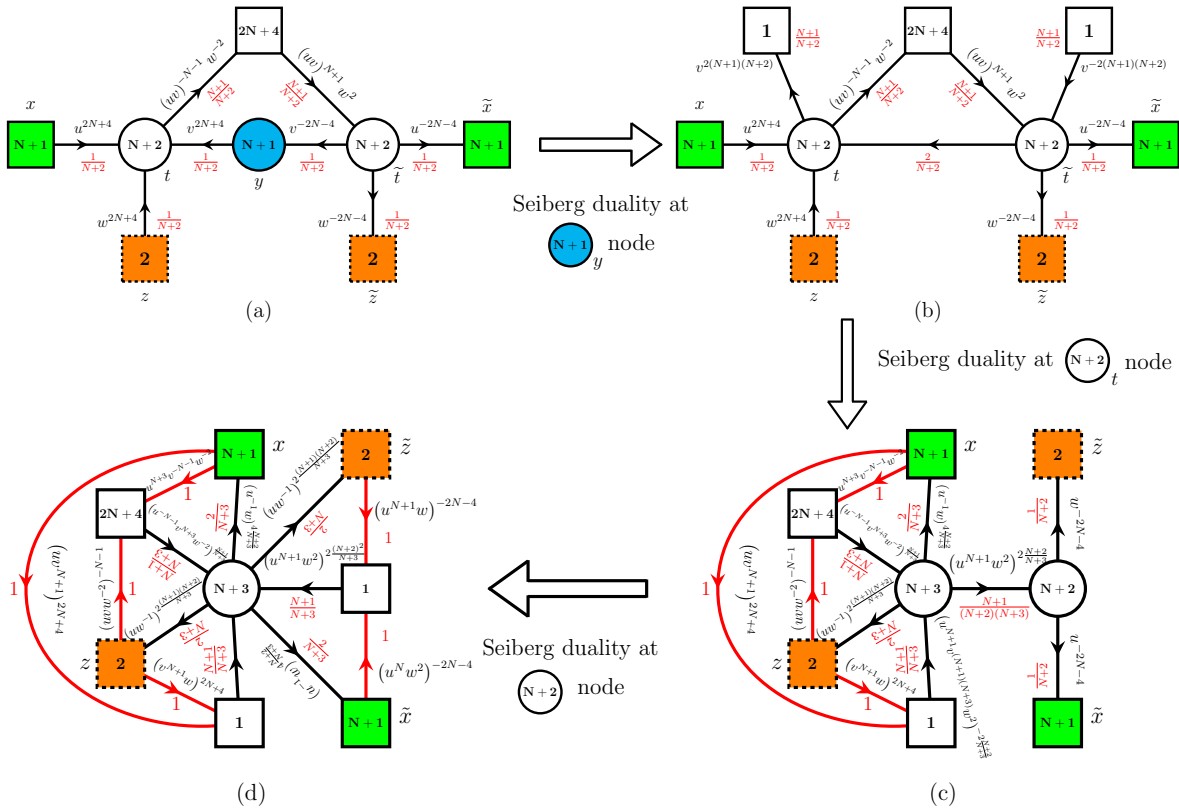

**Figure 16**. Chain of duality transformations of the four-punctured sphere theory. As a starting point we use two S glued $A_N$ trinions. Red lines correspond to the conttributions of theflip singlets.

start with the trinion index (4.1) and S glue it to a conjugated trinion. This leads to

$$
K_4^{A_N}(x, \tilde{x}, z, \tilde{z}) = \kappa_{N+1}^2 \kappa_N \oint \prod_{i=1}^{N+1} \frac{dt_i}{2\pi i t_i} \frac{d\tilde{t}_i}{2\pi i \tilde{t}_i} \prod_{j=1}^{N} \frac{dy_j}{2\pi i y_j} \prod_{i \neq j}^{N+2} \frac{1}{\Gamma_e\left(\frac{t_i}{t_j}\right) \Gamma_e\left(\frac{\tilde{t}_i}{\tilde{t}_j}\right)} \times
$$

$$
\prod_{i \neq j}^{N+1} \frac{1}{\Gamma_e\left(\frac{y_i}{y_j}\right)} \prod_{i=1}^{N+2} \prod_{j=1}^{N+1} \prod_{l=1}^{2N+4} \Gamma_e\left((pq)^{\frac{1}{2N+4}} u^{-2N-4} \tilde{t}_i^{-1} \tilde{x}_j^{-1}\right) \Gamma_e\left((pq)^{\frac{1}{2N+4}} w^{2N+4} t_i z^{\pm 1}\right) \times
$$

$$
\Gamma_e\left((pq)^{\frac{1}{2(N+2)}} u^{2N+4} t_i x_j\right) \Gamma_e\left((pq)^{\frac{1}{2N+4}} v^{2N+4} t_i y_j\right) \times
$$

$$
\Gamma_e\left((pq)^{\frac{1}{2N+4}} v^{-2N-4} \tilde{t}_i^{-1} y_j^{-1}\right) \Gamma_e\left((pq)^{\frac{1}{2N+4}} w^{-2N-4} \tilde{t}_i^{-1} \tilde{z}^{\pm 1}\right) \times
$$

$$
\Gamma_e\left((pq)^{\frac{N+1}{2N+4}} (uv)^{-N-1} w^{-2} t_i^{-1} a_l\right) \Gamma_e\left((pq)^{\frac{N+1}{2N+4}} (uv)^{N+1} w^2 \tilde{t}_i a_l^{-1}\right) . \quad \text{(B.1)}
$$

Now starting from this expression we perform a chain of Seiberg dualities which is shown in the Figure 16. As the first step to simplify this expression we can notice that $SU(N+1)_y$ node of this theory is S-confining so we can integrate $y$ out using $A_N$ elliptic $\beta$-integral formula

$$K_4^{A_N}(x,\tilde{x},z,\tilde{z}) = \kappa_{N+1}^2 \oint \prod_{i=1}^{N+1} \frac{dt_i}{2\pi i t_i} \frac{d\tilde{t}_i}{2\pi i \tilde{t}_i} \prod_{i\neq j}^{N+2} \frac{1}{\Gamma_e\left(\frac{t_i}{t_j}\right)\Gamma_e\left(\frac{\tilde{t}_i}{\tilde{t}_j}\right)} \times$$

$$\prod_{i=1}^{N+2}\prod_{j=1}^{N+1}\prod_{l=1}^{2N+4} \Gamma_e\left((pq)^{\frac{1}{2N+4}}u^{-2N-4}\tilde{t}_i^{-1}\tilde{x}_j^{-1}\right)\Gamma_e\left((pq)^{\frac{1}{2N+4}}w^{2N+4}t_iz^{\pm1}\right) \times$$

$$\Gamma_e\left((pq)^{\frac{1}{2N+4}}w^{-2N-4}\tilde{t}_i^{-1}\tilde{z}^{\pm1}\right)\Gamma_e\left((pq)^{\frac{N+1}{2N+4}}(uv)^{-N-1}w^{-2}t_i^{-1}a_l\right) \times$$

$$\Gamma_e\left((pq)^{\frac{1}{N+2}}t_i\tilde{t}_j^{-1}\right)\Gamma_e\left((pq)^{\frac{N+1}{2N+4}}(uv)^{N+1}w^2\tilde{t}_ia_l^{-1}\right)\Gamma_e\left((pq)^{\frac{1}{2(N+2)}}u^{2N+4}t_ix_j\right) \times$$

$$\Gamma_e\left((pq)^{\frac{N+1}{2N+4}}v^{2(N+1)(N+2)}t_i^{-1}\right)\Gamma_e\left((pq)^{\frac{N+1}{2N+4}}v^{-2(N+1)(N+2)}\tilde{t}_i\right) \quad \text{(B.2)}$$

Now we can perform Seiberg duality on the $\mathrm{SU}(N+2)_t$ node which leads to the following index:

$$K_4^{A_N}(x,\tilde{x},z,\tilde{z}) = \kappa_{N+1}\kappa_{N+2} \oint \prod_{i=1}^{N+2}\frac{dt_i}{2\pi i t_i}\prod_{i=1}^{N+1}\frac{d\tilde{t}_i}{2\pi i \tilde{t}_i}\prod_{i\neq j}^{N+3}\frac{1}{\Gamma_e\left(\frac{t_i}{t_j}\right)}\prod_{i\neq j}^{N+2}\frac{1}{\Gamma_e\left(\frac{\tilde{t}_i}{\tilde{t}_j}\right)} \times$$

$$\prod_{i=1}^{N+3}\prod_{j=1}^{N+2}\prod_{k=1}^{N+1}\prod_{l=1}^{2N+4}\Gamma_e\left((pq)^{\frac{1}{2N+4}}u^{-2N-4}\tilde{t}_j^{-1}\tilde{x}_k^{-1}\right)\Gamma_e\left((pq)^{\frac{1}{2N+4}}w^{-2N-4}\tilde{t}_j^{-1}\tilde{z}^{\pm1}\right) \times$$

$$\Gamma_e\left((pq)^{\frac{N+1}{2(N+2)(N+3)}}u^{2\frac{(N+1)(N+2)}{N+3}}w^{4\frac{N+2}{N+3}}\tilde{t}_jt_i\right)\Gamma_e\left((pq)^{\frac{1}{N+3}}(u^{-1}w)^{4\frac{N+2}{N+3}}x_k^{-1}t_i\right) \times$$

$$\Gamma_e\left((pq)^{\frac{1}{N+3}}(uw^{-1})^{2\frac{(N+2)(N+1)}{N+3}}z^{\pm1}t_i\right)\Gamma_e\left((pq)^{\frac{N+1}{2(N+3)}}u^{-\frac{(N+1)^2}{N+3}}v^{N+1}w^{-2\frac{N+1}{N+3}}a_l^{-1}t_i^{-1}\right) \times$$

$$\Gamma_e\left((pq)^{\frac{1}{2}}v^{2(N+1)(N+2)}w^{2N+4}z^{\pm1}\right)\Gamma_e\left((pq)^{\frac{1}{2}}u^{N+3}v^{-N-1}w^{-2}a_lx_k\right) \times$$

$$\Gamma_e\left((pq)^{\frac{1}{2}}u^{2N+4}v^{2(N+1)(N+2)}x_k\right)\Gamma_e\left((pq)^{\frac{1}{2}}(uv)^{-N-1}w^{2N+2}a_lz^{\pm1}\right) \times$$

$$\Gamma_e\left((pq)^{\frac{N+1}{2(N+3)}}u^{-2\frac{(N+1)(N+2)}{N+3}}v^{-2(N+1)(N+2)}w^{-4\frac{N+2}{N+3}}t_i^{-1}\right).\text{(B.3)}$$

In this index $\tilde{t}$-integration corresponds to the $\mathrm{SU}(N+2)$ gauge theory with $(N+3)$ flavors which is S-confining so we can integrate it out using $A_N$ elliptic $\beta$-integral formula (A.6).

This will lead to the following integral,

$$K_4(x, \tilde{x}, z, \tilde{z}) = \kappa_{N+2} \oint \prod_{i=1}^{N+2} \frac{dt_i}{2\pi i t_i} \prod_{i \neq j}^{N+3} \frac{1}{\Gamma_e\left(\frac{t_i}{t_j}\right)} \times$$

$$\prod_{i=1}^{N+3} \prod_{j=1}^{N+1} \prod_{l=1}^{2N+4} \Gamma_e\left((pq)^{\frac{1}{N+3}} \left(u^{-1}w\right)^{4\frac{N+2}{N+3}} x_j^{-1} t_i\right) \Gamma_e\left((pq)^{\frac{1}{N+3}} \left(uw^{-1}\right)^{2\frac{(N+2)(N+1)}{N+3}} z^{\pm 1} t_i\right)$$

$$\Gamma_e\left((pq)^{\frac{N+1}{2(N+3)}} u^{-\frac{(N+1)^2}{N+3}} v^{N+1} w^{-2\frac{N+1}{N+3}} a_l^{-1} t_i^{-1}\right) \Gamma_e\left((pq)^{\frac{1}{2}} u^{N+3} v^{-N-1} w^{-2} a_l x_j\right) \times$$

$$\Gamma_e\left((pq)^{\frac{N+1}{2(N+3)}} u^{-2\frac{(N+1)(N+2)}{N+3}} v^{-2(N+1)(N+2)} w^{-4\frac{N+2}{N+3}} t_i^{-1}\right) \times$$

$$\Gamma_e\left((pq)^{\frac{1}{2}} u^{2N+4} v^{2(N+1)(N+2)} x_j\right) \Gamma_e\left((pq)^{\frac{1}{2}} \left(uvw^{-2}\right)^{-N-1} a_l z^{\pm 1}\right) \times$$

$$\Gamma_e\left((pq)^{\frac{1}{2}} v^{2(N+1)(N+2)} w^{2N+4} z^{\pm 1}\right) \Gamma_e\left((pq)^{\frac{N+1}{2(N+3)}} u^{2\frac{(N+1)(N+2)^2}{N+3}} w^{4\frac{(N+2)^2}{N+3}} t_i^{-1}\right) \times$$

$$\Gamma_e\left((pq)^{\frac{1}{2}} u^{-2N(N+2)} w^{-4N-8} \tilde{x}_j\right) \Gamma_e\left((pq)^{\frac{1}{2}} u^{-2(N+1)(N+2)} w^{-2N-4} \tilde{z}^{\pm 1}\right) \times$$

$$\Gamma_e\left((pq)^{\frac{1}{N+3}} \left(u^{-1}w\right)^{4\frac{N+2}{N+3}} t_i \tilde{x}_j^{-1}\right) \Gamma_e\left((pq)^{\frac{1}{N+3}} \left(uw^{-1}\right)^{2\frac{(N+1)(N+2)}{N+3}} t_i \tilde{z}^{\pm 1}\right). \quad \text{(B.4)}$$

This expression stands for the index of $SU(N + 3)$ gauge theory with $(2N + 6)$ flavors (and a superptential which can be detemined from the charges of the various fields), and will be used as a starting point for closing punctures in our derivations.

## B.2 Closing minimal puncture with $1_{w^{2N+4} u^{(2N+4)(N+1)}}$ vev.

We will derive next AΔO obtained by closing $SU(2)_z$ minimal puncture of the $A_N$ trinion (3.8) giving space-dependent vev to the baryonic moment map $1_{w^{2N+4} u^{(2N+4)(N+1)}}$.

We start with the index of the four-punctured sphere (B.4) having two punctures, $SU(2)_z$ and $SU(2)_{\tilde{z}}$, that we want to close. First let's compute the residue of the index at $\tilde{z} = (pq)^{-\frac{1}{2}} \left(u^{N+1}w\right)^{2N+4}$ corresponding to closure of $SU(2)_{\tilde{z}}$ without the defect. This residue is simple to calculate because the pole is present in the index explicitly due to the term

$$\Gamma_e\left((pq)^{\frac{1}{2}} u^{-2(N+1)(N+2)} w^{-2N-4} \tilde{z}^{\pm 1}\right).$$

Hence we just need to substitute corresponding value of $\tilde{z}$ into expression. As the result we get $SU(N + 3)$ gauge theory with $(2N + 5)$ flavors. As the second step in the computation we should compute the residue of the resulting expression at $z = (pq)^{-\frac{1}{2}} \left(u^{N+1}w\right)^{-2N-4} q^{-1}$ corresponding to space dependent vev of the baryonic moment map $1_{w^{2N+4} u^{(2N+4)(N+1)}}$. This pole corresponds to certain pinching of the integration contour. Before performing this step it is useful to apply a Seiberg duality transformation resulting in the $SU(N + 2)$ gauge theory

with $(2N + 5)$ flavors and the following superconformal index:

$$K_4^{A_N}(x, \tilde{x}, z) = \kappa_{N+1} \oint \prod_{i=1}^{N+1} \frac{dt_i}{2\pi i t_i} \prod_{i \neq j}^{N+2} \frac{1}{\Gamma_e\left(\frac{t_i}{t_j}\right)} \prod_{i=1}^{N+2} \prod_{j=1}^{N+1} \prod_{l=1}^{2N+4} \Gamma_e\left((pq)^{\frac{1}{2(N+2)}} u^{2N+4} x_j t_i\right) \times$$

$$\Gamma_e\left((pq)^{\frac{1}{2}} u^{-N-3} v^{N+1} w^2 a_l^{-1} \tilde{x}_j^{-1}\right) \Gamma_e\left((pq)^{\frac{1}{2}} \left(uv^{N+1}\right)^{-2N-4} \tilde{x}^{-1}\right) \times$$

$$\Gamma_e\left((pq)^{\frac{1}{2(N+2)}} u^{2N+4} \tilde{x}_j t_i\right) \Gamma_e\left((pq)^{\frac{1}{2(N+2)}} w^{2N+4} z^{\pm 1} t_i\right) \times$$

$$\Gamma_e\left((pq)^{\frac{N+3}{2(N+2)}} u^{-2(N+1)(N+2)} t_i\right) \Gamma_e\left((pq)^{\frac{N+1}{2(N+2)}} (uv)^{-N-1} w^{-2} a_l t_i^{-1}\right) \times$$

$$\Gamma_e\left((pq)^{\frac{N+1}{2(N+2)}} v^{2(N+1)(N+2)} t_i^{-1}\right) \Gamma_e\left((pq)^{\frac{1}{2}} \left(u^N w^2\right)^{-2N-4} \tilde{x}_j\right)$$

$$\tag{B.5}$$

where we omit all overall irrelevant constants. Now the pinching leading to the desired pole in the expression happens at the following points:

$$t_i = (pq)^{-\frac{1}{2(N+2)}} u^{-2(N+2)} x_i^{-1} q^{-k_i}, \quad t_{N+2} = (pq)^{\frac{N+1}{2(N+2)}} u^{2(N+1)(N+2)} q^{1-k_{N+2}},$$

or

$$t_i = (pq)^{-\frac{1}{2(N+2)}} u^{-2(N+2)} \tilde{x}_i^{-1} q^{-k_i}, \quad t_{N+2} = (pq)^{\frac{N+1}{2(N+2)}} u^{2(N+1)(N+2)} q^{1-k_{N+2}}, \quad \text{(B.6)}$$

where $k_i$ are partitions of 1, i.e. $\sum_i k_i = 1$. This is only one particular choice of pinching point. Of course there are $(N + 1)!$ such choices coming from the total number of roots permutations. However contribution of each of them is the same due to the symmetry of the Weyl group. Hence calculation can be done just for the point (B.6) and the result should be multiplied with $(N + 1)!$. Since overall constants are irrelevant for our calculations we ommit the letter coefficient. Let's start with the pinching specified in the first line of (B.6). Computing this contribution we get the following expression for the index of the tube theory

depending on the partition $k = (k_1, k_2, \cdots, k_{N+2})$

$$K^{A_N}_{(4;\vec{k})}(x, \tilde{x}) = \prod_{i=1}^{N+1} \prod_{l=1}^{2N+4} \frac{\Gamma_e\left((pq)^{\frac{1}{2}} u^{2(N+2)^2} q^{1-k_{N+2}} x_i\right)}{\Gamma_e\left((pq)^{\frac{1}{2}} u^{2(N+2)^2} q^{1+k_i-k_{N+2}} x_i\right)} \times$$

$$\frac{\Gamma_e\left((pq)^{-\frac{1}{2}} u^{-2(N+2)^2} x_i^{-1} q^{-1-k_i}\right)}{\Gamma_e\left((pq)^{-\frac{1}{2}} u^{-2(N+2)^2} x_i^{-1} q^{k_{N+2}-1-k_i}\right)} \Gamma_e\left((pq)^{\frac{1}{2}} u^{2(N+2)^2} q^{1-k_{N+2}} \tilde{x}_i\right) \times$$

$$\Gamma_e\left((pq)^{\frac{1}{2}} u^{N+3} v^{-N-1} w^{-2} a_l x_i q^{k_i}\right) \Gamma_e\left((pq)^{\frac{1}{2}} \left(u^N w^2\right)^{2N+4} x_i^{-1} q^{1-k_i}\right) \times$$

$$\Gamma_e\left((pq)^{\frac{1}{2}} u^{-2(N+2)^2} x_i^{-1} q^{-k_i}\right) \Gamma_e\left((pq)^{\frac{1}{2}} \left(uv^{N+1}\right)^{2N+4} x_i q^{k_i}\right) \times$$

$$\Gamma_e\left((pq)^{\frac{1}{2}} \left(uv^{N+1}\right)^{-2N-4} \tilde{x}_i^{-1}\right) \Gamma_e\left((pq)^{\frac{1}{2}} u^{-N-3} v^{N+1} w^2 a_l^{-1} \tilde{x}_i^{-1}\right) \times$$

$$\Gamma_e\left((pq)^{\frac{1}{2}} \left(u^N w^2\right)^{-2N-4} \tilde{x}_i\right) \Gamma_e\left(\frac{\tilde{x}_i}{x_i} q^{-k_i}\right) \prod_{j\neq i}^{N+1} \frac{\Gamma_e\left(\frac{x_j}{x_i} q^{-k_i}\right)}{\Gamma_e\left(\frac{x_j}{x_i} q^{k_j-k_i}\right)} \Gamma_e\left(\frac{\tilde{x}_j}{x_i} q^{-k_i}\right) \times$$

$$\Gamma_e\left(pq^{2-k_{N+2}}\right) \Gamma_e\left(pq^{3-k_{N+2}} \left(u^{N+1} w\right)^{4N+8}\right) \Gamma_e\left(q^{k_{N+2}-1} \left(vu^{-1}\right)^{2(N+1)(N+2)}\right) \times$$

$$\Gamma_e\left(\left(u^{2N+5} v\right)^{-N-1} w^{-2} a_l q^{k_{N+2}-1}\right) . \quad \text{(B.7)}$$

To get the full expression for the tube theory index we should sum expressions above over different partitions such that $\sum_{i=1}^{N+1} k_i = 1$. The final expression for the index should be S glued to the index $\mathcal{I}(\tilde{x})$ of an arbitrary theory with $SU(N+1)_{\tilde{x}}$ maximal puncture:

$$\mathcal{I}_1^{\vec{k}}(x) = \kappa_N \oint \prod_{i=1}^{N} \frac{d\tilde{x}_i}{2\pi i \tilde{x}_i} \prod_{i\neq j}^{N+1} \frac{1}{\Gamma_e\left(\frac{\tilde{x}_i}{\tilde{x}_j}\right)} K^{A_N}_{(4;\vec{k})}(\tilde{x}, x) \mathcal{I}_0(\tilde{x}) , \quad \text{(B.8)}$$

There are two kinds of partitions $\vec{k}$ to consider here.

**I.** First class corresponds to the following partition $\vec{k}_i$

$$k_i = 1 , 1 \leq i \leq N+1 , \quad k_j = 0 \ \forall \ j \neq i . \quad \text{(B.9)}$$

In this case the tube becomes

$$
K^{A_N}_{(4;\vec{k})}(x,\tilde{x}) = \Gamma_e\left(pq^2\right) C_i \prod_{l=1}^{2N+4}\prod_{j\neq i}^{N+1} \theta_p\left(\frac{x_i}{x_j}\right)^{-1} \theta_p\left((pq)^{\frac{1}{2}}u^{2(N+2)^2}qx_i\right) \times
$$

$$
\Gamma_e\left((pq)^{\frac{1}{2}}u^{2(N+2)^2}q\tilde{x}_i\right)\Gamma_e\left((pq)^{\frac{1}{2}}u^{2(N+2)^2}q\tilde{x}_j\right)\Gamma_e\left((pq)^{\frac{1}{2}}u^{N+3}v^{-N-1}w^{-2}a_l x_i q\right) \times
$$

$$
\Gamma_e\left((pq)^{\frac{1}{2}}\left(u^N w^2\right)^{2N+4}x_i^{-1}\right)\Gamma_e\left((pq)^{\frac{1}{2}}u^{-2(N+2)^2}q^{-1}x_i^{-1}\right) \times
$$

$$
\Gamma_e\left((pq)^{\frac{1}{2}}\left(u^N w^2\right)^{2N+4}qx_j^{-1}\right)\Gamma_e\left((pq)^{\frac{1}{2}}u^{-2(N+2)^2}x_j^{-1}\right)\Gamma_e\left((pq)^{\frac{1}{2}}\left(uv^{N+1}\right)^{2N+4}qx_i\right) \times
$$

$$
\Gamma_e\left((pq)^{\frac{1}{2}}u^{-N-3}v^{N+1}w^2 a_l^{-1}\tilde{x}_i^{-1}\right)\Gamma_e\left((pq)^{\frac{1}{2}}\left(uv^{N+1}\right)^{-2N-4}\tilde{x}_i^{-1}\right) \times
$$

$$
\Gamma_e\left((pq)^{\frac{1}{2}}u^{-N-3}v^{N+1}w^2 a_l^{-1}\tilde{x}_j^{-1}\right)\Gamma_e\left((pq)^{\frac{1}{2}}\left(uv^{N+1}\right)^{-2N-4}\tilde{x}_j^{-1}\right) \times
$$

$$
\Gamma_e\left((pq)^{\frac{1}{2}}\left(u^N w^2\right)^{-2N-4}\tilde{x}_i\right)\Gamma_e\left((pq)^{\frac{1}{2}}\left(u^N w^2\right)^{-2N-4}\tilde{x}_j\right)\Gamma_e\left((pq)^{\frac{1}{2}}\left(uv^{N+1}\right)^{2N+4}x_j\right) \times
$$

$$
\Gamma_e\left((pq)^{\frac{1}{2}}u^{N+3}v^{-N-1}w^{-2}a_l x_j\right)\prod_{k\neq i}^{N+1}\prod_{j=1}^{N+1}\Gamma_e\left(\frac{\tilde{x}_j}{x_k}\right)\Gamma_e\left(\frac{\tilde{x}_j}{x_i}q^{-1}\right)\Gamma_e\left(\frac{\tilde{x}_i}{x_k}\right)\Gamma_e\left(\frac{\tilde{x}_i}{x_i}q^{-1}\right),
$$

$$(B.10)$$

where $C_i$ is the following constant:

$$
C_i = \Gamma_e\left(q^{-1}\left(u^{-1}v\right)^{2(N+1)(N+2)}\right)\Gamma_e\left(pq^3\left(u^{N+1}w\right)^{4N+8}\right) \times
$$

$$
\prod_{l=1}^{2N+4}\Gamma_e\left(q^{-1}u^{-(N+1)(2N+5)}v^{-N-1}w^{-2}a_l\right). \qquad (B.11)
$$

Notice that as usually the tube theory has prefactor $\Gamma_e\left(pq^2\right)$ which is equal to 0. Once we perform S gluing (B.8) this zero is canceled by a pole which comes from the $\tilde{x}$ integration contour pinchings.[18] These pinchings happen at

$$
\tilde{x}_i = x_{\sigma(i)}q^{1-k_i}, \quad \tilde{x}_j = x_{\sigma(j)}q^{-k_j}, \quad \sum_{i=1}^{N+1}k_i = 1, \qquad (B.12)
$$

where $\sigma(i)$ are permutations. Due to the symmetry of all expressions w.r.t. $\tilde{x}_i$ permutations we can fix to the choice $\sigma(i) = i$. All other permutations will give equivalent contributions resulting in overall irrelevant factor of $(N+1)!$ which we will omit in any case. Once again we have two classes of partitions $k_i$ of 1.

---

[18]Although we perform the computations in steps, the procedure really is to be understood as first gluing all the parts together and then closing the punctures. In this way we will generate various vanishing and divergent contributions, some of which will cancel out. In the end we will have only poles which correspond to closing the punctures.

**I.a.** First possible partition is $k_i = 1$, $k_j = 0$ $\forall$ $j \neq i$ corresponds to the pinching at $\tilde{x}_j = x_j$ $\forall$ $j$. In this case we obtain

$$\mathcal{I}_1^{(\vec{k}_i;x)}(x) = C_i \frac{\theta_p\left((pq)^{\frac{1}{2}}\left(uv^{N+1}\right)^{2N+4}x_i\right)}{\theta_p\left((pq)^{\frac{1}{2}}u^{2(N+2)^2}qx_i\right)} \prod_{l=1}^{2N+4} \theta_p\left((pq)^{\frac{1}{2}}u^{N+3}v^{-N-1}w^{-2}a_lx_i\right) \times$$

$$\prod_{j\neq i}^{N+1} \frac{\theta_p\left((pq)^{\frac{1}{2}}\left(u^Nw^2\right)^{2N+4}x_j^{-1}\right)\theta_p\left((pq)^{\frac{1}{2}}u^{2(N+2)^2}x_j\right)}{\theta_p\left(\frac{x_i}{x_j}\right)\theta_p\left(q^{-1}\frac{x_j}{x_i}\right)}\mathcal{I}_0(x)\,, \text{(B.13)}$$

**I.b.** Second possible type of partitions is when $k_m = 1$ for some $m \neq i$ and all other $k_j = 0$ including $j = i$. This choice of the partition corresponds to the following pinching:

$$\tilde{x}_i = qx_i\,, \quad \tilde{x}_m = q^{-1}x_m\,, \quad \tilde{x}_j = x_j\,, \ \forall j \neq i \neq m\,. \tag{B.14}$$

Substituting these values into the integrand of (B.8) we obtain the following contribution

$$\mathcal{I}_1^{(\vec{k}_i;qx)}(x) = C_i \sum_{m\neq i}^{N+1} \frac{\theta_p\left((pq)^{\frac{1}{2}}\left(u^Nw^2\right)^{-2N-4}x_i\right)\theta_p\left((pq)^{\frac{1}{2}}\left(uv^{N+1}\right)^{-2N-4}x_m^{-1}\right)}{\theta_p\left(\frac{x_i}{x_m}\right)\theta_p\left(q\frac{x_i}{x_m}\right)} \times$$

$$\prod_{l=1}^{2N+4} \theta_p\left((pq)^{\frac{1}{2}}u^{-N-3}v^{N+1}w^2a_l^{-1}x_m^{-1}\right) \times$$

$$\prod_{j\neq m\neq i}^{N+1} \frac{\theta_p\left((pq)^{\frac{1}{2}}u^{2(N+2)^2}x_j\right)\theta_p\left((pq)^{\frac{1}{2}}\left(u^Nw^2\right)^{2N+4}x_j^{-1}\right)}{\theta_p\left(\frac{x_i}{x_j}\right)\theta_p\left(\frac{x_j}{x_m}\right)}\Delta_{mi}^{(1,0)}\mathcal{I}_0(x)\,, \text{(B.15)}$$

where we have summed over all possible $m$ and $\Delta_{mi}$ is the shift operator defined in (4.8).

**II.** Finally we should consider the second type of partitions in (B.6) corresponding to

$$k_{N+2} = 1\,, \quad k_i = 0\,. \tag{B.16}$$

This partition leads to the following index of the tube theory:

$$K^{A_N}_{(4;k_{N+2})}(x,\tilde{x}) = \Gamma_e\left(pq\right)C_{N+2}\prod_{i=1}^{N+1}\prod_{l=1}^{2N+4}\theta_p\left((pq)^{-\frac{1}{2}}u^{-2(N+2)^2}q^{-1}x_i^{-1}\right)^{-1} \times$$

$$\Gamma_e\left((pq)^{\frac{1}{2}}u^{2(N+2)^2}\tilde{x}_i\right)\Gamma_e\left((pq)^{\frac{1}{2}}u^{N+3}v^{-N-1}w^{-2}a_lx_i\right)\Gamma_e\left((pq)^{\frac{1}{2}}\left(u^Nw^2\right)^{2N+4}qx_i^{-1}\right) \times$$

$$\Gamma_e\left((pq)^{\frac{1}{2}}u^{-2(N+2)^2}x_i^{-1}\right)\Gamma_e\left((pq)^{\frac{1}{2}}\left(uv^{N+1}\right)^{2N+4}x_i\right)\Gamma_e\left((pq)^{\frac{1}{2}}\left(uv^{N+1}\right)^{-2N-4}\tilde{x}_i^{-1}\right) \times$$

$$\Gamma_e\left((pq)^{\frac{1}{2}}u^{-N-3}v^{N+1}w^2a_l^{-1}\tilde{x}_i^{-1}\right)\Gamma_e\left((pq)^{\frac{1}{2}}\left(u^Nw^2\right)^{-2N-4}\tilde{x}_i\right)\prod_{j=1}^{N+1}\Gamma_e\left(\frac{\tilde{x}_j}{x_i}\right)\,,$$

$$\text{(B.17)}$$

where

$$C_{N+2} = \Gamma_e \left( \left( vu^{-1} \right)^{2(N+1)(N+2)} \right) \Gamma_e \left( pq^2 \left( u^{N+1}w \right)^{4N+8} \right) \times$$

$$\prod_{l=1}^{2N+4} \Gamma_e \left( u^{-(N+1)(2N+5)} v^{-N-1} w^{-2} a_l \right) \tag{B.18}$$

Once we glue it to an arbitrary theory as in (B.8) we obtain the pinching at $\tilde{x}_i = x_i \ \forall \ i$ compensating the zero coming from $\Gamma_e(pq)$ prefactor in the tube index. The calculation of the residue results in

$$\mathcal{I}^{k_{N+2}}(x) = C_{N+2} \prod_{j=1}^{N+1} \frac{\theta_p \left( (pq)^{\frac{1}{2}} \left( u^N w^2 \right)^{2N+4} x_j^{-1} \right)}{\theta_p \left( (pq)^{-\frac{1}{2}} u^{-2(N+2)^2} x_j^{-1} q^{-1} \right)} \mathcal{I}_0(x) . \tag{B.19}$$

One should also perform similar computation for the pinching specified in the second line of (B.6). This calculation is identical to the one presented above and leads to exactly the same results. We leave this calculation to the interested reader. Summing terms (B.13), (B.15) and (B.19) we finally obtain the finite difference operator (4.7) with the shift part given in (4.9) and the constant part given in (4.10).

## B.3 Closing minimal puncture with the flipped $1_{w^{2N+4}u^{(2N+4)(N+1)}}$ vev.

Let us derive the A$\Delta$O corresponding to the closure of a minimal punctures by flipping the baryon $\mathbf{1}_{(wu^{N+1})^{2(N+2)}}$ and giving vev to its derivatives.[19] The calculation here is very similar to the calculation with no flip presented in the previous Appendix B.2. However some details of these two calculations differ so we summarize the calculation with the flip below.

We once again start with the index of the four-punctured sphere theory given in (B.4). This time we add contributions of the flip multiplets:

$$K_4^{A_N, flip}(x, \tilde{x}, z, \tilde{z}) \equiv \Gamma_e \left( (pq)^{\frac{1}{2}} \left( u^{N+1}w \right)^{-2N-4} z^{\pm 1} \right) \times$$

$$\Gamma_e \left( (pq)^{\frac{1}{2}} \left( u^{N+1}w \right)^{2N+4} \tilde{z}^{\pm 1} \right) K_4^{A_N}(x, \tilde{x}, z, \tilde{z}) \tag{B.20}$$

Now in order to close two minimal punctures we should compute the residues of the index located at

$$z = (pq)^{-\frac{1}{2}} \left( u^{N+1}w \right)^{2N+4} q^{-1} , \quad \tilde{z} = (pq)^{-\frac{1}{2}} \left( u^{N+1}w \right)^{-2N-4} . \tag{B.21}$$

Like in a previous section we give required weights to $z$ and $\tilde{z}$ one by one. We start with putting $\tilde{z} = (pq)^{-\frac{1}{2}} \left( u^{N+1}w \right)^{-2N-4}$. As we have seen in the previous section without the

---

[19]Usually flipping minimal punctures is not a simple geometric procedure, *e.g.* it is not in class $\mathcal{S}$. However, as the minimal puncture here is obtained by partially closing an USp($2N$) maximal puncture [38], the flipping of the minimal puncture can be thought of as partially flipping the maximal puncture and closing it to the minimal one.

flip the pole at this value of $\tilde{z}$ is explicit. However now with the flip the pole is not explicit anymore and originates from the pinching of one of the integration contours at

$$t_j = (pq)^{\frac{N+1}{2(N+3)}} u^{2 \frac{(N+1)(N+2)^2}{N+3}} w^{4 \frac{(N+2)^2}{N+3}} ,$$ (B.22)

for any choice of $j$. Without loss of generality let us discuss the case of $j = N+3$. All other choices should give the same result so they contribute with an overall factor of $(N+3)$ which is not relevant for our conclusions and will be omitted. Also in order to preserve SU$(N+2)$ constraint we should rescale

$$t_i \to t_i (pq)^{-\frac{N+1}{2(N+2)(N+3)}} u^{-2 \frac{(N+1)(N+2)}{N+3}} w^{-4 \frac{N+2}{N+3}} ,$$ (B.23)

After this we obtain the following expression for the four-punctured sphere with one puncture closed:

$$K_4^{A_N}(x, \tilde{x}, z) = \kappa_{N+1} \oint \prod_{i=1}^{N+1} \frac{dt_i}{2\pi i t_i} \prod_{i \neq j}^{N+2} \frac{1}{\Gamma_e \left( \frac{t_i}{t_j} \right)} \times$$

$$\prod_{i=1}^{N+2} \prod_{j=1}^{N+1} \prod_{l=1}^{2N+4} \Gamma_e \left( (pq)^{\frac{1}{2(N+2)}} u^{-2N-4} x_j^{-1} t_i \right) \Gamma_e \left( (pq)^{\frac{1}{2N+4}} w^{-2N-4} z^{\pm 1} t_i \right) \times$$

$$\Gamma_e \left( (pq)^{\frac{1}{2N+4}} u^{-2N-4} \tilde{x}_j^{-1} t_i \right) \Gamma_e \left( (pq)^{\frac{N+3}{2N+4}} u^{2(N+1)(N+2)} t_i \right) \times$$

$$\Gamma_e \left( (pq)^{\frac{N+1}{2N+4}} (uv)^{N+1} w^2 a_l^{-1} t_i^{-1} \right) \Gamma_e \left( (pq)^{\frac{N+1}{2N+4}} v^{-2(N+1)(N+2)} t_i^{-1} \right)$$

$$\Gamma_e \left( (pq)^{\frac{1}{2}} u^{2N+4} v^{2(N+1)(N+2)} x_j \right) \Gamma_e \left( (pq)^{\frac{1}{2}} u^{2N(N+2)} w^{4N+8} x_j^{-1} \right) \times$$

$$\Gamma_e \left( (pq)^{\frac{1}{2}} (uvw^{-2})^{-N-1} a_l z^{\pm 1} \right) \Gamma_e \left( (pq)^{\frac{1}{2}} (v^{N+1} w)^{2N+4} z^{\pm 1} \right) \times$$

$$\Gamma_e \left( (pq)^{\frac{1}{2}} u^{N+3} v^{-N-1} w^{-2} a_l x_j \right) .$$ (B.24)

Now notice that the pole at $z = (pq)^{-\frac{1}{2}} \left( u^{N+1} w \right)^{2N+4} q^{-1}$ which was explicit before closing SU$(2)_{\tilde{z}}$ puncture is not explicit anymore and originates from the contour pinchings at two points:[20]

$$t_i = (pq)^{-\frac{1}{2N+4}} u^{2N+4} q^{-k_i} x_i , \quad t_{N+2} = (pq)^{\frac{N+1}{2N+4}} u^{-2(N+1)(N+2)} q^{K-k_{N+2}} , \quad \sum_{i=1}^{N+2} k_i = 1 ,$$

or

$$t_i = (pq)^{-\frac{1}{2N+4}} u^{2N+4} q^{-k_i} \tilde{x}_i , \quad t_{N+2} = (pq)^{\frac{N+1}{2N+4}} u^{-2(N+1)(N+2)} q^{K-k_{N+2}} , \quad \sum_{i=1}^{N+2} k_i = 1 ,$$ (B.25)

---

[20]Calculation can be done in another way. Namely we could have first closed SU$(2)_z$ puncture by computing residue of the explicit pole. This calculation would have lead to different but equivalent expression for A$\Delta$O. In particular the constant part $W(x)$ of the operator (4.15) would take different form. For this reason here we choose another approach leading to the result qualitatively similar to the non-flipped result of the previous apppendix.

and all possible permutations that result in an irrelevant overall constant factor. Like in the previous section let's consider pinching specified in the first line. Substituting these expressions into the integrand and computing the residue of the pole we obtain the following expression for the index of the tube theory

$$
\begin{aligned}
K^{A_N}_{(4;\vec{k})}(x,\tilde{x}) &= \Gamma_e\left(pq^{2-k_{N+2}}\right) \prod_{j=1}^{N+1}\prod_{l=1}^{2N+4} \Gamma_e\left((pq)^{\frac{1}{2}}u^{N+3}v^{-N-1}w^{-2}a_l x_j\right) \times \\
&\quad \Gamma_e\left((pq)^{\frac{1}{2}}\left(uv^{N+1}\right)^{2N+4} x_j\right)\Gamma_e\left((pq)^{\frac{1}{2}}\left(u^N w\right)^{2N+4} x_j^{-1}\right) \times \\
&\quad \frac{\Gamma_e\left((pq)^{-\frac{1}{2}}u^{2(N+2)^2}x_j q^{-1-k_j}\right)}{\Gamma_e\left((pq)^{-\frac{1}{2}}u^{2(N+2)^2}x_j q^{-1-k_j+k_{N+2}}\right)} \frac{\Gamma_e\left((pq)^{\frac{1}{2}}u^{-2(N+2)^2}x_j^{-1}q^{1-k_{N+2}}\right)}{\Gamma_e\left((pq)^{\frac{1}{2}}u^{-2(N+2)^2}x_j^{-1}q^{1-k_{N+2}+k_j}\right)} \times \\
&\quad \Gamma_e\left(q^{k_{N+2}-1}\left(uv^{-1}\right)^{2(N+1)(N+2)}\right)\left[\prod_{i\neq j}^{N+1}\frac{\Gamma_e\left(\frac{x_i}{x_j}q^{-k_i}\right)}{\Gamma_e\left(\frac{x_i}{x_j}q^{k_j-k_i}\right)}\right]\prod_{i=1}^{N+1}\Gamma_e\left(x_i\tilde{x}_j^{-1}q^{-k_i}\right) \times \\
&\quad \Gamma_e\left((pq)^{\frac{1}{2}}u^{-2(N+2)^2}\tilde{x}_j^{-1}q^{1-k_{N+2}}\right)\Gamma_e\left((pq)^{\frac{1}{2}}\left(u^N w^2\right)^{-2N-4}x_j q^{1-k_j}\right) \times \\
&\quad \Gamma_e\left((pq)^{\frac{1}{2}}u^{-N-3}v^{N+1}w^2 a_l^{-1}x_j^{-1}q^{k_j}\right)\Gamma_e\left(pq^{3-k_{N+2}}\left(u^{N+1}w\right)^{-4N-8}\right) \times \\
&\quad \Gamma_e\left((pq)^{\frac{1}{2}}\left(uv^{N+1}\right)^{-2N-4}x_j^{-1}q^{k_j}\right)\Gamma_e\left((pq)^{\frac{1}{2}}u^{2(N+2)^2}x_j q^{-k_j}\right), \quad \text{(B.26)}
\end{aligned}
$$

where $\vec{k}$ labels a partitions of 1, *i.e.* it is the vector with only one of the entries equal to one and all others are zero. Now we S glue this tube to an arbitrary theory with $\mathrm{SU}(N+1)_{\tilde{x}}$ global symmetry in a usual way

$$
\mathcal{I}_1^{\vec{k}}(x) = \kappa_N \oint \prod_{i=1}^N \frac{d\tilde{x}_i}{2\pi i \tilde{x}_i}\prod_{i\neq j}^{N+1}\frac{1}{\Gamma_e\left(\frac{\tilde{x}_i}{\tilde{x}_j}\right)}K^{A_N}_{(4;\vec{k})}(\tilde{x},x)\mathcal{I}_0(\tilde{x}), \quad \text{(B.27)}
$$

As usually there are contour pinchings that depend on a particular choice of the partition $\vec{k}$ in the tube (B.26). These pinchings cancel the zero in the index of the tube coming from the term $\Gamma_e\left(pq^{2-k_{N+2}}\right)$. Just like in the previous section there are two distinct classes of partitions contributing.

**I.** First class corresponds to the following partition $\vec{k}_i$

$$
k_i = 1, 1 \leq i \leq N+1, \quad k_j = 0 \ \forall \ j \neq i. \quad \text{(B.28)}
$$

For this partition tube index (B.26) reduces to

$$K_{(4;\vec{k}_i)}(\tilde{x}, x) = \Gamma_e\left(pq^2\right) C_i \prod_{l=1}^{2N+4} \prod_{j \neq i}^{N+1} \theta_p\left((pq)^{\frac{1}{2}}\left(u^N w^2\right)^{-2N-4} x_j\right) \Gamma_e\left((pq)^{\frac{1}{2}} u^{-2(N+2)^2} \tilde{x}_j^{-1} q\right) \times$$

$$\Gamma_e\left((pq)^{\frac{1}{2}}\left(u^N w^2\right)^{-2N-4} x_j q\right) \Gamma_e\left((pq)^{\frac{1}{2}} u^{2(N+2)^2} x_j\right) \Gamma_e\left((pq)^{\frac{1}{2}}\left(uv^{N+1}\right)^{2N+4} x_i\right) \times$$

$$\Gamma_e\left((pq)^{\frac{1}{2}} u^{N+3} v^{-N-1} w^{-2} a_l x_i\right) \Gamma_e\left((pq)^{\frac{1}{2}} u^{-2(N+2)^2} \tilde{x}_i^{-1} q\right) \Gamma_e\left((pq)^{\frac{1}{2}} u^{2(N+2)^2} x_i q^{-1}\right) \times$$

$$\Gamma_e\left((pq)^{\frac{1}{2}} u^{-N-3} v^{N+1} w^2 x_i^{-1} a_l^{-1} q\right) \Gamma_e\left((pq)^{\frac{1}{2}}\left(uv^{N+1}\right)^{-2N-4} x_i^{-1} q\right) \theta_p\left(\frac{x_j}{x_i}\right)^{-1} \times$$

$$\theta_p\left((pq)^{\frac{1}{2}} u^{-2(N+2)^2} x_i^{-1} q\right)^{-1} \left[\prod_{m=1}^{N+1} \Gamma_e\left(x_j \tilde{x}_m^{-1}\right) \Gamma_e\left(q^{-1} x_i \tilde{x}_m^{-1}\right)\right],$$

$$\text{(B.29)}$$

where

$$C_i = \Gamma_e\left(q^{-1}\left(uv^{-1}\right)^{2(N+1)(N+2)}\right) \Gamma_e\left(q^{-1} u^{(N+1)(2N+5)} v^{N+1} w^2 a_l^{-1}\right) \times$$

$$\Gamma_e\left(pq^3\left(u^{N+1} w\right)^{-4N-8}\right). \qquad \text{(B.30)}$$

Once we glue this tube to an arbitrary theory we obtain contour pinching at

$$\tilde{x}_j = q^{k_j} x_{\sigma(j)}, \qquad \tilde{x}_i = q^{k_i-1} x_{\sigma(i)}, \qquad \sum_{i=1}^{N+1} k_i = 1. \qquad \text{(B.31)}$$

where $\sigma(i)$ are permutations. We fix these permutations to $\sigma(i) = i$. All other permutations give the same result so we can just multiply the answer with the irrelevant constant factor of $(N+1)!$. Once again we have two distinct classes of partitions of 1.

**I.a.** First partition is $k_i = 1, k_j = 0 \; \forall \; j \neq i$ which corresponds to the pinching at $\tilde{x}_j = x_j \; \forall \; j$. In this case we obtain

$$\mathcal{I}_1^{(\vec{k}_i;x)}(x) = \prod_{j \neq i}^{N+1} \frac{\theta_p\left((pq)^{\frac{1}{2}}\left(u^N w^2\right)^{-2N-4} x_j\right) \theta_p\left((pq)^{\frac{1}{2}} u^{-2(N+2)^2} x_j^{-1}\right)}{\theta_p\left(\frac{x_j}{x_i}\right) \theta_p\left(q^{-1} \frac{x_i}{x_j}\right)} \times$$

$$\frac{\theta_p\left((pq)^{\frac{1}{2}}\left(uv^{N+1}\right)^{-2N-4} x_i^{-1}\right)}{\theta_p\left((pq)^{\frac{1}{2}} u^{-2(N+2)^2} q x_i^{-1}\right)} \prod_{l=1}^{2N+4} \theta_p\left((pq)^{\frac{1}{2}} u^{-N-3} v^{N+1} w^2 a_l^{-1} x_i^{-1}\right) C_i \mathcal{I}_0(x), \quad \text{(B.32)}$$

**I.b.** Second class of possible partitions is $k_m = 1$ for some $m \neq i$ and all other $k_j = 0$ including $j = i$. This choice of the partition corresponds to the following pinching:

$$\tilde{x}_i = q^{-1} x_i, \quad \tilde{x}_m = q x_m, \quad \tilde{x}_j = x_j, \; \forall \; j \neq i \neq m. \qquad \text{(B.33)}$$

– 56 –

In this case we obtain:

$$
\mathcal{I}_1^{(\vec{k}_i;qx)}(x) = \sum_{m \neq i} \frac{\theta_p\left((pq)^{\frac{1}{2}}\left(u^N w^2\right)^{-2N-4} x_m\right) \theta_p\left((pq)^{\frac{1}{2}}\left(uv^{N+1}\right)^{-2N-4} x_i^{-1}\right)}{\theta_p\left(\frac{x_m}{x_i}\right)\theta_p\left(q\frac{x_m}{x_i}\right)} \times
$$

$$
\prod_{l=1}^{2N+4} \theta_p\left((pq)^{\frac{1}{2}} u^{-N-3} v^{N+1} w^2 a_l^{-1} x_i^{-1}\right) \times
$$

$$
\prod_{j \neq m \neq i}^{N+1} \frac{\theta_p\left((pq)^{\frac{1}{2}} u^{-2(N+2)^2} x_j^{-1}\right) \theta_p\left((pq)^{\frac{1}{2}}\left(u^N w^2\right)^{-2N-4} x_j\right)}{\theta_p\left(\frac{x_j}{x_i}\right)\theta_p\left(\frac{x_m}{x_j}\right)} \Delta_{im}^{(1,0)} C_i \mathcal{I}_0(x) . \quad (B.34)
$$

**II.** Second class of pinchings in the gluing (B.27) corresponds to the partition:

$$
k_{N+2} = 1 , \quad k_i = 0 . \quad (B.35)
$$

This partition leads to the following index of the tube theory:

$$
K_{(4;k_{N+2})}^{A_N}(x,\tilde{x}) = \Gamma_e\left(pq\right) C_{N+2} \prod_{j=1}^{N+1} \prod_{l=1}^{2N+4} \Gamma_e\left((pq)^{\frac{1}{2}}\left(u^N w^2\right)^{2N+4} x_j^{-1}\right) \times
$$

$$
\Gamma_e\left((pq)^{\frac{1}{2}} u^{-2(N+2)^2} \tilde{x}_j^{-1}\right) \Gamma_e\left((pq)^{\frac{1}{2}}\left(u^N w^2\right)^{-2N-4} x_j q\right) \Gamma_e\left((pq)^{\frac{1}{2}} u^{2(N+2)^2} x_j\right) \times
$$

$$
\theta_p\left((pq)^{-\frac{1}{2}} u^{2(N+2)^2} x_j q^{-1}\right)^{-1} \prod_{i=1}^{N+1} \Gamma_e\left(x_i \tilde{x}_j^{-1}\right) , \quad (B.36)
$$

where

$$
C_{N+2} = \Gamma_e\left(\left(uv^{-1}\right)^{2(N+1)(N+2)}\right) \Gamma_e\left(u^{(N+1)(2N+5)} v^{N+1} w^2 a_l^{-1}\right) \times
$$

$$
\Gamma_e\left(pq^2 \left(u^{N+1} w\right)^{-4N-8}\right) . \quad (B.37)
$$

Once we glue it to the index of the arbitrary theory we obtain the pinching at $\tilde{x}_i = x_i$ resulting in

$$
\mathcal{I}^{k_{N+2}}(x) = C_{N+2} \prod_{j=1}^{N+1} \frac{\theta_p\left((pq)^{\frac{1}{2}}\left(u^N w^2\right)^{-2N-4} x_j\right)}{\theta_p\left((pq)^{-\frac{1}{2}} u^{2(N+2)^2} x_j q^{-1}\right)} \mathcal{I}_0(x) . \quad (B.38)
$$

Finally one should perform similar computation for the pinching specified in the second line of (B.25). This calculation is basically identical and leads to exactly the same results. We leave the calculation to the reader. Summing terms (B.32), (B.34) and (B.38) and performing conjugation (4.11) we finally obtain the A$\Delta$O (4.15) with the shift term given in (4.16) and constant term of (4.17).

## B.4  Difference operator from $A_1^1$ trinion.

In the present appendix we give details of the derivation of the A$\Delta$O (3.28) otained using $A_1^1$ trinion theory. We start with computing indices of the tube theories (3.27),(3.26) with and without the defects. For this we start with the trinon index (3.24) and set $\epsilon = (pq)^{\frac{1}{2}} t^2 a^2 q$ or $\epsilon = (pq)^{\frac{1}{2}} t^2 a^2$. Let's consider the case of closing without the defect. In this case substituting the value above we obtain

$$
\begin{aligned}
K_{(2;0)}^{A_1^1}(v,z) = \kappa^2 \oint \frac{dy_1}{4\pi i y_1} \oint \frac{dy_2}{4\pi i y_2} \frac{\prod_{i=1}^4 \Gamma_e\left(t^{-1} a^{-1} c_i y_1^{\pm 1}\right) \Gamma_e\left(t^{-1} a^{-1} \widetilde{c}_i y_2^{\pm 1}\right)}{\Gamma_e\left(y_1^{\pm 2}\right) \Gamma_e\left(y_2^{\pm 2}\right)} \\
\prod_{i=1}^3 \Gamma_e\left(pq t^2 a^2 c_i c_4\right) \Gamma_e\left(pq t^2 a^2 \widetilde{c}_i \widetilde{c}_4\right) \Gamma_e(a^{-4}) \\
\Gamma_e\left((pq)^{1/2} a^2 y_1^{\pm 1} z^{\pm 1}\right) \Gamma_e\left((pq)^{1/2} z^{\pm 1} y_2^{\pm 1}\right) \Gamma_e\left(t^{-2} a^{-2} z^{\pm 1} v^{\pm 1}\right) \\
\Gamma_e\left((pq)^{1/2} t^2 y_1^{\pm 1} v^{\pm 1}\right) \Gamma_e\left((pq)^{1/2} t^2 a^2 v^{\pm 1} y_2^{\pm 1}\right) .
\end{aligned}
\tag{B.39}
$$

Now we can use the duality identity (A.10). We split variables so that $t^{-1} a^{-1} \widetilde{c}_i$ is in the first group of four and $(pq)^{\frac{1}{2}} z^{\pm 1}$ with $(pq)^{\frac{1}{2}} t^2 a^2 v^{\pm 1}$ in the second group of four. We then obtain:

$$
\begin{aligned}
K_{(2;0)}^{A_1^1}(v,z) = \kappa^2 \Gamma_e\left(pq\right) \oint \frac{dy_1}{4\pi i y_1} \oint \frac{dy_2}{4\pi i y_2} \frac{\prod_{i=1}^4 \Gamma_e\left(t^{-1} a^{-1} c_i y_1^{\pm 1}\right) \Gamma_e\left((pq)^{\frac{1}{2}} t a \widetilde{c}_i y_2^{\pm 1}\right)}{\Gamma_e\left(y_1^{\pm 2}\right) \Gamma_e\left(y_2^{\pm 2}\right)} \\
\prod_{i=1}^3 \Gamma_e\left(pq t^2 a^2 c_i c_4\right) \Gamma_e\left(t^{-2} a^{-2} \widetilde{c}_i \widetilde{c}_4\right) \Gamma_e(a^{-4}) \Gamma_e(pq t^4 a^4) \Gamma_e\left(v^{\pm 1} y_2^{\pm 1}\right) \\
\Gamma_e\left((pq)^{1/2} a^2 y_1^{\pm 1} z^{\pm 1}\right) \Gamma_e\left(t^{-2} a^{-2} z^{\pm 1} y_2^{\pm 1}\right) \Gamma_e\left((pq)^{1/2} t^2 y_1^{\pm 1} v^{\pm 1}\right) .
\end{aligned}
\tag{B.40}
$$

Notice that there is $\Gamma_e\left(pq\right)$ factor which equals zero. However, as usual, at the same time there is pinching of the integration contour taking place at $y_1 = v^{\pm 1}$ which cancels this zero. Computing contribution of this pinching we finally obtain an expression (3.26). In an identical way we can obtain the tube (3.27) with the defect denoted as $K_{(2;1)}^{A_1^1}(v,z)$.

Now having expressions for both tubes we can conjugate one of them and perform S gluings of the following form:

$$
\mathcal{O} \cdot \mathcal{I}(v) = \kappa^2 \oint \frac{dz}{4\pi i z} \frac{dw}{4\pi i w} \frac{1}{\Gamma_e\left(z^{\pm 2}\right) \Gamma_e\left(w^{\pm 2}\right)} K_{(2;1)}^{A_1^1}(v,z) \bar{K}_{(2;0)}^{A_1^1}(w,z) \mathcal{I}(w) ,
\tag{B.41}
$$

where $\bar{K}_{(2,0)}^{A_1^1}$ is the conjugated tube, *i.e.* the one with all of the charges flipped. Substituting

(3.26) and (3.27) into expression above we obtain the following result:

$$
\mathcal{OI}(v) = \kappa^4 \oint \frac{dw}{4\pi i w} \frac{1}{\Gamma_e(w^{\pm 2})} \oint \frac{dz}{4\pi i z} \frac{1}{\Gamma_e(z^{\pm 2})} \oint \frac{dy_1}{4\pi i y_1} \frac{\prod_{i=1}^{4} \Gamma_e\left(q^{-\frac{1}{2}} t^{-1} a^{-1} c_i y_1^{\pm 1}\right)}{\Gamma_e\left(y_1^{\pm 2}\right)}
$$

$$
\Gamma_e\left((pq)^{1/2} q^{\frac{1}{2}} a^2 y_1^{\pm 1} z^{\pm 1}\right) \Gamma_e\left((pq)^{1/2} q^{\frac{1}{2}} t^2 y_1^{\pm 1} v^{\pm 1}\right) \left[ \frac{\Gamma_e(v^2)}{\Gamma_e(qv^2)} \Gamma_e\left(t^{-2} a^{-2} v z^{\pm 1}\right) \times \right.
$$

$$
\Gamma_e\left(q^{-1} t^{-2} a^{-2} v^{-1} z^{\pm 1}\right) \Gamma_e\left((pq)^{\frac{1}{2}} q t a \widetilde{c}_i v\right) \Gamma_e\left((pq)^{\frac{1}{2}} t a \widetilde{c}_i v^{-1}\right) + \{v \leftrightarrow v^{-1}\} \Big]
$$

$$
\oint \frac{dx_1}{4\pi i x_1} \frac{\prod_{i=1}^{4} \Gamma_e\left((pq)^{\frac{1}{2}} t^{-1} a^{-1} c_i^{-1} x_1^{\pm 1}\right)}{\Gamma_e\left(x_1^{\pm 2}\right)} \Gamma_e\left(t^2 x_1^{\pm 1} z^{\pm 1}\right) \Gamma_e\left(a^2 x_1^{\pm 1} w^{\pm 1}\right) \times
$$

$$
\Gamma_e\left((pq)^{\frac{1}{2}} t^{-1} a^{-1} \widetilde{c}_i^{-1} w^{\pm 1}\right) \mathcal{I}(w) . \tag{B.42}
$$

Next we perform two duality transformations of (A.10) type. First one is on $x_1$ node. The split of the fields is chosen so that those charged under $z$ or $w$ are grouped together and so the gauge singlets of the dual theory contain a bifundamental of $SU(2)_z$ and $SU(2)_w$ gauge groups. Second duality is performed on the $z$ node. Similarly to the first duality the fields charged under $w$ or $v$ are grouped together. We then get,

$$
\kappa^4 \frac{\Gamma_e(v^2)}{\Gamma_e(qv^2)} \oint \frac{dw}{4\pi i w} \frac{1}{\Gamma_e(w^{\pm 2})} \oint \frac{dz}{4\pi i z} \frac{1}{\Gamma_e(z^{\pm 2})} \oint \frac{dy_1}{4\pi i y_1} \frac{\prod_{i=1}^{4} \Gamma_e\left(q^{-\frac{1}{2}} t^{-1} a^{-1} c_i y_1^{\pm 1}\right)}{\Gamma_e\left(y_1^{\pm 2}\right)} \times
$$

$$
\Gamma_e\left(a^2 y_1^{\pm 1} z^{\pm 1}\right) \Gamma_e\left((pq)^{1/2} q^{\frac{1}{2}} t^2 y_1^{\pm 1} v^{\pm 1}\right) \Gamma_e\left((pq)^{\frac{1}{2}} q^{\frac{1}{2}} t^{-2} a^{-2} v z^{\pm 1}\right) \Gamma_e\left((pq)^{\frac{1}{2}} q t a \widetilde{c}_i v\right) \times
$$

$$
\Gamma_e\left((pq)^{\frac{1}{2}} q^{-\frac{1}{2}} t^{-2} a^{-2} v^{-1} z^{\pm 1}\right) \Gamma_e\left((pq)^{\frac{1}{2}} t a \widetilde{c}_i v^{-1}\right) \oint \frac{dx_1}{4\pi i x_1} \frac{\prod_{i=1}^{4} \Gamma_e\left(t a c_i^{-1} x_1^{\pm 1}\right)}{\Gamma_e\left(x_1^{\pm 2}\right)} \times
$$

$$
\Gamma_e\left(q^{-\frac{1}{2}} a^{-2} x_1^{\pm 1} z^{\pm 1}\right) \Gamma_e\left((pq)^{\frac{1}{2}} t^{-2} x_1^{\pm 1} w^{\pm 1}\right) \Gamma_e\left((pq)^{\frac{1}{2}} q^{\frac{1}{2}} t^2 a^2 z^{\pm 1} w^{\pm 1}\right) \Gamma_e(v w^{\pm 1}) \times
$$

$$
\Gamma_e(q^{-1} v^{-1} w^{\pm 1}) \Gamma_e(pq q^{\frac{1}{2}} x_1^{\pm 1} y_1^{\pm 1}) \Gamma_e\left((pq)^{\frac{1}{2}} t^{-1} a^{-1} \widetilde{c}_i^{-1} w^{\pm 1}\right) \mathcal{T}(w) + \{v \leftrightarrow v^{-1}\} \tag{B.43}
$$

The poles of $\Gamma_e\left(q^{-1} v^{-1} w^{\pm 1}\right)$ and $\Gamma_e\left(v w^{\pm 1}\right)$ collide when $w = v^{\pm 1}$ or $w = (qv)^{\pm 1}$ which results in the integration contour pinching. We should sum over contributions of all these pinchings. Terms with opposite powers contribute equivalently since $w$ is a fugacity for the Cartan generator of $SU(2)$. Substituting values of $w$ at the pinching points we can integrate out first $z$ and then $x_1$ integrals using $A_1$ elliptic beta integral (A.4). The result of these

integrations is

$$\kappa\Gamma_e(a^4)\Gamma_e(q^{-1}a^{-4})\oint\frac{dy_1}{4\pi i y_1}\frac{\prod_{i=1}^4\Gamma_e\left(q^{-\frac{1}{2}}t^{-1}a^{-1}c_iy_1^{\pm1}\right)}{\Gamma_e\left(y_1^{\pm2}\right)}\Gamma_e\left((pq)^{\frac{1}{2}}q^{\frac{1}{2}}t^{-2}vy_1^{\pm1}\right)\times$$

$$\Gamma_e\left((pq)^{\frac{1}{2}}q^{\frac{1}{2}}t^2a^4v^{-1}y_1^{\pm1}\right)\Gamma_e\left((pq)^{1/2}q^{\frac{1}{2}}t^2y_1^{\pm1}v^{\pm1}\right)\frac{\Gamma_e(v^2)\Gamma_e(q^{-1}v^{-2})}{\Gamma_e(qv^2)\Gamma_e(v^{-2})}\times$$

$$\prod_{i=1}^4\Gamma_e\left((pq)^{\frac{1}{2}}qta\widetilde{c}_iv\right)\Gamma_e\left((pq)^{\frac{1}{2}}t^{-1}a^{-1}\widetilde{c}_i^{-1}v^{-1}\right)\Gamma_e(pqt^{-4}a^{-4})\prod_{i=1}^4\Gamma_e\left((pq)^{\frac{1}{2}}t^{-1}a^{-3}c_i^{-1}v\right)\times$$

$$\Gamma_e\left((pq)^{\frac{1}{2}}t^{-1}ac_i^{-1}v^{-1}\right)\prod_{j<k}\Gamma_e(t^2a^2c_j^{-1}c_k^{-1})\mathcal{I}(v)+\{v\leftrightarrow v^{-1}\}$$

$+$

$$\kappa\Gamma_e(pqt^{-4}a^{-4})\Gamma_e(a^4)\Gamma_e(q^{-1}a^{-4})\oint\frac{dy_1}{4\pi i y_1}\frac{\prod_{i=1}^4\Gamma_e\left(q^{-\frac{1}{2}}t^{-1}a^{-1}c_iy_1^{\pm1}\right)}{\Gamma_e\left(y_1^{\pm2}\right)}\times$$

$$\Gamma_e\left((pq)^{\frac{1}{2}}q^{-\frac{1}{2}}t^{-2}v^{-1}y_1^{\pm1}\right)\Gamma_e\left((pq)^{\frac{1}{2}}q^{\frac{3}{2}}t^2a^4vy_1^{\pm1}\right)\Gamma_e\left((pq)^{1/2}q^{\frac{1}{2}}t^2y_1^{\pm1}v^{\pm1}\right)\frac{\Gamma_e(v^2)}{\Gamma_e(q^2v^2)}\times$$

$$\prod_{i=1}^4\Gamma_e\left((pq)^{\frac{1}{2}}qta\widetilde{c}_iv\right)\Gamma_e\left((pq)^{\frac{1}{2}}ta\widetilde{c}_iv^{-1}\right)\prod_{j<k}\Gamma_e(t^2a^2c_j^{-1}c_k^{-1})\prod_{i=1}^4\Gamma_e\left((pq)^{\frac{1}{2}}q^{-1}t^{-1}a^{-3}c_i^{-1}v^{-1}\right)\times$$

$$\Gamma_e\left((pq)^{\frac{1}{2}}qt^{-1}ac_i^{-1}v\right)\Gamma_e\left((pq)^{\frac{1}{2}}t^{-1}a^{-1}\widetilde{c}_i^{-1}(qv)^{\pm1}\right)\mathcal{I}(qv)+\{v\leftrightarrow v^{-1}\}\,,\tag{B.44}$$

where the first term comes from the pinching at $w=v^{\pm1}$ and the second from the pinching at $w=(qv)^{\pm1}$. In both of the terms above we get an SU(2) theory with 4 flavors. In order to proceed we perform (A.10) duality for these two theories. For this in each of the cases we group together fields transforming in the fundamental representation of $SU(4)_c$ global symmetry. Dual expressions are then given by

$$\kappa\Gamma_e(a^4)\Gamma_e(q^{-1}a^{-4})\Gamma_e(pq^2)\Gamma_e(pq^2a^4)\Gamma_e(pq^2t^4a^4)\Gamma_e(pq^2t^4)\Gamma_e(pq^2v^2)\Gamma_e(pq^2t^4a^4v^{-2})\times$$

$$\prod_{j<k}\Gamma_e(q^{-1}t^{-2}a^{-2}c_jc_k)\oint\frac{dy_1}{4\pi i y_1}\frac{1}{\Gamma_e(y_1^{\pm2})}\prod_{i=1}^4\Gamma_e\left((pq)^{\frac{1}{2}}q^{\frac{1}{2}}tac_iy_1^{\pm1}\right)\Gamma_e\left(q^{-\frac{1}{2}}t^{-4}a^{-2}vy_1^{\pm1}\right)\times$$

$$\Gamma_e\left(q^{-\frac{1}{2}}a^2v^{-1}y_1^{\pm1}\right)\Gamma_e\left(q^{-\frac{1}{2}}a^{-2}v^{\pm1}y_1^{\pm1}\right)\frac{\Gamma_e(v^2)\Gamma_e(q^{-1}v^{-2})}{\Gamma_e(qv^2)\Gamma_e(v^{-2})}\prod_{i=1}^4\Gamma_e\left((pq)^{\frac{1}{2}}qta\widetilde{c}_iv\right)\times$$

$$\Gamma_e\left((pq)^{\frac{1}{2}}t^{-1}a^{-1}\widetilde{c}_i^{-1}v^{-1}\right)\Gamma_e(pqt^{-4}a^{-4})\prod_{i=1}^4\Gamma_e\left((pq)^{\frac{1}{2}}t^{-1}a^{-3}c_i^{-1}v\right)\Gamma_e\left((pq)^{\frac{1}{2}}t^{-1}ac_i^{-1}v^{-1}\right)\times$$

$$\prod_{j<k}\Gamma_e(t^2a^2c_j^{-1}c_k^{-1})\mathcal{I}(v)+\{v\leftrightarrow v^{-1}\}$$

$$+$$

$$\kappa\Gamma_e(pq)\Gamma_e(pqv^{-2})\Gamma_e(pq^2a^4)\Gamma_e(pq^3t^4a^4)\Gamma_e(pq^3t^4a^4v^2)\Gamma_e(pq^2t^4)\Gamma_e(pqt^{-4}a^{-4})\Gamma_e(a^4)\Gamma_e(q^{-1}a^{-4})\times$$

$$\oint \frac{dy_1}{4\pi i y_1}\frac{\prod_{i=1}^4\Gamma_e\left((pq)^{\frac{1}{2}}q^{\frac{1}{2}}tac_iy_1^{\pm1}\right)}{\Gamma_e\left(y_1^{\pm2}\right)}\Gamma_e\left(q^{-\frac{3}{2}}t^{-4}a^{-2}v^{-1}y_1^{\pm1}\right)\Gamma_e\left(q^{\frac{1}{2}}a^2vy_1^{\pm1}\right)\frac{\Gamma_e(v^2)}{\Gamma_e(q^2v^2)}\times$$

$$\Gamma_e\left(q^{-\frac{1}{2}}a^{-2}v^{\pm1}y_1^{\pm1}\right)\prod_{i=1}^4\Gamma_e\left((pq)^{\frac{1}{2}}qta\widetilde{c}_iv\right)\Gamma_e\left((pq)^{\frac{1}{2}}ta\widetilde{c}_iv^{-1}\right)\prod_{j<k}\Gamma_e(t^2a^2c_j^{-1}c_k^{-1})\times$$

$$\Gamma_e(q^{-1}t^{-2}a^{-2}c_jc_k)\prod_{i=1}^4\Gamma_e\left((pq)^{\frac{1}{2}}q^{-1}t^{-1}a^{-3}c_i^{-1}v^{-1}\right)\Gamma_e\left((pq)^{\frac{1}{2}}qt^{-1}ac_i^{-1}v\right)\times$$

$$\Gamma_e\left((pq)^{\frac{1}{2}}t^{-1}a^{-1}\widetilde{c}_i^{-1}(qv)^{\pm1}\right)\mathcal{I}(qv)+\{v\leftrightarrow v^{-1}\}\tag{B.45}$$

As usually we obtain zeroes in these expressions. In particular first term contains $\Gamma_e\left(pq^2\right)$ and second $\Gamma_e(pq)$ prefactors. These zeroes are cancelled by pinchings of integration contours. In the first term pinching takes place at $y_1=(q^{\frac{1}{2}}a^2v^{-1})^{\pm1}$ and $y_1=(q^{-\frac{1}{2}}a^2v^{-1})^{\pm1}$. Pinching in the second term is located at $y_1=(q^{\frac{1}{2}}a^2v)^{\pm1}$. Substituting these values into expressions above we precisely get the operator $\mathcal{O}_v^{(\epsilon;t^2a^2;1,0)}$ given by (3.28) and equal to the van Diejen operator (3.1) upon the identification (3.29) of its parameters.

In a completely similar way we can derive A$\Delta$Os for any pair of punctures we close and we act on. For example the operator acting on $\mathrm{SU}(2)_z$ puncture that is obtained by giving vev to the moment map of $\mathrm{U}(1)_\epsilon$ puncture with the charge $t^2a^2$ is given by,

$$\mathcal{O}_z^{(\epsilon;t^2a^2;1,0)}\cdot\mathcal{I}(z)=\frac{\prod_{i=1}^4\theta_p\left((pq)^{\frac{1}{2}}ta^{-1}c_i^{-1}z\right)\theta_p\left((pq)^{\frac{1}{2}}ta\widetilde{c}_i^{-1}z\right)}{\theta_p\left(z^2\right)\theta_p\left(qz^2\right)}\mathcal{I}(qz)+$$

$$+\frac{\theta_p\left(q^{-1}t^{-4}\right)\prod_{i=1}^4\theta_p\left((pq)^{\frac{1}{2}}ta^3c_i^{-1}z^{-1}\right)\theta_p\left((pq)^{\frac{1}{2}}ta\widetilde{c}_i^{-1}z\right)}{\theta_p\left(q^{-2}t^{-4}a^{-4}\right)\theta_p\left(z^2\right)\theta_p\left(a^4z^{-2}\right)}\mathcal{I}(z)+$$

$$+\frac{\prod_{i=1}^4\theta_p\left((pq)^{\frac{1}{2}}ta^{-1}c_i^{-1}z\right)\theta_p\left((pq)^{\frac{1}{2}}ta\widetilde{c}_i^{-1}z\right)}{\theta_p\left(q^{-2}t^{-4}a^{-4}\right)\theta_p\left(z^2\right)\theta_p\left(q^{-1}z^{-2}\right)\theta_p\left(a^{-4}z^2\right)}\times$$

$$\theta_p\left(q^{-1}a^{-4}\right)\theta_p\left(q^{-1}t^{-4}a^{-4}z^2\right)\mathcal{I}(z)+\{z\leftrightarrow z^{-1}\},\tag{B.46}$$

Another example is an operator acting on the $\mathrm{U}(1)_\epsilon$ puncture obtained by closing the

$SU(2)_v$ puncture giving vev to the moment map with the cahrge $ta\widetilde{c}_4$. The result is given by,

$$\mathcal{O}_\epsilon^{(v;ta\widetilde{c}_4;1,0)} \cdot \mathcal{I}(\epsilon) = \frac{\theta_p\left((pq)^{\frac{1}{2}}t^{-2}a^{\pm 2}\epsilon\right)\prod_{j=1}^3 \theta_p\left((pq)^{\frac{1}{2}}\widetilde{c}_4^{-1}\widetilde{c}_j^{-1}\epsilon\right)\theta_p\left((pq)^{\frac{1}{2}}c_4^{-1}c_j^{-1}\epsilon\right)}{\theta_p\left(\epsilon^2\right)\theta_p\left(q\epsilon^2\right)}\mathcal{I}(q\epsilon) +$$

$$\frac{\theta_p\left((pq)^{\frac{1}{2}}t^2a^2\epsilon\right)\theta_p\left((pq)^{\frac{1}{2}}t^2\widetilde{c}_4c_4^{-1}\epsilon^{-1}\right)\prod_{j=1}^3 \theta_p\left((pq)^{\frac{1}{2}}\widetilde{c}_4\widetilde{c}_j\epsilon\right)\theta_p\left((pq)^{\frac{1}{2}}a^2\widetilde{c}_4c_j\epsilon^{-1}\right)}{\theta_p\left(q^{-2}t^{-2}a^{-2}\widetilde{c}_4^{-2}\right)\theta_p\left(\epsilon^2\right)\theta_p\left(a^2\widetilde{c}_4c_4^{-1}\epsilon^{-2}\right)} \times$$

$$\theta_p\left(q^{-1}t^{-2}\widetilde{c}_4^{-1}c_4^{-1}\right)\mathcal{I}(\epsilon) + \frac{\theta_p\left(q^{-1}a^{-2}\widetilde{c}_4^{-1}c_4\right)\theta_p\left(q^{-1}t^{-2}a^{-2}\widetilde{c}_4^{-2}\epsilon^2\right)\theta_p\left((pq)^{\frac{1}{2}}t^2a^{\pm 2}\epsilon\right)}{\theta_p\left(q^{-2}t^{-2}a^{-2}\widetilde{c}_4^{-2}\right)\theta_p\left(\epsilon^2\right)\theta_p\left(q^{-1}\epsilon^{-2}\right)\theta_p\left(a^{-2}\widetilde{c}_4^{-1}c_4\epsilon^2\right)} \times$$

$$\prod_{j=1}^3 \theta_p\left((pq)^{\frac{1}{2}}\widetilde{c}_4\widetilde{c}_j\epsilon\right)\theta_p\left((pq)^{\frac{1}{2}}c_4c_j\epsilon\right)\mathcal{I}(\epsilon) + \{\epsilon \leftrightarrow \epsilon^{-1}\}\,. \quad \text{(B.47)}$$

Both of these operators above can be mapped onto the van Diejen model. Correspodning dictionaries can be read from the difference parts of operators. In particular for operator (B.46) the map is given by:

$$h_i = ta^{-1}c_i^{-1}\,, \qquad h_{i+4} = ta\widetilde{c}_i^{-1}\,, \quad i = 1,\dots,4\,. \quad \text{(B.48)}$$

while for the operator (B.47) we obtain:

$$h_i = c_4^{-1}c_i^{-1}\,, \qquad h_{i+3} = \widetilde{c}_4^{-1}\widetilde{c}_i^{-1}\,, \quad i = 1,2,3\,, \qquad h_{7,8} = t^{-2}a^{\pm 2}\,. \quad \text{(B.49)}$$

Just like in (3.29) van Diejen parameters map to the inverse U(1) charges of the moment maps of the puncture we act on as can be seen from (3.25). Constant parts of both operators (B.46) and (B.47) are elliptic functions with periods 1 and $p$. It can be checked that poles and corresponding residues of these constant parts coincide with those of van Diejen model summarized in (3.6) and (3.7). Hence we conclude that both of these operators can differ from the van Diejen model at most by an irrelevant constant part.

## B.5 Derivation of the $USp(2N)$ trinion

In this appendix we construct a three punctured sphere, with two maximal $USp(2N)$ punctures and one minimal $SU(2)$ puncture, of the minimal $(D_{N+3}, D_{N+3})$ conformal matter. In the bulk of the paper we use only the degenerate case of $N = 1$ ($SU(2) \sim USp(2)$) but here we will outline the more general construction. The construction starts from the trinion [34] with two maximal $SU(N+1)$ punctures discussed in Section 3, and gluing to the two maximal $SU(N + 1)$ punctures two tubes with one $SU(N + 1)$ puncture and one $USp(2N)$ puncture discussed in [46] (see Figure 17). The dynamics of these gluings turns out to be S-confining. The relevant S-confining gauge theory is depicted in Figure 18. This duality was discussed by Spiridonov and Vartanov in [90] following mathematical results of Spiridonov and Warnaar [91] and we will thus refer to it as Spiridonov-Warnaar-Vartanov (SWV) duality. For $N = 1, 2$ these dualities are special cases of the standard Seiberg duality.

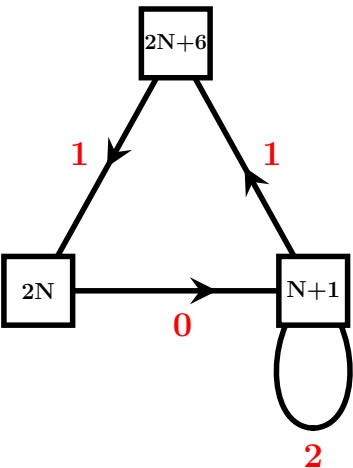

**Figure 17**. A tube theory with one $USp(2N)$ puncture and one $SU(N + 1)$ puncture. The flux corresponding to this tube breaks the $6d$ $SO(4N + 12)$ symmetry to $U(1) \times SU(2N + 6)$ for general $N$ (breaks $E_8$ to $U(1) \times E_7$ for $N = 1$). The line starting and ending on the same gauge node is in two index antisymmetric representation. We have a superpotential terms corresponding to the triangle and coupling the two index antisymmetric to the antisymmetric square of the bifundamental between $USp(2N)$ and $SU(N + 1)$. The R charges depicted here are the six dimensional ones.

Let us briefly discuss the SWV duality. We start with asymptotically free $SU(N + 1)$ SQCD with one chiral field in the antisymmetric representation, $2N$ fields in the anti-fundamental representation, and $N + 3$ fields in the fundamental representation. This is a non-anomalous vector like matter content. We have two non-anomalous abelian symmetries: one depicted as $U(1)_t$ in Figure 18 under which the anti-fundamentals have charge $+1$ and the antisymmetric field has charge $-2$, while under another one, denoted by $U(1)_b$, the antifundamentals have charge $+\frac{1}{2N}$, the antisymmetric charge $0$ and the fundamentals charge $-\frac{1}{N+3}$. Anomaly free choice of R charges, which we denote by $R_{6d}$, is depicted in Figure 18. One can perform a-maximization [92] on the two abelian symmetries and discover that the theory flows to an SCFT with all the operators above the unitarity bound. For example for the case $N = 2$ the superconformal R-symmetry is,

$$R = R_{6d} + \frac{4}{5}q_t - \frac{8}{5}q_b. \tag{B.50}$$

Next we consider deforming the theory by a superpotential of the form $W = q^2 A$. This is a relevant deformation for any $N$. *E.g.* for $N = 2$ the R-charge of this deformation is $\frac{6}{5} < 2$. This deformation breaks the $U(1)_b$ symmetry. The claim of the SWV [90, 91] is that this theory flows to a WZ model in the IR, as depicted in Figure 18. In particular it was shown that the supersymmetric indices of the two dual sides are exactly the same. For our computations this equality is an essential input.

Let us now use the SWV duality to derive a three punctured sphere with two $USp(2N)$ maximal punctures. For $N > 1$ the three punctured sphere we will derive will be an IR free gauge theory while for the $N = 1$ case it will be an interacting SCFT in the IR. We first give

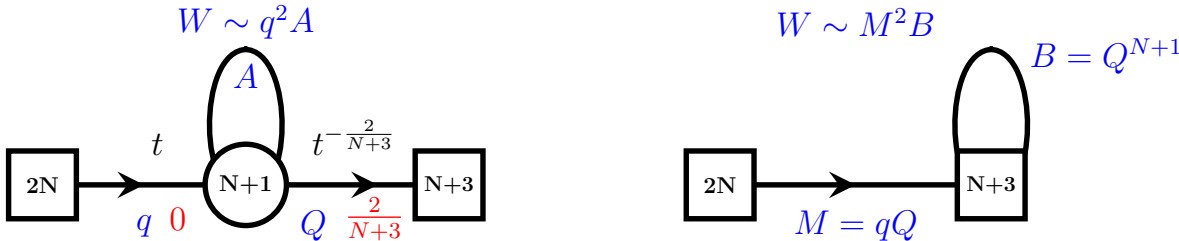

**Figure 18**. Spiridonov-Warnaar-Vartanov (SWV) S-confining duality. The line starting and ending on the same gauge node is in two index antisymmetric representation. The theory on the left has a superpotential in the UV and the theory on the right has a dynamically generated superpotential. The duality can be reduced to Seiberg duality [93] for low values of $N$. Note that for $N = 1$ the duality reduces to the S-confinement of SU(2) with six fundamentals. For $N = 2$ we have SU(3) SQCD with five flavors and a baryonic superpotential breaking SU(5) to USp(4). The Seiberg dual theory is SU(2) SQCD with five flavors, gauge singlets, and the effect of the baryonic superpotential is a mass term for four fundamentals. This leaves us again with S-confining SU(2) with six fundamentals and a bunch of gauge singlet fields.

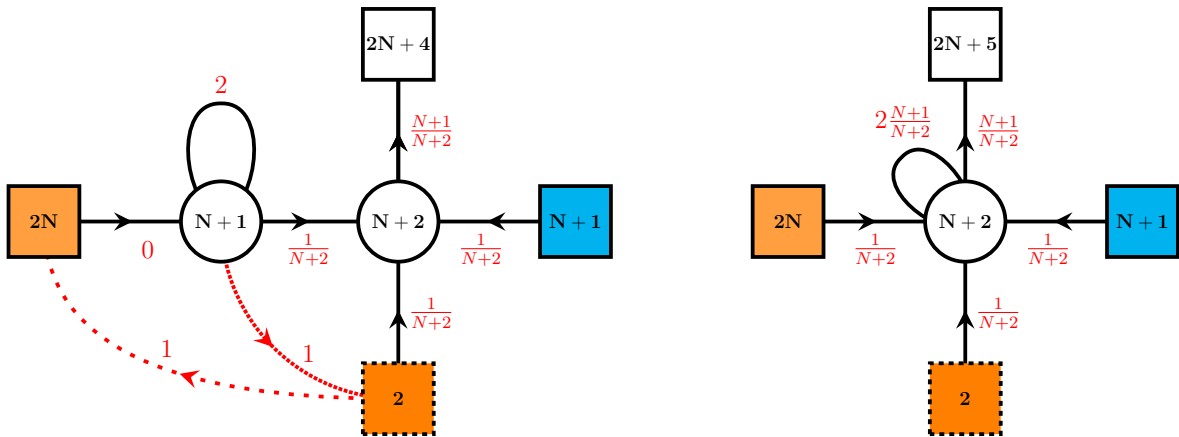

**Figure 19**. We glue a tube to one of the maximal SU($N + 1$) punctures. The dotted line represents the moment map field coming from the tube flipping the baryonic moment map of the trinion. We then use the SWV duality to reduce the dynamics of the SU($N + 1$) gauge node to a WZ model and obtain the theory on the right.

a derivation for general $N$ and then give more details for $N = 1$. We start with the SU($N+1$) trinion and glue it to one of its SU($N+1$) maximal punctures the tube. Note that we need to choose whether to $\Phi$ glue, to S-glue, or a combination of the two. Because of gauge anomalies we cannot use only $\Phi$ gluing or S-gluing here. In fact the difference between the number of components of the moment maps we use to glue in either way needs to be odd. We choose to $\Phi$-glue all the components of the moment maps but one: the one we S-glue flips the baryonic moment map component which is built from $N$ of the quarks charged under the SU($N + 1$) of the puncture we glue and the 2 quarks charged under the minimal puncture SU(2). See Figure 19.

In order to analyze the dynamics we turn on deformations one by one. We consider the three punctured sphere with two $SU(N + 1)$ punctures at the CFT point where all the chiral fields have superconformal R-charge $+\frac{1}{2}$. First we turn on the gauge coupling. The anomaly $\mathrm{Tr}RSU(N + 1)^2 = \frac{N+2}{4}$ is positive and thus the gauging is UV free. We flow to the IR and turn on the superpotentials one by one. First, the superpotential flipping the baryonic moment map is irrelevant at this stage while the two superpotential terms involving the bifundamental of $SU(N + 1)$ and $USp(2N)$ are relevant. We first turn on the one also involving the antisymmetric field. In the IR we perform a-maximization again and now obtain that both remaining superpotentials are relevant. We then turn on the superpotential flipping the baryonic moment map. The remaining superpotential is still relevant so we turn it on and flow to the IR.

Then we use the SWV duality in the version of Figure 20 to obtain a simpler theory with only one $SU(N + 2)$ gauge group. We thus obtain a trinion with one maximal $SU(N + 1)$ puncture, one maximal $USp(2N)$ puncture and one minimal $SU(2)$ puncture. The procedure is depicted in Figure 19. The theory on the right side of the Figure has two superpotential terms coupling the antisymmetric field to the antisymmetrics of $SU(N + 2)$ built from quadratic combinations of the fields transforming under the puncture symmetries $SU(2)$ and $USp(2N)$. After performing the duality we have a UV free gauge theory with $\mathrm{Tr}R_{free}SU(N + 1)^2 = \frac{2}{3}$, which is independent of $N$.

Next we can perform the same procedure on the remaining $SU(N+1)$ maximal puncture and we obtain a trinion with two maximal $USp(2N)$ punctures of Figure 21. The same analysis of the dynamics however gives different result. The gauging is UV free as before and then the two superpotentials involving the bifundamental of $USp(2N)$ and $SU(N + 1)$ are relevant while the one involving the baryonic map is irrelevant. We can turn on the two relevant deformations one by one. However, in the end for $N > 1$ the flipping of the baryonic moment map is still irrelevant deformation. We can perform the SWV duality to see another vantage point of the issue. The theory after the duality is depicted in Figure 21 and for $N > 1$ this is an IR free gauge theory. Thus we obtain the trinion with two $USp(2N)$ maximal punctures and one $SU(2)$ puncture which is an IR free gauge theory. We can still use the index of this model to deduce various properties of indices of general theories built from such theories: in particular more complicated theories built from this might flow to interacting SCFTs in the IR. In this paper we explicitly discuss only the interacting case of $N = 1$ and thus we will comment on it a bit more further in the text.

Let us now analyze the case of $N = 1$. The S-confinement of Figure 18 is the Seiberg duality between $SU(2)$ with six fundamentals and a WZ model of fifteen chiral fields. The antisymmetric field becomes a gauge singlet field coupled to one of the quadratic invariants with a cubic superpotential and thus giving it a mass in the IR. The trinion of Figure 21 for $N = 1$ is $SU(3)$ SQCD with eight flavors and a baryonic superpotential. Note that the superconformal R-symmetry of all the chiral fields here is $\frac{5}{8}$. Thus the baryonic superpotentials have R-charge $\frac{15}{8}$ and are relevant. Turning these on one by one we arrive at an SCFT. One can wonder what is the flux to be associated to this three punctured sphere. The theory manifestly preserves $SU(8) \times U(1)_w$ and thus the flux should at least preserve this symmetry.

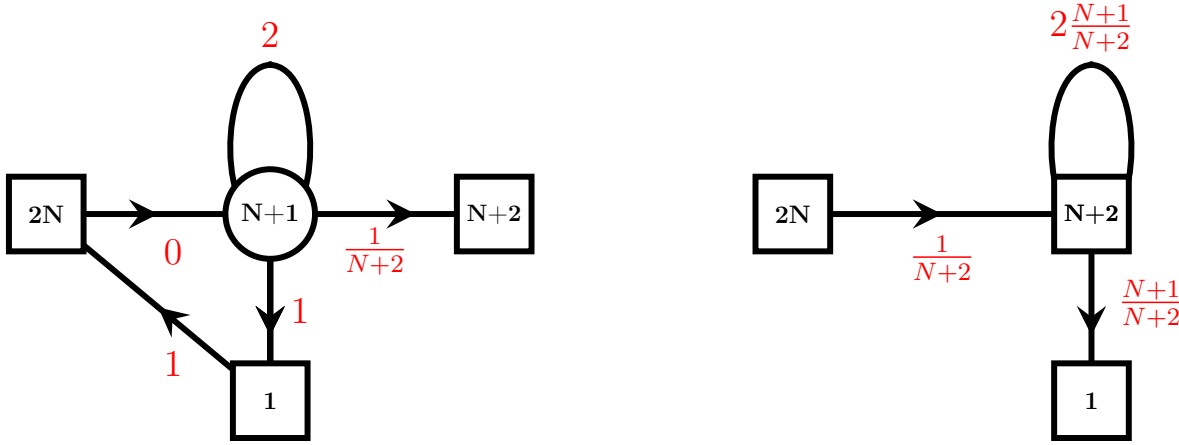

**Figure 20**. A version of SWV duality with an additional field and a superpotential. We split $N + 3$ to $(N + 2) + 1$ and add chiral field in fundamental of USp$(2N)$ which couples to the rest through a cubic superpotential corresponding to a triangle in the figure.

There are two possible choices which do that, an $E_7 \times U(1)$ or SU$(8) \times U(1)$ preserving fluxes [24]. By $\Phi$ gluing two three punctured spheres into a genus two surface we can show that actually the latter one is the right answer. Let us first glue two three punctured spheres into a four punctured sphere. To glue we need to add an octet of SU$(2)$ fundamental fields, turn on cubic superpotentials and gauge the SU$(2)$ symmetry. The gauging is IR free before we turn on the superpotentials. Thus we first take one trinion, add the octet of fields and turn on the cubic superpotential, which is relevant. After turning on the superpotentials the gauging becomes exactly marginal. Note that $\text{Tr}\, R SU(2)^2 = 0$ and there are no anomalous symmetries. We can now glue all the other punctures following the same sequence of steps and arrive at an SCFT. We can compute the index of the resulting theory and read the flux from it. In fact let us quote the index of the theory obtained by following this procedure to construct a theory corresponding to a generic genus $g$ surface,

$$\mathcal{I} = 1 + \tag{B.51}$$
$$\left( 3g - 3 + (\mathbf{63} + \mathbf{1})(g - 1) + (g - 1 + 3F)\mathbf{8}w^{-\frac{27}{2}} + (g - 1 - 3F)\overline{\mathbf{8}}w^{\frac{27}{2}} + (g - 1 - 2F)\mathbf{28}w^9 + \right.$$
$$\left. (g - 1 + 2F)\overline{\mathbf{28}}w^{-9} + (g - 1 + F)\mathbf{56}w^{-\frac{9}{2}} + (g - 1 - F)\overline{\mathbf{56}}w^{\frac{9}{2}} \right) p\,q + \cdots .$$

with $F = g - 1$. As the decomposition of the adjoint of $E_8$ into irreps of SU$(8) \times U(1)_w$ is,

$$\mathbf{248} \to \mathbf{1} + \mathbf{63} + \mathbf{8}_3 + \overline{\mathbf{8}}_{-3} + \mathbf{28}_{-2} + \overline{\mathbf{28}}_2 + \mathbf{56}_1 + \overline{\mathbf{56}}_{-1} , \tag{B.52}$$

the above follows the expected pattern of [94] (see also Appendix E of [24]). We can thus deduce that to build a genus $g$ surface we glue together $2g - 2$ three punctured spheres with each such sphere having half a unit of SU$(8) \times U(1)$ preserving flux associated to it. Yet another check is to compare the anomalies computed using this trinion versus the ones predicted from six dimensions. For example the $a$ anomaly for genus two surface obtained

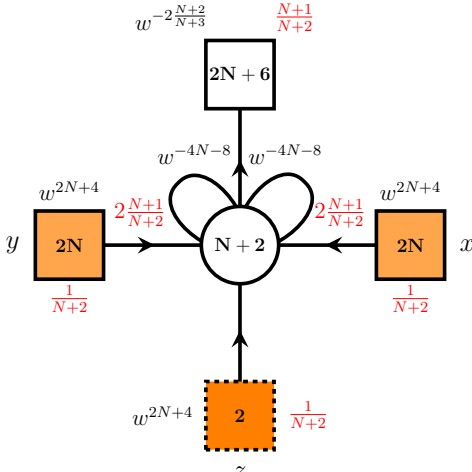

**Figure 21**. A trinion with two maximal USp($2N$) punctures and a minimal SU(2) puncture. Note that the two maximal punctures are of the same type (color). Moreover, following the logic of [38] gluing together $N$ such trinions, the $N$ minimal punctures will recombine into an USp($2N$) maximal puncture somewhere on the conformal manifold. Such a theory then will have tree maximal USp($2N$) punctures all of which are of the same color. For $N = 1$ this theory is asymptotically free. For $N = 2$ the gauge and superpotential interactions are marginally irrelevant. The theory has a $U(1)$ symmetry, such that the fundamentals have charge $+1$ and the antisymmetrics charge $-5$, so that all the marginal couplings have the same charge and thus there is no interacting conformal manifold [95, 96]. For higher values of $N$ the gauge interactions are IR free.

using the trinion here is $\frac{1}{16}\left(17\sqrt{17} + 50\right)$ which can be matched to the $6d$ value extracted *e.g.* from [27] ( $g = 2$, $\xi_G = 4$ for SU(8) flux, and $z = 1$ in equation (2.1) there).

## B.6 Difference operator from $C_1$ trinion.

In this Appendix we present some details of the derivations of van Diejen operator from $C_1$ trinion summarized in the Section 3. We start by S gluing the trinion (3.30) to a tube with no defect (3.32) along $y$ puncture by gauging corresponding SU(2)$_y$ global symmetry. This gluing is S-confining, *i.e.* the resulting SU(2)$_y$ gauge node can be eliminated using $A_1$ elliptic beta integral (A.4). We end up with SU(3) × SU(2) gauge theory. Then we perform Seiberg duality on the SU(3) node which has 7 flavors, so the duality results in SU(4) gauge node with the same number of flavors. Once we do that the SU(2) node becomes S-confining. Integrating it out we are left with SU(4) gauge theory with 7 flavors, one antisymmetric and some flip singlets. This chain of dualities is shown on the Figure 22. Calculation results in the following index of a trinion theory:

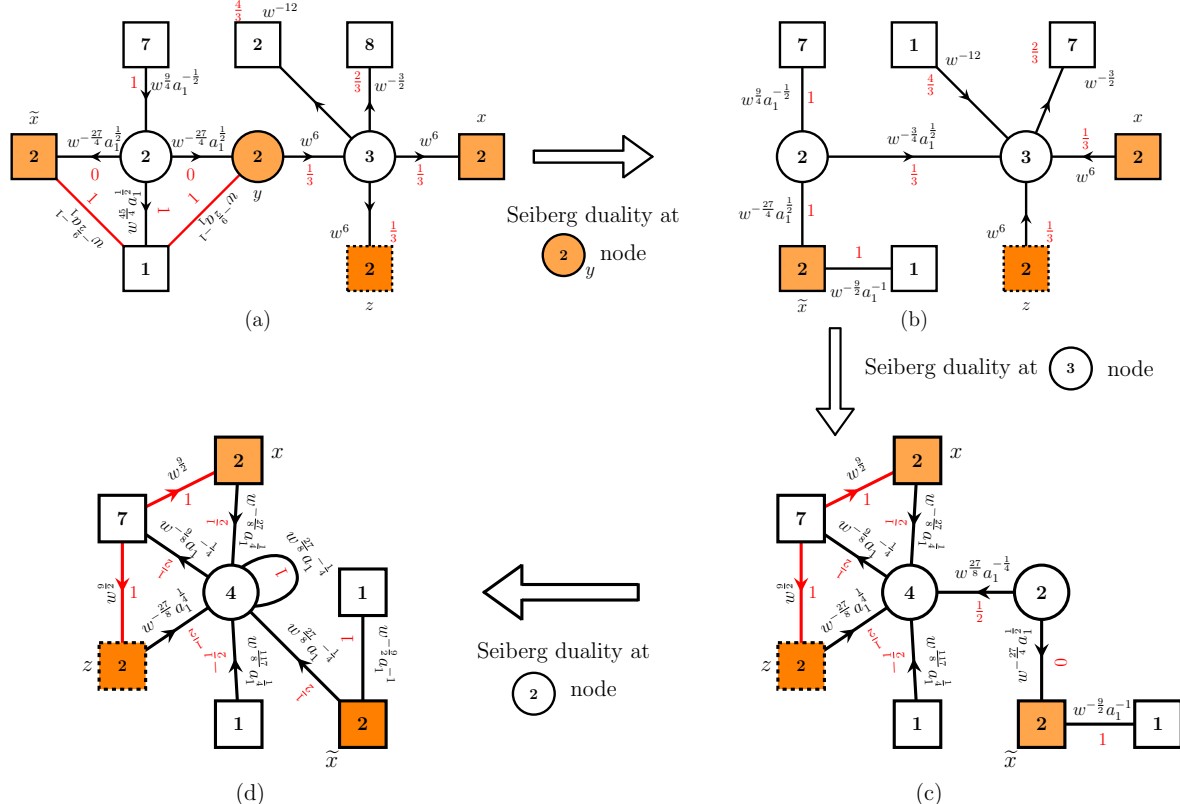

**Figure 22**. On this figure we show the chain of Seiberg dualities used in the derivation of the $C_1$ trinion given in (B.53). On the Figure (a) we start with the S gluing of the $C_1$ trinion (3.30) with the tube (3.32) obtained by closing $SU(2)_z$ minimal puncture with no defect. Then we use Seiberg duality on the S-confining $SU(2)_y$ node. Resulting gauge theory is shown on the Figure (b). After this we perform two Seiberg dualities on one and another gauge nodes one after another. In the end we arrive to the $SU(4)$ gauge theory with 7 flavors, one antisymmetric and some singlets. Corresponding quiver is shown on the Figure (d).

$$K_{(3;0)}^{C_1}(x, x_1, z) = \frac{\kappa^3}{4!} \oint \frac{dt_1}{2\pi i t_1} \frac{dt_2}{2\pi i t_2} \frac{dt_3}{2\pi i t_3} \frac{1}{\prod_{i \neq j}^4 \Gamma_e(t_i/t_j)} \Gamma_e\left((pq)^{\frac{1}{2}} w^{\frac{-9}{2}} a_1^{-1} \widetilde{x}^{\pm 1}\right)$$

$$\prod_{j=1}^4 \Gamma_e\left((pq)^{\frac{1}{4}} w^{-\frac{27}{8}} a_1^{\frac{1}{4}} t_j x^{\pm 1}\right) \Gamma_e\left((pq)^{\frac{1}{4}} w^{-\frac{27}{8}} a_1^{\frac{1}{4}} t_j z^{\pm 1}\right) \Gamma_e\left((pq)^{-\frac{1}{4}} w^{\frac{117}{8}} a_1^{\frac{1}{4}} t_j\right)$$

$$\prod_{i=2}^8 \Gamma_e\left((pq)^{\frac{1}{4}} w^{-\frac{9}{8}} a_1^{-\frac{1}{4}} a_i^{-1} t_j^{-1}\right) \Gamma_e\left((pq)^{\frac{1}{4}} w^{\frac{-27}{8}} a_1^{\frac{1}{4}} t_j \widetilde{x}^{\pm 1}\right) \prod_{k<j}^4 \Gamma_e\left((pq)^{\frac{1}{2}} w^{\frac{27}{4}} a_1^{-\frac{1}{2}} t_j t_k\right)$$

$$\prod_{i=2}^8 \Gamma_e\left(pq w^{-\frac{27}{2}} a_i\right) \Gamma_e\left((pq)^{\frac{1}{2}} w^{\frac{9}{2}} a_i x^{\pm 1}\right) \Gamma_e\left((pq)^{\frac{1}{2}} w^{\frac{9}{2}} a_i z^{\pm 1}\right) \tag{B.53}$$

Now we glue this to a general theory with $\mathrm{SU}(2)_{\tilde{x}}$ global symmetry by gauging it and close $z$-puncture with the defect introduced. It can be seen from the expression above that the mesonic vev, which corresponded to setting $z = (pq)^{-\frac{1}{2}} w^{-\frac{9}{2}} a_1^{-1} q^{-1}$, is now a baryonic one. At this value of $z$ the trinion written above has a pole. So we have to calculate the following residue:

$$\mathcal{O}_x^{(z;w^{\frac{9}{2}}a_1;1,0)} \cdot \mathcal{I}(x) \propto \mathrm{Res}_{z \to (pq)^{-\frac{1}{2}} w^{-\frac{9}{2}} a_1^{-1} q^{-1}} \kappa \oint \frac{d\tilde{x}}{4\pi i \tilde{x}} \frac{1}{\Gamma_e\left(\tilde{x}^{\pm 2}\right)} K_{(3;0)}^{C_1}(x, \tilde{x}, z) \mathcal{I}(\tilde{x}) \quad \text{(B.54)}$$

As always this pole is coming from the contour pinching. Without loss of generality let's consider $t_1$ integration in (B.53). The integrand has a sequence of poles outside the contour at,

$$t_1^{out} = (pq)^{-\frac{1}{4}} w^{-\frac{27}{8}} z^{-1} q^{-l_1} = (pq)^{\frac{1}{4}} q w^{\frac{63}{8}} a_1^{\frac{3}{4}} q^{-l_1} \quad \text{(B.55)}$$

On the other hand remaining integrations have poles at the following positions,

$$t_2 = (pq)^{-\frac{1}{4}} w^{\frac{27}{8}} a_1^{-\frac{1}{4}} x q^{-l_2}, \ \ t_3 = (pq)^{-\frac{1}{4}} w^{\frac{27}{8}} a_1^{-\frac{1}{4}} x^{-1} q^{-l_3}, \ \ t_4 = (pq)^{\frac{1}{4}} w^{-\frac{117}{8}} a_1^{-\frac{1}{4}} q^{-l_4} \ ;$$
$$\text{and} \quad \text{(B.56)}$$
$$t_2 = (pq)^{-\frac{1}{4}} w^{\frac{27}{8}} a_1^{-\frac{1}{4}} \tilde{x} q^{-l_2}, \ \ t_3 = (pq)^{-\frac{1}{4}} w^{\frac{27}{8}} a_1^{-\frac{1}{4}} \tilde{x}^{-1} q^{-l_3}, \ \ t_4 = (pq)^{\frac{1}{4}} w^{-\frac{117}{8}} a_1^{-\frac{1}{4}} q^{-l_4} \ ;$$

Using the SU(4) condition $\prod_{i=1}^{4} t_i = 1$ both of these sets of poles can be rewritten as poles of $t_1$ but this time inside integration contour:

$$t_1^{in} = t_2^{-1} t_3^{-1} t_4^{-1} = (pq)^{\frac{1}{4}} w^{\frac{63}{8}} a_1^{\frac{3}{4}} q^{l_2 + l_3 + l_4} \quad \text{(B.57)}$$

Poles inside and outside the contour collide whenever $l_1 + l_2 + l_3 + l_4 = 1$. Thus, we have 4 contributions $(l_1, l_2, l_3, l_4) = (1, 0, 0, 0)$ or $(0, 1, 0, 0)$ or $(0, 0, 1, 0)$ or $(0, 0, 0, 1)$. We have 4! such contributions due to the permutations of the $t$'s. There is also a factor of 2 due to the sum over equal contributions of poles in the two lines of (B.56)[21]. However this overall factor is not relevant for the form of operator, and as elsewhere in the text of the paper we disregard it. Finally once we evaluate the residue at the pole in $z$ we are left only with $\tilde{x}$ integration and an overall factor of zero. This zero is cancelled noticing that $\tilde{x}$ integration contour is pinched either at $\tilde{x} = x^{\pm 1}$ or $\tilde{x} = (q^{\pm 1} x)^{\pm 1}$. Once we compute residues corresponding to these pinchings we obtain the A$\Delta$O (3.33).

## C  Derivation of the $(D_5, D_5)$ trinion

Let us discuss here the description for the sphere with two minimal and two maximal punctures that we have obtained in the previous section Figure 16. The description is $\mathrm{SU}(N + 3)$ SQCD in the middle of the conformal window with flip fields. The superpotentials correspond

---

[21]The fact that poles with $x$ and with $\tilde{x}$ contribute equally has to be shown in the detailed calculation which we leave to the interested reader

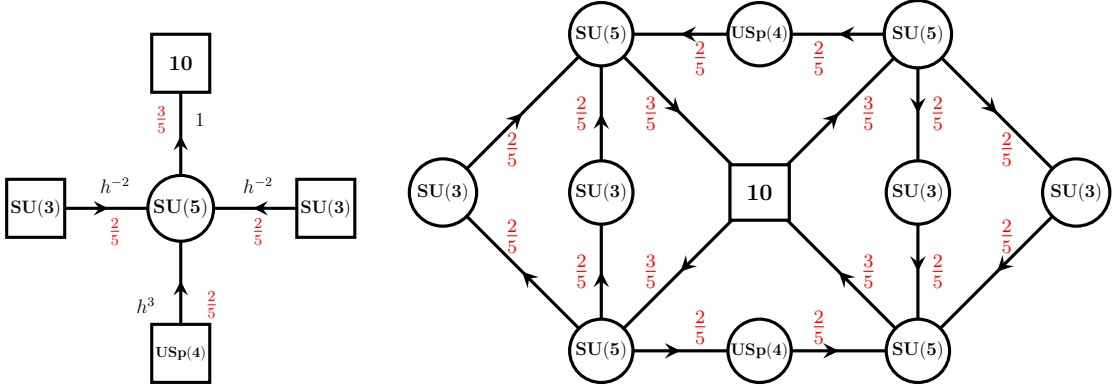

**Figure 23**. On the left we have the three punctured sphere, two maximal SU(3) punctures and one maximal USp(4) puncture, with zero flux of the $(D_5, D_5)$ minimal conformal matter theory. The theory has baryonic superpotentials consistent with the assigned charges. The superpotential is built by taking a triplet of fundamentals transforming under one of the SU(3)s, computing third antisymmetric power to build a singlet of the SU(3), and then taking the fields transforming under USp(4) to the second antisymmetric power to build a singlet under it. This superpotential is irrelevant for the trinion. On the right hand side we glue four trinions into a genus three surface. This theory has the baryonic superpotentials as well as the ones corresponding to closed loops in the quiver which are consistent with the charges. The symmetry of the theory is $\mathrm{SU}(10) \times \mathrm{U}(1)_h \subset \mathrm{SO}(20)$ where $\mathrm{SO}(2)$ is the symmetry of the 6d SCFT.

to triangles in the quiver and to certain baryonic operators: these are all the operators of R-charge two with vanishing charges for global symmetries. Note that although the triangle superpotentials are relevant the baryonic superpotentials are irrelevant for $N > 1$.

First the form of the theory after the sequence of dualities in Figure 16 (depicted also in Figure 12 in a more collapsed way) implies that conjugating one pair of minimal/maximal punctures moment maps (by introducing flip fields) to make them of the same type, the two minimal punctures and the two maximal ones appear exactly on the same footing. We expect that same type of punctures should be exchangeable by duality arguments [34], and here they are manifestly so. The model flows to a point on the conformal manifold which is invariant under the exchange of the punctures. Moreover since the superpotential distinguishing all the punctures is the irrelevant baryonic one there is no actually a direction on the conformal manifold preserving all the symmetries we want. However when we will glue these theories together in principle a conformal manifold can arise.[22] The issue is that for low enough number of punctures/"small" enough punctures/ low genus/low flux some general expectations (such as existence of exactly marginal deformations corresponding to complex structure moduli) from the 4d theory might fail.

Moreover there is another way to view this theory. As was argued in [34, 38] in general we expect that the SU(2) punctures might combine into a bigger puncture of USp type. The SU(2) punctures are obtained by partial closing of the USp(2N) puncture. These partial

---

[22]The situation is very similar to what happens in the discussion of the compactification of the minimal SU(3) SCFT [21].

closings are obtained by giving vevs to some components of the moment map (with addition of certain flip fields) and the symmetry can be broken to $USp(2k)$ with $k < N$. The claim of [34, 38] is that if we have $k \leq N$ minimal SU(2) punctures somewhere on the conformal manifold of the theory they might recombine into an $USp(2k)$ puncture. Thus for $N > 1$ we can view the theories of Figure 12 alternatively as a three punctured sphere with two maximal $SU(N + 1)$ punctures and a non maximal/minimal $USp(4)$ puncture. Note again that this symmetry is trivially manifest in the Lagrangian and is preserved by the irrelevant baryonic deformation.

For the special case of $N = 2$ the $USp(4)$ puncture is a maximal one. Thus this new description discussed here gives us a completely Lagrangian formulation of a sphere with three maximal punctures in this case. Since we obtain this theory by S-gluing two copies of identical theories the flux associated to it is zero. Starting from this sphere we can explicitly build theories corresponding to arbitrary surfaces. For example, in Figure 23 a genus three surface with zero flux is depicted. Doing so all the deformations (including the baryonic ones) become relevant/exactly marginal. In particular we can compute say the index of the genus $g$ theory and obtain,

$$\mathcal{I} = 1 + q \, p \left(3g - 3 + (g - 1)\left(1 + \mathbf{99}_{\text{SU}(10)} + \mathbf{45}_{\text{SU}(10)} \, h^6 + \overline{\mathbf{45}}_{\text{SU}(10)} \, h^{-6}\right)\right) + \cdots . \quad \text{(C.1)}$$

which is exactly as expected [94]. Note that,

$$1 + \mathbf{99}_{\text{SU}(10)} + \mathbf{45}_{\text{SU}(10)} \, h^6 + \overline{\mathbf{45}}_{\text{SU}(10)} \, h^{-6} \quad \rightarrow \quad \mathbf{190}_{\text{SO}(20)} , \quad \text{(C.2)}$$

using the decomposition of SO(20) into irreps of SU(10) × U(1). The symmetry of the 6d theory is SO(20) and as the flux is zero we expect to see it appear somewhere on the conformal manifold.

# D RS AΔO as square root of van Diejen AΔO

The van Diejen model has a simple relation to the more well know elliptic Ruijsenaars-Schneider (RS) system. Let us review here the relation. In a sense one can think of the $BC_1$ van Diejen Hamiltonian as a refined square of the $A_1$ RS Hamiltonian. The basic observation is a simple identity satisfied by the theta function: the theta functions $\theta_p(z)$ satisfy a simple "square root" identity,

$$\theta_p(z^2) = \theta_p(z)\theta_p(-z)\theta_p(p^{\frac{1}{2}}z)\theta_p(-p^{\frac{1}{2}}z) . \quad \text{(D.1)}$$

Using this identity let us consider the shift part of the $BC_1$ van Diejen operator,

$$\frac{\prod_{i=1}^{8} \theta_p((qp)^{\frac{1}{2}} h_i x)}{\theta_p(qx^2)\theta_p(x^2)} f(qx) , \quad \text{(D.2)}$$

and make the specialization of the octet of parameters,

$$\{h_i\} \rightarrow \left\{ \pm\sqrt{\frac{pq}{t}}, \pm\sqrt{\frac{q}{t}}, \pm\sqrt{\frac{p}{t}}, \pm\frac{1}{\sqrt{t}} \right\} . \quad \text{(D.3)}$$

Then (D.2) becomes

$$\frac{\theta_p(\frac{qp}{t}x^2)\theta_p(\frac{qp}{t}qx^2)}{\theta_p(x^2)\theta_p(qx^2)}f(q\,x),\tag{D.4}$$

Now let us consider the $A_1$ Ruijsenaars-Schneider operator,

$$\mathcal{O}_{RS}\cdot f(x)=\frac{\theta_p(\frac{qp}{t}x^2)}{\theta_p(x^2)}f(q^{\frac{1}{2}}x)+\frac{\theta_p(\frac{qp}{t}x^{-2})}{\theta_p(x^{-2})}f(q^{-\frac{1}{2}}x)\,,\tag{D.5}$$

and compute

$$(\mathcal{O}_{RS})^2\cdot f(x)=\frac{\theta_p(\frac{qp}{t}x^2)\theta_p(\frac{qp}{t}qx^2)}{\theta_p(qx^2)\theta_p(x^2)}\,f(qx)+\frac{\theta_p(\frac{qp}{t}x^{-2})\theta_p(\frac{qp}{t}qx^{-2})}{\theta_p(qx^{-2})\theta_p(x^{-2})}\,f(q^{-1}x)$$
$$\left(\frac{\theta_p(\frac{qp}{t}x^{-2})}{\theta_p(x^{-2})}\frac{\theta_p(\frac{qp}{t}q^{-1}x^2)}{\theta_p(q^{-1}x^2)}+\frac{\theta_p(\frac{qp}{t}x^2)}{\theta_p(x^2)}\frac{\theta_p(\frac{qp}{t}q^{-1}x^{-2})}{\theta_p(q^{-1}x^{-2})}\right)\,f(x)\,.\tag{D.6}$$

This is the $BC_1$ van Diejen operator with the parameters specialization (D.3). To see that the constant part agrees let us compute the residues at the poles of the constant piece of (D.6). We find,

$$\mathrm{Res}_{x=sq^{1/2}}\,(\mathcal{O}_{RS})^2_{const}=-s\frac{\theta_p\,(t)\,\theta_p\,(q^{-1}t)}{2q^{-\frac{1}{2}}\theta_p\,(q^{-1})\,(p;p)^2_\infty}\,,$$

$$\mathrm{Res}_{x=sq^{-1/2}}\,(\mathcal{O}_{RS})^2_{const}=s\frac{\theta_p\,(t)\,\theta_p\,(q^{-1}t)}{2q^{\frac{1}{2}}\theta_p\,(q^{-1})\,(p;p)^2_\infty}\,,$$

$$\mathrm{Res}_{x=sq^{1/2}p^{1/2}}\,(\mathcal{O}_{RS})^2_{const}=-s\frac{\theta_p\,(t)\,\theta_p\,(q^{-1}t)}{2q^{-\frac{1}{2}}p^{-\frac{1}{2}}\theta_p\,(q^{-1})\,(p;p)^2_\infty}\,,$$

$$\mathrm{Res}_{x=sq^{-1/2}p^{1/2}}\,(\mathcal{O}_{RS})^2_{const}=s\frac{\theta_p\,(t)\,\theta_p\,(q^{-1}t)}{2q^{\frac{1}{2}}p^{-\frac{1}{2}}\theta_p\,(q^{-1})\,(p;p)^2_\infty}\,,\tag{D.7}$$

where $s=\pm1$. There are also poles at $x=\pm p^{\frac{1}{2}}$ but computation yields zero residues. Using (D.1), it is easy to check that the above residues match exactly the ones in (3.7) with the specialization (D.3). Since both operators are elliptic with the same period, this means that they can differ only by x-independent function (which they do as can be checked in $p$ expansion). So we conclude that,

$$(\mathcal{O}_{RS})^2\cdot f(x)=\mathcal{O}_{VD}\cdot f(x)\tag{D.8}$$

up to x-independent function and $\mathcal{O}_{VD}$ is the van Diejen operator with the specialization (D.3).

Moreover, also the elliptic Gamma function satisfies the following "square root" identity,

$$\Gamma_e(x^2)=\Gamma_e(\pm x)\Gamma_e(\pm\sqrt{q}x)\Gamma_e(\pm\sqrt{p}x)\Gamma_e(\pm\sqrt{qp}x)\,.\tag{D.9}$$

Using this identity and specialization (D.3) on the $\Phi$-gluing integration measure in the rank one E-string case we obtain,

$$\frac{\prod_{i=1}\Gamma_e((qp)^{\frac{1}{2}}h_i x^{\pm 1})}{\Gamma_e(x^{\pm 2})} \rightarrow \frac{\Gamma_e(\frac{qp}{t}x^{\pm 2})}{\Gamma_e(x^{\pm 2})}, \tag{D.10}$$

which is the relevant $\Phi$-gluing measure for the Ruijsenaars-Schneider model. For example it is simply the contribution of the SU(2) $\mathcal{N}=2$ vector multiplet [2] (without the term coming from the Cartan of the adjoint chiral field).[23]

# E  Duality properties

In this Appendix we will provide supporting evidences for various duality properties the operators are expected to satisfy following duality arguments. Namely in Appendix E.1 we test the kernel property (3.41) claiming that the tube index (3.40) is the kernel function of the van Diejen model. In Appendix E.2 we check the commutavity of the $A_N$ operators (4.20). Finally in the Appendix E.3 we discuss the kernel property (4.32) of the tube index (4.31) and our $A_N$ operator.

## E.1  Proving kernel property for the $A_1$ tube.

In this appendix we give a proof of the kernel identity (3.41).[24] We will first write down explicit expressions for right and left sides of the relation and then prove that they are actually equal. To start we compute lhs of the equation

$$\tilde{\mathcal{O}}_x^{(z;u^{12}w^6;1,0)} \cdot \tilde{K}_{(2;0)}^{A_1}(x,y) = C\tilde{K}_{(2;1)}^{A_1}(x,y), \tag{E.1}$$

where $\tilde{K}_{(2;1)}^{A_1}(x,y)$ is the tube theory that we obtain closing $z$ puncture of $A_1$ trinion (3.8) with the defect introduced and performing conjugation similar to (3.40). It is just obtained by computing the residue of the trinion at the pole $z = (pq)^{-\frac{1}{2}}u^{-12}w^{-6}q^{-1}$. At this point the integration contour is pinched as can be seen from (3.38) with the choice $m_i = 0$, $\forall i$ and $(k_1, k_2, k_3) = (1,0,0)$ or $(0,1,0)$ or $(0,0,1)$, where we should sum over all possible partitions. This calculation leads to the following expression for the tube theory:

---

[23]Similar considerations can be used to relate USp($2Q$) Ruijsenaars-Schneider model to the $BC_Q$ van Diejen model. There for example the adjoint of USp($2Q$) can be obtained from the antisymmetric representation and the octets of fundamentals with a similar specification (again without the contribution of the Cartans in the adjoint representation).

[24]This is already considered in [70] and we include the discussion here for completeness.

$$K^{A_1}_{(2;1)}(x,y) = \frac{\Gamma_e\left(pq^2w^{12}u^{24}\right)}{2\theta_p\left((pq)^{-\frac{1}{2}}u^{-18}q^{-1}x^{\pm1}\right)}\Gamma_e\left((pq)^{\frac{1}{2}}v^6u^{12}y^{\pm1}\right)\Gamma_e\left(v^6u^{-6}x^{\pm1}y^{\pm1}\right)\times$$

$$\Gamma_e\left((pq)^{\frac{1}{2}}u^6w^{12}qx^{\pm1}\right)\prod_{j=1}^{6}\Gamma_e\left((pq)^{\frac{1}{2}}u^4v^{-2}w^{-2}a_jx^{\pm1}\right)\Gamma_e\left(u^{-14}v^{-2}w^{-2}a_j\right)+$$

$$\frac{\Gamma_e\left(pq^3w^{12}u^{24}\right)}{\theta_p\left((pq)^{\frac{1}{2}}u^{18}qx\right)\theta_p\left(x^2\right)}\Gamma_e\left((pq)^{\frac{1}{2}}v^6u^{12}qy^{\pm1}\right)\Gamma_e\left(v^6u^{-6}xy^{\pm1}\right)\Gamma_e\left(v^6u^{-6}q^{-1}x^{-1}y^{\pm1}\right)\times$$

$$\Gamma_e\left((pq)^{\frac{1}{2}}u^6w^{12}x^{-1}\right)\Gamma_e\left((pq)^{\frac{1}{2}}u^6w^{12}qx\right)\prod_{j=1}^{6}\Gamma_e\left(u^{-14}v^{-2}w^{-2}a_jq^{-1}\right)\times$$

$$\Gamma_e\left((pq)^{\frac{1}{2}}u^4v^{-2}w^{-2}a_jqx\right)\Gamma_e\left((pq)^{\frac{1}{2}}u^4v^{-2}w^{-2}a_jx^{-1}\right)+\left(x\to x^{-1}\right),$$

$$\tilde{K}^{A_1}_{(2;1)}(x,y)\equiv\Gamma_e\left((pq)^{\frac{1}{2}}u^{-6}w^{-12}x^{\pm1}\right)\Gamma_e\left((pq)^{\frac{1}{2}}v^{-6}w^{-12}y^{\pm1}\right)K^{A_1}_{(2;1)}(x,y)\qquad\text{(E.2)}$$

Calculation of the lhs of (E.1) leads to $K^{A_1}_{(2;1)}(x,y)$ with the extra constant $C$ given by

$$C = \frac{\theta_p\left(q^{-1}v^{12}u^{-12}\right)}{\theta_p\left(pqw^{12}u^{24}\right)\theta_p\left(pq^2w^{12}u^{24}\right)}\prod_{j=1}^{6}\theta_p\left(u^{-14}v^{-2}w^{-2}a_jq^{-1}\right).\qquad\text{(E.3)}$$

This constant appears since in our derivations of the tubes we were often omitting overall factors independent of $x$ and $y$. If one keeps track of all constants this coefficient should be just equal to one. In order to show relation (E.1) one needs to use the following $\theta$-function identity,

$$\frac{\theta_p\left((pq)^{\frac{1}{2}}u^{-6}v^{-12}x\right)\theta_p\left(v^6u^{-6}xy^{\pm1}\right)\theta_p\left((pq)^{\frac{1}{2}}u^{18}qx\right)}{\theta_p\left(q^{-1}v^{12}u^{-12}\right)\theta_p\left((pq)^{\frac{1}{2}}u^{12}v^6y^{\pm1}\right)\theta_p\left(qx^2\right)}+$$

$$\frac{\theta_p\left((pq)^{\frac{1}{2}}u^6v^{12}x\right)\theta_p\left(v^6u^{-6}q^{-1}x^{-1}y^{\pm1}\right)\theta_p\left((pq)^{\frac{1}{2}}u^{18}x^{-1}\right)}{\theta_p\left(q^{-1}v^{12}u^{-12}\right)\theta_p\left((pq)^{\frac{1}{2}}u^{12}v^6y^{\pm1}\right)\theta_p\left(q^{-1}x^{-2}\right)}=1.\qquad\text{(E.4)}$$

Let us provide the following proof of the identity. First of all we notice that both terms on the lhs of this expression are elliptic functions with periods $1$ and $p$. In the fundamental domains these functions have poles at $x=\pm q^{-1/2}$ and $x=\pm q^{-1/2}p^{1/2}$ as well as at $y=\left((pq)^{\frac{1}{2}}u^{18}\right)^{\pm1}$. It can be checked that the residues of two terms at these poles are exactly the same but of the opposite signs. Hence sum of these terms is just constant that does not depend neither on $x$ nor on $y$. In order to prove that this constant is actually $1$ we have to choose simple values for $x$ and $y$. For example if we choose $y=qv^{-6}u^6x$ we can see that the second term on the lhs vanishes and the first term is exactly $1$ proving the identity (E.4).

Similar calculation can be performed for the rhs of the kernel property identity (3.41). In this case by construction we should obtain:

$$\tilde{\mathcal{O}}_y^{(z;u^{12}w^6;1,0)} \cdot \tilde{K}_{(2;0)}^{A_1}(x,y) = C\tilde{K}_{(2;1)}^{A_1,dual}(x,y)\,, \tag{E.5}$$

where $\tilde{K}_{(2;1)}^{A_1,dual}(x,y,z)$ is the tube with the defect that can be obtained starting from the Seiberg dual of $A_1$ trinion (3.8). This Seiberg dual is basically the same as the original theory with extra flip singlets. The quiver of the Seiberg dual theory for general rank is shown on the Figure 11 for the higher rank case and $A_1$ expression can be obtained simply putting $N = 1$. Corresponding index is given by:

$$K_3^{A_1,dual}(x,y,z) = \frac{\kappa^2}{2}\prod_{j=1}^{6}\Gamma_e\left((pq)^{\frac{1}{2}}u^4v^{-2}w^{-2}a_jx^{\pm1}\right)\Gamma_e\left((pq)^{\frac{1}{2}}v^4u^{-2}w^{-2}a_jy^{\pm1}\right)\times$$

$$\Gamma_e\left((pq)^{\frac{1}{2}}w^4v^{-2}u^{-2}a_jz^{\pm1}\right)\oint\frac{dt_1}{2\pi it_1}\frac{dt_2}{2\pi it_2}\frac{1}{\prod\limits_{i\neq j}^{3}\Gamma_e\left(\frac{t_i}{t_j}\right)}\prod_{i=1}^{3}\Gamma_e\left((pq)^{1/6}u^{-2}v^4w^4t_ix^{\pm1}\right)\times$$

$$\Gamma_e\left((pq)^{1/6}u^4v^{-2}w^4t_iy^{\pm1}\right)\Gamma_e\left((pq)^{1/6}w^{-2}u^4v^4t_iz^{\pm1}\right)\Gamma_e\left((pq)^{1/3}u^{-2}v^{-2}w^{-2}t_i^{-1}a_j^{-1}\right)\tag{E.6}$$

As function of $x,y$ and $z$ fugacities this function is of course exactly the same as the index (3.8) of the original trinion theory. To obtain the index $\tilde{K}_{(2;1)}^{A_1,dual}(x,y)$ of this trinion we should once again capture the residue of the pole at $z = (pq)^{-\frac{1}{2}}u^{-12}w^{-6}q^{-1}$ and add flip multiplets.

$$\tilde{K}^{A_1,dual}_{(2;1)}(x,y) \equiv \Gamma_e\left((pq)^{\frac{1}{2}}u^{-6}w^{-12}x^{\pm 1}\right)\Gamma_e\left((pq)^{\frac{1}{2}}v^{-6}w^{-12}y^{\pm 1}\right)K^{A_1,dual}_{(2;1)}(x,y)\,;$$

$$K^{A_1,dual}_{(2;1)}(x,y) \equiv \sum_{k_1+k_2+k_3=1}K^{A_1,dual}_{(2;\{k_1,k_2,k_3\})}(x,y)\,,$$

$$K^{A_1,dual}_{(2;\{k_1,k_2,k_3\})}(x,y) = \Gamma_e\left(u^{-14}w^{-2}v^{-2}a_jq^{-K}\right)\Gamma_e\left(pqw^{10}u^{10}v^{-2}a_jq^{K}\right)\frac{\Gamma_e\left(y^2q^{-k_3}\right)}{\Gamma_e\left(y^2q^{k_2-k_3}\right)}\times$$

$$\frac{\Gamma_e\left(y^{-2}q^{-k_2}\right)}{\Gamma_e\left(y^{-2}q^{k_3-k_2}\right)}\frac{\Gamma_e\left((pq)^{\frac{1}{2}}u^{12}w^{12}v^{-6}q^{k_2+k_3}y^{\pm 1}\right)\Gamma_e\left((pq)^{-\frac{1}{2}}w^{-12}u^{-12}v^6y^{-1}q^{-k_2-K}\right)}{\Gamma_e\left(\left((pq)^{-\frac{1}{2}}u^{12}w^{12}v^{-6}yq^{2k_2+k_3}\right)^{\pm 1}\right)\Gamma_e\left(\left((pq)^{\frac{1}{2}}u^{12}w^{12}v^{-6}y^{-1}q^{2k_3+k_2}\right)^{\pm 1}\right)}\times$$

$$\Gamma_e\left((pq)^{-\frac{1}{2}}u^{-12}w^{-12}v^6yq^{-k_3-K}\right)\Gamma_e\left((pq)^{\frac{1}{2}}u^6w^{12}x^{\pm 1}q^{k_2+k_3}\right)\Gamma_e\left(u^{-6}v^6yx^{\pm 1}q^{-k_3}\right)\times$$

$$\Gamma_e\left(u^{-6}v^6y^{-1}x^{\pm 1}q^{-k_2}\right)\Gamma_e\left((pq)^{\frac{1}{2}}u^{12}v^6y^{-1}q^{K-k_2}\right)\Gamma_e\left((pq)^{\frac{1}{2}}u^{12}v^6yq^{K-k_3}\right)\times$$

$$\Gamma_e\left(pqu^{24}w^{12}q^{K+k_2+k_3}\right)\left[\prod_{j=1}^{6}\Gamma_e\left((pq)^{\frac{1}{2}}u^2w^2v^{-4}a_j^{-1}q^{k_2}y\right)\Gamma_e\left((pq)^{\frac{1}{2}}u^2w^2v^{-4}a_j^{-1}q^{k_3}y^{-1}\right)\times\right.$$

$$\left.\Gamma_e\left(u^{-10}w^{-10}v^2a_j^{-1}q^{-k_2-k_3}\right)\Gamma_e\left((pq)^{\frac{1}{2}}u^4v^{-2}w^{-2}a_jx^{\pm 1}\right)\Gamma_e\left((pq)^{\frac{1}{2}}v^4u^{-2}w^{-2}a_jy^{\pm 1}\right)\right]\,, \quad \text{(E.7)}$$

Computing lhs of (E.5) we can check that indeed we get the tube $\tilde{K}^{A_1,dual}_{(2;1)}(x,y)$ with extra constant $C$ given by (E.3).

Now in order to check the kernel property (3.41) we just need to show that the two expressions (E.2) and (E.7) for the tube theories indices are actually equal *i.e.* $\tilde{K}^{A_1}_{(2;1)}(x,y) = \tilde{K}^{A_1,dual}_{(2;1)}(x,y)$. In principle their equality follows directly from the fact that they are derived closing punctures of Seiberg-dual theories in exactly the same way. However we can provide an independent proof for it. Using explicit expressions (E.2) and (E.7) equality of the tube expressions reduces to the following $\theta$-function identity:

$$F_1(x,y) = F_2(x,y)\,,$$

$$F_1(x,y) \equiv \frac{\theta_p\left((pq)^{\frac{1}{2}}u^6w^{12}x^{\pm 1}\right)}{\theta_p\left((pq)^{-\frac{1}{2}}u^{-18}q^{-1}x^{\pm 1}\right)}\prod_{j=1}^{6}\theta_p\left(u^{-14}v^{-2}w^{-2}a_jq^{-1}\right) + \theta_p\left(pq^2w^{12}u^{24}\right)\times$$

$$\theta_p\left((pq)^{\frac{1}{2}}v^6u^{12}y^{\pm 1}\right)\left[\frac{\theta_p\left((pq)^{\frac{1}{2}}u^6w^{12}x\right)\prod_{j=1}^{6}\theta_p\left((pq)^{\frac{1}{2}}u^4v^{-2}w^{-2}a_jx\right)}{\theta_p\left(v^6u^{-6}q^{-1}x^{-1}y^{\pm 1}\right)\theta_p\left((pq)^{\frac{1}{2}}u^{18}qx\right)\theta_p\left(x^2\right)} + \left(x\to x^{-1}\right)\right]\,,$$

$$\text{(E.8)}$$

$$F_2(x,y) \equiv \frac{\theta_p\left((pq)^{\frac{1}{2}}u^{12}v^6y^{\pm1}\right)}{\theta_p\left((pq)^{-\frac{1}{2}}u^{-12}w^{-12}v^6q^{-1}y^{\pm1}\right)} \prod_{j=1}^{6} \theta_p\left(pqw^{10}u^{10}v^{-2}a_j\right) + \theta_p\left(pq^2w^{12}u^{24}\right) \times$$

$$\theta_p\left((pq)^{\frac{1}{2}}u^6w^{12}x^{\pm1}\right)\left[\frac{\theta_p\left((pq)^{\frac{1}{2}}u^{12}v^6y\right)\prod_{j=1}^{6}\theta_p\left((pq)^{\frac{1}{2}}u^2w^2v^{-4}a_j^{-1}y\right)}{\theta_p\left(y^2\right)\theta_p\left(v^6u^{-6}q^{-1}y^{-1}x^{\pm1}\right)\theta_p\left((pq)^{\frac{1}{2}}u^{12}w^{12}v^{-6}qy\right)} + \left(y\to y^{-1}\right)\right].$$

$$(E.9)$$

Let us provide a proof of this identity. For this we notice that both $F_1(x,y)$ and $F_2(x,y)$ are elliptic functions in both $x$ and $y$ with periods $1$ and $p$. In the fundamental domain both these functions have poles at $x = \left(v^6u^{-6}q^{-1}y^{\pm1}\right)^{\pm1}$. The residues of the functions at these poles are equal to:

$$\text{Res}_{x=v^6u^{-6}q^{-1}y^s}F_1(x,y) = \text{Res}_{x=v^6u^{-6}q^{-1}y^s}F_2(x,y) =$$
$$\frac{\theta_p\left(pq^2w^{12}u^{24}\right)\theta_p\left((pq)^{\frac{1}{2}}v^6w^{12}q^{-1}y^s\right)\theta_p\left((pq)^{\frac{1}{2}}u^{12}v^6y^{-s}\right)}{u^6v^{-6}qy^{-s}(p;p)_\infty^2\,\theta_p\left(y^{-2s}\right)\theta_p\left(v^{12}u^{-12}q^{-2}y^{2s}\right)} \prod_{j=1}^{6}\theta_p\left((pq)^{\frac{1}{2}}v^4u^{-2}w^{-2}a_jq^{-1}y^s\right),$$
$$\text{Res}_{x=v^{-6}u^6qy^s}F_1(x,y) = \text{Res}_{x=v^{-6}u^6qy^s}F_2(x,y) =$$
$$-\frac{\theta_p\left(pq^2w^{12}u^{24}\right)\theta_p\left((pq)^{\frac{1}{2}}v^6w^{12}q^{-1}y^{-s}\right)\theta_p\left((pq)^{\frac{1}{2}}u^{12}v^6y^s\right)}{u^{-6}v^6q^{-1}y^{-s}(p;p)_\infty^2\,\theta_p\left(y^{2s}\right)\theta_p\left(v^{12}u^{-12}q^{-2}y^{-2s}\right)} \prod_{j=1}^{6}\theta_p\left((pq)^{\frac{1}{2}}v^4u^{-2}w^{-2}a_jq^{-1}y^{-s}\right),$$

$$(E.10)$$

where $s = \pm1$. It also looks like there are poles at $x = \left((pq)^{\frac{1}{2}}u^{18}q\right)^{\pm1}$, $x = \pm1$, $x = \pm p^{1/2}$, $y = \left((pq)^{\frac{1}{2}}u^{12}w^{12}v^{-6}q\right)^{\pm1}$, $y = \pm1$ and $y = \pm p^{1/2}$. However accurate analysis shows that the residues of both $F_1(x,y)$ and $F_2(x,y)$ at these putative poles are actually zero so that there are actually no poles at these points.

As we see both $F_1(x,y)$ and $F_2(x,y)$ are elliptic functions in $x$ and $y$ with identical sets of poles and residues. This means that they differ at most by a constant independent of $x$ and $y$. Now we need to check some values of $x$ and $y$ to find this constant. The easiest way to do it is to find zeroes of functions. For example both $F_1(x,y)$ and $F_2(x,y)$ have zeroes at $x = (pq)^{-\frac{1}{2}}u^{-6}w^{-12}$ and $y = (pq)^{-\frac{1}{2}}v^{-6}u^{-12}$. Since their zeroes coincide we can finally conclude that indeed $F_1(x,y) = F_2(x,y)$ leading to the equality between the tube indices $\tilde{K}^{A_1}_{(2;1)}(x,y)$ and $\tilde{K}^{A_1,dual}_{(2;1)}(x,y)$. Hence right hand sides of both (E.1) and (E.5) are equal leading to the proof of the desired kernel property (3.41). Here we provided proof of a particular kernel property with the fixed choice of the operator and the tube theories. The latter ones are also defined by the way we close puncture of the trinion. However any other tube theory can be used as the kernel function of any A$\Delta$O derived in the Section (3.2). The proof in each of these cases is identical to the one we presented in the this appendix.

## E.2 Discussion of commutation relations of the $A_N$ operators

Now let's move to the duality properties of the $A_N$ operator (4.20). In this Appendix in particular we will give evidence of the commutativity relations (4.26) of all of these operators. For this purpose it would be useful to know how the shifts of variables of the form

$$x_m \to q x_m \,, x_l \to q^{-1} x_l \,, \quad \text{and} \quad x_m \to p x_m \,, x_l \to p^{-1} x_l \,. \tag{E.11}$$

act on various parts of operators (4.20). These shifts are precisely the ones realized by the shift operators $\Delta_{lm}^{(1,0)}$ and $\Delta_{lm}^{(0,1)}$ defined in (4.8).

First of all let's notice that all constant parts (4.22) are invariant under these shifts:

$$\Delta_{lm}^{(0,1)}(x) W^{(\xi_i;1,0;\alpha)}(x,h) = W^{(\xi_i;1,0;\alpha)}(x,h) \,,$$
$$\Delta_{lm}^{(1,0)}(x) W^{(\xi_i;0,1;\alpha)}(x,h) = W^{(\xi_i;0,1;\alpha)}(x,h) \,, \quad \forall \, i \,. \tag{E.12}$$

where $\alpha$ index can take values "$f.$" and "$n.f.$" distinguishing between flipped and non-flipped operators. For convenience we have also introduced corresponding index for the constant parts $W(x,h)$. While in non-flipped case constant part is given by (4.22) in case of flip we just postulated according to (4.25)

$$W^{(\xi_i;1,0;f.)}(x,h) \equiv W^{(\xi_i;1,0;n.f.)}\left(x^{-1}, h^{-1}\right) \,. \tag{E.13}$$

Also here and everywhere further in this section we omit the $(a)$ index of $x$ variables for brevity.

Functions $\tilde{A}_{lm}^{(\xi_i;1,0)}(x)$ defined in (4.21) are not invariant under the shifts but rather acquire extra prefactor as follows:

$$\Delta_{nr}^{(0,1)} \tilde{A}_{lm}^{(\xi_i;1,0)}(x) = \alpha_{nrlm}^{(1,0)}(x) \tilde{A}_{lm}^{(\xi_i;1,0)}(x) \,, \quad \Delta_{nr}^{(1,0)} \tilde{A}_{lm}^{(\xi_i;0,1)}(x) = \alpha_{nrlm}^{(0,1)}(x) \tilde{A}_{lm}^{(\xi_i;0,1)}(x) \,, \tag{E.14}$$

where

$$\alpha_{nrlm}^{(1,0)}(x) = \begin{cases} (-1)^{N+1} p^{-1} q^{N+2} \tilde{h}^{-1/2} x_l^{-N-3} \,, & n \neq m \,, r = l \,; \\ (-1)^{N+1} q^{-1} p^{N+2} \tilde{h}^{-1/2} x_m^{-N-3} \,, & n = m \,, r \neq l \,; \\ 1 \,, & n, r \neq m, l \,; \\ (-1)^{N+1} (pq)^{-N-2} \tilde{h}^{1/2} x_l^{N+3} \,, & r \neq m \,, n = l \,; \\ (-1)^{N+1} pq \tilde{h}^{1/2} x_m^{N+3} \,, & n \neq l \,, r = m \,; \\ (pq)^{N+1} \tilde{h}^{-1} (x_m x_l)^{-N-3} \,, & l = r \,, n = m \,; \\ (pq)^{-N-1} \tilde{h} (x_m x_l)^{N+3} \,, & l = n \,, r = m \,; \end{cases} \tag{E.15}$$

and $\alpha_{nrlm}^{(0,1)}$ can be obtained from the expressions above by simple exchange of $p$ and $q$ parameters. Here we use the same notations of the Section 4. Namely $\tilde{h}_i$ are U(1) charges of the moment maps of the puncture operator acts on and $\tilde{h} = \prod_j \tilde{h}_j$. Notice that the prefactor $\alpha$ is independent of the $i$ index of the operator, *i.e.* of the choice of moment map we give vev to in order to introduce the defect.

Now using expressions above we can find the following commutators:

$$\left[\tilde{\mathcal{O}}_x^{(\xi_i;1,0;\alpha)},\ \tilde{\mathcal{O}}_x^{(\xi_j;0,1;\beta)}\right] = \sum_{l\neq m}^{N+1}\sum_{n\neq r}^{N+1}\left[\tilde{A}_{lm}^{(\xi_i;1,0)}\Delta_{lm}^{(1,0)}\tilde{A}_{nr}^{(\xi_j;0,1)}\Delta_{nr}^{(0,1)}-\right.$$

$$\left.\tilde{A}_{lm}^{(\xi_j;0,1)}\Delta_{lm}^{(0,1)}\tilde{A}_{nr}^{(\xi_i;1,0)}\Delta_{nr}^{(1,0)}\right]+\sum_{l\neq m}^{N+1}\left[\tilde{A}_{lm}^{(\xi_i;1,0)}\Delta_{lm}^{(1,0)}W^{(\xi_j;0,1;\beta)}-\right.$$

$$\left. W^{(\xi_j;0,1;\beta)}\tilde{A}_{lm}^{(\xi_i;1,0)}\Delta_{lm}^{(1,0)} + W^{(\xi_i;1,0;\alpha)}\tilde{A}_{lm}^{(\xi_j;0,1)}\Delta_{lm}^{(0,1)} - \tilde{A}_{lm}^{(\xi_j;0,1)}\Delta_{lm}W^{(\xi_i;1,0;\alpha)}\right],\quad\text{(E.16)}$$

where as usually $\alpha$ and $\beta$ indices can take values "$f$." and "$n.f$." distinguishing between flipped and non-flipped operators. Since all constant parts $W^{(\xi_i;\dots)}$ are invariant under corresponding shifts as specified in (E.12) we can immediately see that the second term in the expression above is zero. The remaining term in the first line can be rewritten using (E.14) in the following form

$$\left[\tilde{\mathcal{O}}_x^{(\xi_i;0,1;\alpha)},\ \tilde{\mathcal{O}}_x^{(\xi_j;1,0;\beta)}\right] = \sum_{l\neq m}^{N+1}\sum_{n\neq r}^{N+1}\left(\alpha_{lmnr}^{(0,1)}-\alpha_{nrlm}^{(1,0)}\right)\tilde{A}_{lm}^{(\xi_i;1,0)}\tilde{A}_{nr}^{(\xi_j;0,1)}\Delta_{lm}^{(1,0)}\Delta_{nr}^{(0,1)}.\quad\text{(E.17)}$$

Finally using explicit expressions for $\alpha$ coefficients from (E.15) we can see that the difference in the brackets above is zero for each term in the sum. Hence we conclude

$$\left[\tilde{\mathcal{O}}_x^{(\xi_i;0,1;\alpha)},\ \tilde{\mathcal{O}}_x^{(\xi_j;1,0;\beta)}\right] = 0,\qquad \forall\ i,j\,.\quad\text{(E.18)}$$

This result is expected from duality arguments and says that two shift operators constructed by closing two punctures in various ways (i.e. exchange in $p$ and $q$ shifts as well as baryon vevs used to close puncture) commute with each other.

Finally we should prove that the commutators $\left[\tilde{\mathcal{O}}_x^{(\xi_i;1,0;\alpha)},\ \tilde{\mathcal{O}}_x^{(\xi_j;1,0;\beta)}\right]$ and $\left[\tilde{\mathcal{O}}_x^{(\xi_i;0,1;\alpha)},\ \tilde{\mathcal{O}}_x^{(\xi_j;0,1;\beta)}\right]$ are also zero. It is enough to show that one of them vanishes and the other one will follow automatically since they differ only by the exchange of parameters $p\leftrightarrow q$. Proving this kind of commutation identities is more complicated since in this case we can not use periodicity of theta functions. Let's directly study the following expression $\left[\tilde{\mathcal{O}}_x^{(\xi_i;1,0;\alpha)},\ \tilde{\mathcal{O}}_x^{(\xi_j;0,1;\beta)}\right]\mathcal{I}(x)$, where $\mathcal{I}(x)$ is some arbitrary function and both $\xi_i$ and $\xi_j$ can be either flipped or not flipped moment maps. In order to approach this calculation we will consider separately prefactors of various possible shifts of $\mathcal{I}(x)$ function. Each such prefactor has to be independently equal to zero in order for the whole commutation to work.

*1) Term* $\mathcal{I}(q^{-1}x_l,qx_m,q^{-1}x_n,qx_r)$ *with* $l\neq m\neq n\neq r$. Such contribution originates from the action of the product of two $\tilde{A}_{lm}\Delta_{lm}$ terms with different indices in the shift operators. Also since in shift parts of operators (4.20) we sum over all possible indices we should consider four different possible combinations of this indices:

$$\left[\tilde{\mathcal{O}}_x^{(\xi_i;1,0;\alpha)},\ \tilde{\mathcal{O}}_x^{(\xi_j;1,0;\beta)}\right]\mathcal{I}(x) \sim \left(\tilde{A}_{lm}^{(\xi_i;1,0)}(x)\tilde{A}_{nr}^{(\xi_j;1,0)}(q^{-1}x_l,qx_m,\dots)+\right.$$

$$\tilde{A}_{nr}^{(\xi_i;1,0)}(x)\tilde{A}_{lm}^{(\xi_j;1,0)}(q^{-1}x_n,qx_r,\dots)+\tilde{A}_{nm}^{(\xi_i;1,0)}(x)\tilde{A}_{lr}^{(\xi_j;1,0)}(q^{-1}x_n,qx_m,\dots)+$$

$$\left.\tilde{A}_{lr}^{(\xi_i;1,0)}(x)\tilde{A}_{nm}^{(\xi_j;1,0)}(q^{-1}x_l,qx_r,\dots)-(\xi_i\leftrightarrow\xi_j)\right)\mathcal{I}(q^{-1}x_l,qx_m,q^{-1}x_n,qx_r)\,.\quad\text{(E.19)}$$

Substituting expressions for the $\tilde{A}_{lm}$ functions from (4.21) we can see that this part of the commutator is zero when the following theta function identity is valid:

$$F^{\xi_i}_{lmnr}(x)F^{\xi_j}_{nrlm}(q^{-1}x_l, qx_m) + F^{\xi_i}_{nmlr}(x)F^{\xi_j}_{lrnm}(q^{-1}x_n, qx_m) +$$
$$F^{\xi_i}_{lrnm}(x)F^{\xi_j}_{nmlr}(q^{-1}x_l, qx_r) + F^{\xi_i}_{lmnr}(x)F^{\xi_j}_{nrlm}(q^{-1}x_l, qx_m) -$$
$$F^{\xi_j}_{lrnm}(x)F^{\xi_i}_{nmlr}(q^{-1}x_l, qx_r) - F^{\xi_j}_{lmnr}(x)F^{\xi_i}_{nrlm}(q^{-1}x_l, qx_m) -$$
$$F^{\xi_j}_{lmnr}(x)F^{\xi_i}_{nrlm}(q^{-1}x_l, qx_m) - F^{\xi_j}_{nmlr}(x)F^{\xi_i}_{lrnm}(q^{-1}x_n, qx_m) = 0 \,, \qquad \text{(E.20)}$$

where the functions $F^{\xi_i}_{lmnr}$ are defined as follows

$$F^{\xi_i}_{lmnr}(x) \equiv \frac{\theta_p\left((pq)^{\frac{1}{2}}\xi_i\tilde{h}^{\frac{1}{4}}x_n\right)\theta_p\left((pq)^{\frac{1}{2}}\xi_i\tilde{h}^{-\frac{1}{4}}x_n^{-1}\right)\theta_p\left((pq)^{\frac{1}{2}}\xi_i\tilde{h}^{\frac{1}{4}}x_r\right)\theta_p\left((pq)^{\frac{1}{2}}\xi_i\tilde{h}^{-\frac{1}{4}}x_r^{-1}\right)}{\theta_p\left(q\frac{x_m}{x_l}\right)\theta_p\left(\frac{x_m}{x_l}\right)\theta_p\left(\frac{x_n}{x_l}\right)\theta_p\left(\frac{x_m}{x_n}\right)\theta_p\left(\frac{x_r}{x_l}\right)\theta_p\left(\frac{x_m}{x_r}\right)} \,,$$

$$\text{(E.21)}$$

We do not provide a proof of the condition for the $F$-functions specified above. However we do check it in the series expansion in $q$ and $p$ instead for each pair of $i$ and $j$ indices. Indeed the check up to the order $O(p^3q^3)$ strongly suggests that the relation is true and hence this contribution to the commutator is just zero.

*2) Term* $\mathcal{I}(q^2x_m, q^{-1}x_l, q^{-1}x_n)$ *with* $l \neq m \neq n$. This kind of terms come from the action of the operators $\tilde{A}^{(\xi_i;1,0)}_{lm}(x)\Delta^{(1,0)}_{lm}\tilde{A}^{(\xi_j;1,0)}_{nm}(x)\Delta^{(1,0)}_{nm}$ with $n \neq l \neq m$. Summing over various possible combinations of indices we obtain the following contributions:

$$\left[\tilde{\mathcal{O}}^{(\xi_i;1,0;\alpha)}_x, \tilde{\mathcal{O}}^{(\xi_j;1,0;\beta)}_x\right]\mathcal{I}(x) \sim \left(\tilde{A}^{(\xi_i;1,0)}_{lm}(x)\tilde{A}^{(\xi_j;1,0)}_{nm}(q^{-1}x_l, qx_m, \dots) +\right.$$
$$\left.\tilde{A}^{(\xi_i;1,0)}_{nm}(x)\tilde{A}^{(\xi_j;1,0)}_{lm}(q^{-1}x_n, qx_m, \dots) - (\xi_i \leftrightarrow \xi_j)\right)\mathcal{I}(q^2x_m, q^{-1}x_l, q^{-1}x_n) \qquad \text{(E.22)}$$

In order for this combination to be zero the following identity has to be satisfied

$$G^{\xi_i}_{lmn}(x)G^{\xi_j}_{nml}\left(q^{-1}x_l, qx_m, \dots\right) + G^{\xi_i}_{nml}(x)G^{\xi_j}_{lmn}\left(q^{-1}x_n, qx_m, \dots\right) -$$
$$G^{\xi_j}_{lmn}(x)G^{\xi_i}_{nml}\left(q^{-1}x_l, qx_m, \dots\right) - G^{\xi_j}_{nml}(x)G^{\xi_i}_{lmn}\left(q^{-1}x_n, qx_m, \dots\right) = 0 \,, \qquad \text{(E.23)}$$

where

$$G^{\xi_i}_{lmn}(x) \equiv \frac{\theta_p\left((pq)^{\frac{1}{2}}\xi_i\tilde{h}^{-\frac{1}{4}}x_n^{-1}\right)\theta_p\left((pq)^{\frac{1}{2}}\xi_i\tilde{h}^{\frac{1}{4}}x_n\right)}{\theta_p\left(q\frac{x_m}{x_l}\right)\theta_p\left(\frac{x_m}{x_l}\right)\theta_p\left(\frac{x_n}{x_l}\right)\theta_p\left(\frac{x_m}{x_n}\right)} \qquad \text{(E.24)}$$

Using expansion in $p$ and $q$ series we can check that the identity above is indeed valid and hence considered terms do not contribute to the commutator.

*3) Term* $\mathcal{I}(q^{-2}x_l, qx_m, qx_r)$ *with* $l \neq m \neq r$. This term is similar to the previous term. It comes from the following terms in the commutator

$$\left[\tilde{\mathcal{O}}^{(\xi_i;1,0;\alpha)}_x, \tilde{\mathcal{O}}^{(\xi_j;1,0;\beta)}_x\right]\mathcal{I}(x) \sim \left(\tilde{A}^{(\xi_i;1,0)}_{lm}(x)\tilde{A}^{(\xi_j;1,0)}_{lr}(q^{-1}x_l, qx_m, \dots) +\right.$$
$$\left.\tilde{A}^{(\xi_i;1,0)}_{lr}(x)\tilde{A}^{(\xi_j;1,0)}_{lm}(q^{-1}x_l, qx_r, \dots) - (\xi_i \leftrightarrow \xi_j)\right)\mathcal{I}(q^{-2}x_l, qx_m, qx_r) \qquad \text{(E.25)}$$

In order for this combination to be zero the following identity has to be satisfied

$$
G^{\xi_i}_{lmr}(x)G^{\xi_j}_{lrm}\left(q^{-1}x_l,\, qx_m,\dots\right) + G^{\xi_i}_{lrm}(x)G^{\xi_j}_{lmr}\left(q^{-1}x_l,\, qx_r,\dots\right) -
$$
$$
G^{\xi_j}_{lmr}(x)G^{\xi_i}_{lrm}\left(q^{-1}x_l,\, qx_m,\dots\right) - G^{\xi_j}_{lrm}(x)G^{\xi_i}_{lmr}\left(q^{-1}x_l,\, qx_r,\dots\right) = 0\,, \qquad \text{(E.26)}
$$

where $G^{\xi_i}_{lmn}(x)$ are the previously defined functions. Once again we checked the validity of the relations written above using $p$ and $q$ expansion.

4) *Term* $\mathcal{I}(q^{-2}x_l,\, q^2 x_m)$. This term comes from the contributions of the form

$$
\left[\tilde{\mathcal{O}}^{(\xi_i;1,0;\alpha)}_x,\, \tilde{\mathcal{O}}^{(\xi_j;1,0;\beta)}_x\right]\mathcal{I}(x) \sim \left(\tilde{A}^{(\xi_i;1,0)}_{lm}(x)\tilde{A}^{(\xi_j;1,0)}_{lm}(q^{-1}x_l, qx_m,\dots) -\right.
$$
$$
\left. \tilde{A}^{(\xi_j;1,0)}_{lm}(x)\tilde{A}^{(\xi_i;1,0)}_{lm}(q^{-1}x_l, qx_m,\dots)\right)\mathcal{I}(q^{-2}x_l, q^2x_m) \qquad \text{(E.27)}
$$

This term is obviously zero since shifts act only on $x_l$ and $x_m$ in $\tilde{A}^{(\xi_i;1,0)}_{lm}(x)$. The parts that depend on $x_l$ and $x_m$ in these expressions do not depend on the choice of the moment map $\xi_i$ we give vev to. Hence it is obvious that this term does not contribute to the commutator.

5) *Constant term* $\mathcal{I}(x)$. This term in its nature is similar to the previous one and comes from

$$
\left[\tilde{\mathcal{O}}^{(\xi_i;1,0;\alpha)}_x,\, \tilde{\mathcal{O}}^{(\xi_j;1,0;\beta)}_x\right]\mathcal{I}(x) \sim \left(\tilde{A}^{(\xi_i;1,0)}_{lm}(x)\tilde{A}^{(\xi_j;1,0)}_{ml}(q^{-1}x_l, qx_m,\dots) -\right.
$$
$$
\left. \tilde{A}^{(\xi_j;1,0)}_{lm}(x)\tilde{A}^{(\xi_i;1,0)}_{ml}(q^{-1}x_l, qx_m,\dots)\right)\mathcal{I}(x)\,. \qquad \text{(E.28)}
$$

Just as in the previous term an expression above is zero since shifts act only on $x_l$ and $x_m$ in $\tilde{A}^{(\xi_i;1,0)}_{lm}(x)$ and hence there is no dependence on the moment map $\xi_i$ we give vev to.

6) *Term* $\mathcal{I}(q^{-1}x_l, qx_m)$. This term is the most complicated one. Unlike previous terms it comes not only from the commutation of shift parts but also from the commutation between constant terms $W^{(\xi_i;1,0;\alpha)}(x)$ and shift parts $\tilde{A}^{(\xi_i;1,0)}(x)$. In particular we have the following terms,

$$
\left[\tilde{\mathcal{O}}^{(\xi_i;1,0;\alpha)}_x,\, \tilde{\mathcal{O}}^{(\xi_j;1,0;\beta)}_x\right]\mathcal{I}(x) \sim \left(\sum_{n\neq m\neq l}^{N+1}\left[\tilde{A}^{(\xi_i;1,0)}_{ln}(x)\tilde{A}^{(\xi_j;1,0)}_{nm}\left(q^{-1}x_l, qx_n\right) +\right.\right.
$$
$$
\tilde{A}^{(\xi_i;1,0)}_{nl}(x)\tilde{A}^{(\xi_j;1,0)}_{mn}\left(q^{-1}x_n, qx_l\right) - \tilde{A}^{(\xi_j;1,0)}_{ln}(x)\tilde{A}^{(\xi_i;1,0)}_{nm}\left(q^{-1}x_l, qx_n\right) -
$$
$$
\left.\tilde{A}^{(\xi_j;1,0)}_{nl}(x)\tilde{A}^{(\xi_i;1,0)}_{mn}\left(q^{-1}x_n, qx_l\right)\right] + \tilde{A}^{(\xi_i;1,0)}_{lm}(x)W^{(\xi_j;1,0;\beta)}\left(q^{-1}x_l, qx_m\right) +
$$
$$
W^{(\xi_i;1,0;\alpha)}(x)\tilde{A}^{(\xi_j;1,0)}_{lm}(x) - \tilde{A}^{(\xi_j;1,0)}_{lm}(x)W^{(\xi_i;1,0;\alpha)}\left(q^{-1}x_l, qx_m\right) -
$$
$$
\left.W^{(\xi_j;1,0;\beta)}(x)\tilde{A}^{(\xi_i;1,0)}_{lm}(x)\right)\mathcal{I}(q^{-1}x_l, qx_m) \qquad \text{(E.29)}
$$

Although we do not provide a proof, we once again have checked $(p,q)$ series expansion of the expression above for the ranks $N$ up to $N = 6$ and up to the order $p^2q^2$. As a result we find that contribution of these terms are also zero concluding our proof of all commutation relations (4.26). On top of this there is another check of all the relations for functions $F_{nlmr}(x)$, $G_{nlm}(x)$ and also the relation above. Instead of expansion in both $p$ and $q$ we can set $p \to 0$

keeping $q$ finite. This will simplify all theta functions down to the simple rational functions using the limit of the theta function $\lim_{p \to 0} \theta_p(x) \to (1-x)$. Then we can check that all the relations above hold to any order in $q$. For the expression above we have checked it for the ranks $N$ up to $N = 6$.

As a result we find that all of the contributions to the commutator $\left[ \tilde{\mathcal{O}}_x^{(\xi_i;1,0;\alpha)}, \tilde{\mathcal{O}}_x^{(\xi_j;1,0;\beta)} \right]$ are zero. This concludes our evidence for all commutation relations (4.26).

## E.3 Discussion of the $A_N$ kernel property

Now let us discuss the kernel property (4.32) involving $A_N$ operators (4.20) we derived and the tube index (4.31). The line of arguments here will be exactly the same as in $A_1$ case presented in Appendix E.1. We will start with writing left and right sides of equation (4.32) precisely and then show that the result we get in the two cases is actually the same.

Here we will show how the procedure works for the operator obtained by closing $SU(2)_z$ punctures with the vev of $\mathbf{1}_{(wu^{N+1})^{2N+4}}$ baryon. In this case the left hand side of the kernel identity (4.32) we obtain

$$\tilde{\mathcal{O}}_x^{(\xi;1,0;f.)} \cdot \tilde{K}_{(2;0)}^{A_N}(x,y) = C \tilde{K}_{(2;1)}^{A_N}(x,y), \qquad \xi \equiv \left( wu^{N+1} \right)^{2N+4}, \tag{E.30}$$

where the operator $\tilde{\mathcal{O}}_x^{(\xi;1,0;f.)}$ is given by (4.11) and (4.7). The tube $\tilde{K}_{(2;0)}^{A_N}(x,y)$ is given by (4.31). It is observed from the $A_N$ trinion (4.28) after closing $SU(2)_z$ puncture with the same $\mathbf{1}_{(wu^{N+1})^{2N+4}}$ baryonic vev. Finally $\tilde{K}_{(2;1)}^{A_N}(x,y)$ is the tube also obtained from the same $A_N$ trinion closing $SU(2)_z$ minimal puncture with the defect introduced. It corresponds to capturing the residue of the pole of the trinion theory index located at $z = (pq)^{-\frac{1}{2}} \left( wu^{(N+1)} \right)^{-(2N+4)} q^{-1}$. This pole comes from the pinching of the contour at the points specified in (4.29) and (4.30) with the choice

$$m_i = 0 \; \forall \; i, \qquad k_i = 1, \; k_j = 0 \; \forall \; j \neq i. \tag{E.31}$$

Substituting these values $t_i$ from (4.29) and (4.30) Summing over all possible partitions specified above we arrive to the following expression for the tube:

$$K_{(2;1)}^{A_N}(x,y)(x,y) = K_{\vec{k}_{N+2}}(x,y) + \sum_{m=1}^{N+1} K_{\vec{k}_m}(x,y), \tag{E.32}$$

$$\tilde{K}_{(2;1)}^{A_1}(x,y) \equiv \prod_{i=1}^{N+1} \Gamma_e \left( (pq)^{\frac{1}{2}} \left( u^N w^2 \right)^{-2N-4} x_i \right) \Gamma_e \left( (pq)^{\frac{1}{2}} \left( v^N w^2 \right)^{-2N-4} y_i \right) K_{(2;1)}^{A_N}(x,y),$$

where the first and the second term in the first line of the equation above correspond to the partitions with $k_{N+2} = 1$ and $k_m = 1$ correspondingly. Expressions are given by:

$$T^{\vec{k}_m}(x,y) = \prod_{l=1}^{2N+4} \Gamma_e\left(u^{-(N+1)(2N+5)}v^{-N-1}w^{-2}a_l q^{-1}\right) \times \times$$

$$\frac{\Gamma_e\left((pq)^{\frac{1}{2}}w^{4N+8}u^{N(2N+4)}x_m^{-1}\right)}{\theta_p\left((pq)^{\frac{1}{2}}u^{(N+2)(2N+4)}x_m q\right)} \prod_{i\neq m}^{N+1} \frac{1}{\theta_p\left(\frac{x_m}{x_i}\right)} \prod_{i,j=1;i\neq m}^{N+1} \Gamma_e\left(\left(vu^{-1}\right)^{2N+4}y_j x_i^{-1}\right) \times$$

$$\left[\prod_{i\neq m}^{N+1} \Gamma_e\left((pq)^{\frac{1}{2}}w^{4N+8}u^{N(2N+4)}x_i^{-1}q\right) \prod_{l=1}^{2N+4} \Gamma_e\left((pq)^{\frac{1}{2}}u^{N+3}v^{-N-1}w^{-2}x_i a_l\right)\right] \times$$

$$\left[\prod_{i=1}^{N+1} \Gamma_e\left(\left(vu^{-1}\right)^{2N+4}y_i x_m^{-1}q^{-1}\right) \Gamma_e\left((pq)^{\frac{1}{2}}v^{2N+4}u^{(N+1)(2N+4)}y_i q\right)\right] \times$$

$$\Gamma_e\left(pq^3 w^{4N+8}u^{2(N+1)(2N+4)}\right) \prod_{l=1}^{2N+4} \Gamma_e\left((pq)^{\frac{1}{2}}u^{N+3}v^{-N-1}w^{-2}x_m q a_l\right) \,,$$

$$T^{\vec{k}_{N+2}}(x,y) = \Gamma_e\left(pq^2 w^{4N+8}u^{2(N+1)(2N+4)}\right) \prod_{l=1}^{2N+4} \Gamma_e\left(u^{-(N+1)(2N+5)}v^{-N-1}w^{-2}a_l\right) \times$$

$$\prod_{i,j=1}^{N+1} \Gamma_e\left(\left(vu^{-1}\right)^{2N+4}y_j x_i^{-1}\right) \prod_{i=1}^{N+1} \Gamma_e\left((pq)^{\frac{1}{2}}w^{4N+8}u^{N(2N+4)}x_i^{-1}q\right) \times$$

$$\frac{\Gamma_e\left((pq)^{\frac{1}{2}}v^{2N+4}u^{(N+1)(2N+4)}y_i\right)}{\theta_p\left((pq)^{-\frac{1}{2}}u^{-(N+2)(2N+4)}x_i^{-1}q^{-1}\right)} \prod_{l=1}^{2N+4} \Gamma_e\left((pq)^{\frac{1}{2}}u^{N+3}v^{-N-1}w^{-2}x_i a_l\right) \,. \tag{E.33}$$

Finally the constant factor $C$ in (E.30) is given by

$$C = \frac{\theta_p\left(q^{-1}(u^{-1}v)^{(N+1)(2N+4)}\right) \prod_{j=1}^{2N+4} \theta_p\left(q^{-1}u^{-(N+1)(2N+5)}v^{-N-1}w^{-2}a_j\right)}{\theta_p\left(pqu^{(N+1)(4N+8)}w^{4N+8}\right) \theta_p\left(pq^2 u^{(N+1)(4N+8)}w^{4N+8}\right)}, \tag{E.34}$$

In order to get (E.30) one needs to use the following intricate relation

$$\alpha_m + \sum_{l\neq m}^{N+1} \beta_{lm} = 1, \qquad \forall m \tag{E.35}$$

where

$$\alpha_m \equiv \frac{\prod\limits_{j=1}^{N+1} \theta_p\left((u^{-1}v)^{2N+4} y_j x_m^{-1} q^{-1}\right)}{\theta_p\left(q^{-1}(u^{-1}v)^{(N+1)(2N+4)}\right)} \times$$

$$\frac{\theta_p\left((pq)^{\frac{1}{2}} u^{2N+4} v^{(N+1)(2N+4)} x_m\right) \prod\limits_{i\neq m} \theta_p\left((pq)^{\frac{1}{2}} u^{(N+2)(2N+4)} x_i\right)}{\prod\limits_{j=1}^{N+1} \theta_p\left((pq)^{\frac{1}{2}} u^{(N+1)(2N+4)} v^{2N+4} y_j\right) \prod\limits_{i\neq m} \theta_p\left(q^{-1} x_i x_m^{-1}\right)}, \qquad \text{(E.36)}$$

$$\beta_{lm} \equiv \frac{\theta_p\left((pq)^{\frac{1}{2}} u^{(N+2)(2N+4)} q x_m\right) \theta_p\left((pq)^{\frac{1}{2}} u^{-2N-4} v^{-(N+1)(2N+4)} x_l^{-1}\right)}{\theta_p\left(q^{-1}(u^{-1}v)^{(N+1)(2N+4)}\right) \prod\limits_{j=1}^{N+1} \theta_p\left((pq)^{\frac{1}{2}} u^{(N+1)(2N+4)} v^{2N+4} y_j\right)} \times$$

$$\frac{\prod\limits_{i\neq m\neq l} \theta_p\left((pq)^{\frac{1}{2}} u^{(N+2)(2N+4)} x_i\right) \prod\limits_{j=1}^{N+1} \theta_p\left((u^{-1}v)^{2N+4} x_l^{-1} y_j\right)}{\theta_p\left(q x_m x_l^{-1}\right) \prod\limits_{i\neq m\neq l} \theta_p\left(x_i x_l^{-1}\right)}. \qquad \text{(E.37)}$$

We do not provide a proof for relation (E.35). Instead, as usual, we have checked it term by term in $p$ expansion for several low ranks $N$. These checks strongly suggest the validity of the required relation.

Now we should perform similar computation for the right hand side of the kernel property (4.32). In this case we obtain

$$\tilde{\mathcal{O}}_y^{(\xi;1,0;n.f.)} \cdot \tilde{K}_{(2;0)}^{A_N}(x,y) = C \tilde{K}_{(2;1)}^{A_N,dual}(x,y), \qquad \xi \equiv \left(wu^{N+1}\right)^{2N+4}, \qquad \text{(E.38)}$$

where expression for the operator can be extracted from (4.20) putting $a=v$ and $x_i^{(v)} \equiv y_i$. Notice that while on the r.h.s. (E.30) of the kernel equality we were required to close with the flip, on the l.h.s. specified above closure is performed with no flips. Finally $\tilde{K}_{(2;1)}^{A_N,dual}(x,y)$ is the tube obtained closing the minimal puncture of Seiberg-dual trinion theory with the defect introduced. This Seiberg-dual theory has the following index:

$$K_3^{A_N,dual} = \kappa_{N+1} \prod_{i=1}^{N+1} \prod_{l=1}^{2N+4} \Gamma_e\left((pq)^{\frac{1}{2}} u^{N+3} v^{-N-1} w^{-2} a_l x_i\right) \times$$

$$\Gamma_e\left((pq)^{\frac{1}{2}} v^{N+3} u^{-N-1} w^{-2} a_l y_i\right) \Gamma_e\left((pq)^{\frac{1}{2}} (uv)^{-N-1} w^{2N+2} a_l z^{\pm 1}\right) \oint \prod_{i=1}^{N+1} \frac{dt_i}{2\pi i t_i} \prod_{i\neq j}^{N+2} \frac{1}{\Gamma_e\left(\frac{t_i}{t_j}\right)} \times$$

$$\prod_{i=1}^{N+2} \prod_{j=1}^{N+1} \prod_{l=1}^{2N+4} \Gamma_e\left((pq)^{\frac{1}{2(N+2)}} u^{-2} w^4 v^{2(N+1)} x_j^{-1} t_i\right) \Gamma_e\left((pq)^{\frac{1}{2(N+2)}} v^{-2} w^4 u^{2(N+1)} y_j^{-1} t_i\right) \times$$

$$\Gamma_e\left((pq)^{\frac{1}{2(N+2)}} w^{-2N} (uv)^{2(N+1)} t_i z^{\pm 1}\right) \Gamma_e\left((pq)^{\frac{N+1}{2(N+2)}} (uv)^{-N-1} w^{-2} a_l^{-1} t_i^{-1}\right), \qquad \text{(E.39)}$$

The quiver of this theory is shown on the Figure 11. It is also $\mathrm{SU}(N+1)$ gauge theory with $(2N+4)$ flavors and some extra flip singlets.

Starting from the index (E.39) we can close $\mathrm{SU}(2)_z$ minimal puncture using the very same baryonic vev $\mathbf{1}_{(wu^{N+1})^{2N+4}}$. This once again corresponds to capturing the residue of the pole of (E.39) located at $z = (pq)^{-\frac{1}{2}} \left( wu^{(N+1)} \right)^{-(2N+4)} q^{-1}$. Performing usual calculation we obtain the following index of the tube theory:

$$
\tilde{K}^{A_N,dual}_{(2;1)}(x,y) \equiv \prod_{i=1}^{N+1} \Gamma_e\left( (pq)^{\frac{1}{2}} \left( u^N w^2 \right)^{-2N-4} x_i \right) \Gamma_e\left( (pq)^{\frac{1}{2}} \left( v^N w^2 \right)^{-2N-4} y_i \right) \times
$$

$$
K^{A_N,dual}_{(2;1)}(x,y); \qquad\qquad K^{A_N,dual}_{(2;1)}(x,y) \equiv \sum_{\sum k_i = 1} K^{A_N,dual}_{(2;\vec{k})}(x,y),
$$

$$
K^{A_N,dual}_{(2;\vec{k})}(x,y) = \prod_{i=1}^{N+1}\prod_{l=1}^{2N+4} \Gamma_e\left( (pq)^{\frac{1}{2}} u^{N+3} v^{-N-1} w^{-2} a_l x_i \right) \prod_{i\neq j}^{N+1} \frac{\Gamma_e\left( \frac{y_i}{y_j} q^{-k_i} \right)}{\Gamma_e\left( \frac{y_i}{y_j} q^{k_j-k_i} \right)} \times
$$

$$
\Gamma_e\left( (pq)^{\frac{1}{2}} v^{N+3} u^{-N-1} w^{-2} a_l y_i \right) \Gamma_e\left( u^{-(N+1)(2N+3)} v^{N+1} w^{-2(2N+3)} a_l q^{-K} \right) \times
$$

$$
\Gamma_e\left( pq u^{(N+1)(2N+5)} v^{N+1} w^2 a_l q^K \right) \Gamma_e\left( (pq)^{\frac{1}{2}} \left( vw^{-2} u^{-N-1} \right)^{-2(N+2)} y_i^{-1} q^{K-k_{N+2}} \right) \times
$$

$$
\frac{\Gamma_e\left( (pq)^{-\frac{1}{2}} \left( vw^{-2} u^{-N-1} \right)^{2(N+2)} y_i q^{-K-k_i} \right)}{\Gamma_e\left( \left( (pq)^{\frac{1}{2}} \left( vw^{-2} u^{-N-1} \right)^{-2(N+2)} y_i^{-1} q^{K-k_{N+2}+k_i} \right)^{\pm 1} \right)} \times
$$

$$
\Gamma_e\left( (pq)^{\frac{1}{2}} w^{4(N+2)} u^{2N(N+2)} x_i^{-1} q^{K-k_{N+2}} \right) \Gamma_e\left( (pq)^{\frac{1}{2}} u^{2(N+1)(N+2)} v^{2(N+2)} y_i q^{K-k_i} \right) \times
$$

$$
\prod_{l=1}^{2N+4} \Gamma_e\left( (pq)^{\frac{1}{2}} w^2 u^{N+1} v^{-N-3} y_i^{-1} q^{k_i} a_l^{-1} \right) \Gamma_e\left( pq w^{4(N+2)} u^{4(N+1)(N+2)} q^{2K-k_{N+2}} \right) \times
$$

$$
\prod_{j=1}^{N+1} \Gamma_e\left( \left( vu^{-1} \right)^{2(N+2)} y_i x_j^{-1} q^{-k_i} \right) \Gamma_e\left( v^{N+1} u^{-(N+1)(2N+3)} w^{-2(2N+3)} a_l^{-1} q^{k_{N+2}-K} \right)
$$

$$
\tag{E.40}
$$

Finally the constant $C$ in (E.38) is the same constant defined in (E.34). Deriving (E.38) requires identity similar to (E.35) which we once again check in expansion for low ranks.

Comparing left (E.30) and right (E.38) sides of the kernel function property (4.32) we can finally see that its validity reduces to the equality $\tilde{K}^{A_N}_{(2;1)}(x,y) = \tilde{K}^{A_N,dual}_{(2;1)}(x,y)$ between indices of the tubes obtained from Seiberg dual theories. Since indices of the Seiberg dual theories are equal [28], and we compute the residue of the same pole located

at $z = (pq)^{-\frac{1}{2}} \left(wu^{(N+1)}\right)^{-(2N+4)} q^{-1}$ expressions we obtain should be the same functions of $x$ and $y$. So the required equality should work by construction. Relying on this argument we conclude that the tube index $\tilde{K}^{A_N}_{(2;0)}(x,y)$ is indeed a kernel function of our $A_N$ operator (4.20). Just as in $A_1$ case the same proof can be used for AΔOs and tube theories obtained by closing punctures in all other possible ways.

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
