# Peer review of "Minimal $(D,D)$ conformal matter and generalizations of the van Diejen model"

_SciPost Physics_

## Round 2 · Referee Report · Anonymous · 2021-9-18

Report
The authors have extended the previously known connection between 6d superconformal field theories and elliptic relativistic quantum integrable models. To derive elliptic integrable models associated to a given 6d SCFT the authors compactify the theory on a punctured Riemann surface to four dimensions. The authors study theories where the 4d $\mathcal{N}=1$ theory obtained by compactification on a sphere with two maximal and one minimal punctures (a trinion) has a known Lagrangian description. Specifically, the authors compute the supersymmetric index of two trinion theories, both with at least one maximal puncture, and the 4d theory obtained by gluing these two trinions along the maximal punctures. The indices are observed to be related via analytic difference operators, and these operators are then related to the operators of one-dimensional integrable models.
The authors begin with a focus on compactifications of the E-string theory. The integrable model associated to these compactifications is the $BC_1$ van Diejen model. The authors study the E-string integrable model by utilizing the maximal punctures associated to three different five-dimensional gauge theory descriptions after compactifying the E-string on $S^1$. These are $SU(2)$, $SU(2)$, and $USp(2)$ gauge theories. In each case, they derive the analytic difference operator, and again in each case they show that they are equivalent to the $BC_1$ van Diejen operator. The authors refer to the integrable models associated to each of these 5d descriptions as the $A_1$, $A_1^1$, and $C_1$ models, respectively.
The E-string can be considered as the first entry in an infinite sequence of 6d $(1,0)$ SCFTs known as $(D_{N+3}, D_{N+3})$ minimal conformal matter, where the E-string occurs when $N=1$. When compactifed on an $S^1$ these theories have $SU(N+1)$, $USp(2N)$, and $SU(2)^N$ gauge theory descriptions. Each of these five dimensional descriptions defines a different kind of maximal puncture for the compactification to 4d. The authors study the 6d SCFT compactified on the trinion with maximal punctures associated to the first gauge theory description in great detail, and they obtain the analytic difference operators for the associated integrable model, which they call the $A_N$ model -- a generalization of the $BC_1$ van Diejen model.
The authors include an extensive set of technical appendices to aid the reader in reproducing their results, and which the referee can attest are particularly helpful.
This is a well-written paper and a significant addition to the canon of work exploring the connection between six-dimensional superconformal field theories and one-dimensional elliptic integrable models. I recommend this paper for publication in SciPost Physics.

---

## Round 2 · Referee Report · Anonymous · 2021-10-21

Strengths
This paper implements a systematic approach to find difference operators acting on the
index of various 4d N=1 theories obtained from 6d SCFT on punctured Riemann surfaces. The idea consists in gluing to a maximal puncture a trinion with zero flux and subsequently close another puncture of the trinion by means of a space dependent vev for an operator charged under the puncture symmetry. To meet the zero flux condition authors actually join two conjugated trinions to form a four punctured sphere which is glued to the selected puncture. Then there is a choice of vevs to close two of the remaining punctures. Closing all punctures with space-independent vevs is equivalent to the identity operator. Turning on space dependent vevs for one of the two punctures that are being closed yields a difference operator acting on the fugacity of the remaining open puncture. The operator is labelled by the vevs that have been considered.
They focus on the $(D_{N+3}, D_{N+3})$ family of 6d SCFT which coincides for N=1 with E-string, which was previously associated to the van Diejen (vD) model. For N=1 they consider three trinions with different flux. The difference operator associated to each trinion, is always the BC1 vD model, although the identification between the integral model and the gauge theory parameters is different in each case.
Since the vD operator are known to commute, the authors obtain very nice consistency checks of the geometric realisation of the theory by compactifcation on punctured Riemann surfaces. The commutativity is indeed related to the choice of duality frames or different ways of decomposing surfaces. A more advanced check would involve the kernel property of the trinion which they can prove for the simpler case with one closed puncture, that is for tubes.
The first trinion has an higher rank generalisation with SU(N+1) maximal punctures, the second a generalisation with SU(2)^N punctures and the last one with USP(2N) and are supposed to provide higher rank
A_N, A_1^N and C_N extension of the vD model. They focus on the A_N case.
Weaknesses
There are no serious weak points, the notation could be improved in various points as suggested below.
Report
This paper is very interesting and deserve to published in this Journal.
Requested changes
-In eq. 2.5 $s$ is used as the power of q (power of derivative) and as the number of punctures in C_{g,s}, maybe use a different letter.
- The discussion leading to eqs. 3.19 and 3.21, is quite hard to follow, perhaps because of the notation and should be improved. For example the variables $\tilde h_i^{(a)}$ and $ \tilde h_a$ in 3.19 are not clearly defined. It is also not obvious why $ \tilde h_i^{(a)} \tilde h_a^{-1/4}$ (with $a$ referring to either $x$ or $y$ punctures) is the label of the operator, which before was labelled by the charges of the moment map used to close the $z$ puncture.
-After 3.21 authors say that if $\tilde h_i =u^{-6} w^{-12}$, eq. 3.13 is recovered, but there is no $\tilde h_i$ in eq. 3.21, perhaps the condition is $ \tilde h_i^{(a)} \tilde h_a^{-1/4}=u^{-6} w^{-12}$?
-Figure 6 is not discussed in the main tex. In fact it could be useful to mention that to get this result from the gluing of two trinions several Seiberg dualities have been applied as in the case of figure 16.
-Eq. 2.7 states the kernel property. Here a generic difference operator can act either on the puncture u or z of I(z,u) which is not defined but from fig. 3 it seems to refer to the index of a generic surface with two max puncture and a closed minimal puncture, perhaps to be more consistent with their previous notation should refer to I(z,u) as
$I [C_{g,s}(u,z), F ]$.
-Eq. 3.35 states the kernel property for trinions. The difference operator is labelled by $z$ and $u^{12} w^6$. From the notation used before this would suggest that the puncture $z$ is closed by the vev of an operator with charge $u^{12} w^6?$, but then we would get a tube as in the case later in 3.38 while the kernel property should hold with the $z$ puncture open as well.

---

## Round 3 · Author Response

In the present resubmitted version we have implemented minor corrections suggested by the referees of the initial submission. Below in the List of Changes we briefly go through the changes and answer referees questions.

---

## Round 3 · List of Changes

1. Question: In eq. 2.5 s is used as the power of q (power of derivative) and as the
number of punctures in Cg,s , maybe use a different letter.

Answer: We fixed this by changing s → m in the power of q.

2. Question: The discussion leading to eqs. 3.19 and 3.21, is quite hard to follow, perhaps because of the notation and should be improved. For example the $h_i^{(a)}$ and $h_a$ in 3.19 are not clearly defined. It is also not obvious why $h^{(a)}_ih^{-1/4}_a$ (with a referring to either x or y punctures) is the label of the operator, which before was labelled by the charges of the moment map used to close the z puncture.

Answer: Thank you for noticing that $h^{(a)}$ variables where not defined explicitly. We now added explicit explanation in the sentences just before eq. 3.18. Regarding $h^{(a)}_ih^{-1/4}_a$ label of the index. It is indeed the charge of the puncture we close. It is not independent of the charges of the punctures we act on ($h_i^{(a)}$) so we choose to express everything in terms of one single set of charges. To be more clear we added explanation about this relation after eq.3.18.

3. Question: After eq. 3.21 authors say that if $h_i=u^{-6}w^{-12}$, eq. 3.13 is recovered, but there is no $h_i$ in eq. 3.21, perhaps the condition is $h^{(a)}_ih^{-1/4}_a=u^{-6}w^{-12}$?

Answer: To be more precise here the condition should be $a=u$ and choosing $h_i^{(u)}=u^{-6}w^{-12}$. We added this specification after eq. 3.21 (3.22 in the previous version)

4. Question: To be more precise here the condition should be $a=u$ and choosing $h_i^{(u)}=u^{-6}w^{-12}$. We added this specification after eq. 3.21 (3.22 in the previous version)

Answer: To be more precise here the condition should be $a=u$ and choosing $h_i^{(u)}=u^{-6}w^{-12}$. We added this specification after eq. 3.21 (3.22 in the previous version)

5. Question: Eq. 2.7 states the kernel property. Here a generic difference operator can act either on the puncture $u$ or $z$ of $I(z,u)$ which is not defined but from fig. 3 it seems to refer to the index of a generic surface with two max puncture and a closed minimal puncture, perhaps to be more consistent with their previous notation should refer to $I(z,u)$ as $I[C_{g,s}(u,z),F]$.

Answer: Kernel property is true for any index of any theory obtained compactifying on a surface with two or more maximal punctures, i.e. having at least two global symmetries of certain type. Closure of the minimal puncture we show on Fig. 3 is used to derive this property. But the property itself is only right part of this Figure. Hence we called this general index ${\cal I}(u,z)$ with $u$ and $z$ referring to two global symmetries (or equivalently punctures) we can act on. However we agree that the notation proposed by the referee is even better and we performed suggested modification in the corresponding equation.

6. Question: Eq. 3.35 states the kernel property for trinions. The difference operator is labelled by $z$ and $u^{12}w^6$. From the notation used before this would suggest that the puncture $z$ is closed by the vev of an operator with charge $u^{12}w^6$?, but then we would get a tube as in the case later in 3.38 while the kernel property should hold with the $z$ puncture open as well.

Answer: The difference operators we get are indeed obtained by closing minimal punctures with space-dependent vevs of the moment maps. However the trinions we act on in eq.3.36 have nothing to do with closing procedure. Instead of trinions there can be tube theories or any other theory corresponding to compactification on a surface with at least two maximal punctures. So there is no need to close $\SU(2)_z$ puncture of the trinion we act on. We specify trinions as kernel functions since they together with the tubes are the simplest examples and building blocks for all other kernel functions.

You are currently on this page

Resubmission 2106.08335v3 on 5 November 2021

---

## Editorial Decision

publication_decision_taken:_accept